# Parameter Optimization in Sea Ice Models with Elastic-Viscoplastic Rheology

Gleb Panteleev[1], Max Yaremchuk[1], J. N. Stroh[2], Oceana P. Francis[3], and Richard Allard[1]

[1]Naval Research Laboratory, Stennis Space Center, MS, USA
[2]University of Colorado, Aurora, CO, USA
[3]University of Hawaii, Honolulu, HI, USA

**Correspondence:** Gleb Panteleev (gleb.panteleev@nrlssc.navy.mil)

**Abstract.**

The modern sea ice models include multiple parameters which strongly affect model solution. As an example, in the CICE6 community model, rheology and landfast grounding/arching effects are simulated by functions of the sea ice thickness and concentration with a set of fixed parameters empirically adjusted to optimize the model performance. In this study, we consider extension of a two-dimensional EVP sea ice model using a spatially variable representation of these parameters. The feasibility of optimization of the landfast sea ice parameters and rheological parameters is assessed via idealized variational data assimilation experiments with synthetic observations of ice concentration, thickness and velocity. The experiments are configured for a 3-day data assimilation window in a rectangular basin with variable wind forcing.

The tangent linear and adjoint models featuring EVP rheology are found to be unstable, but can be stabilized by adding a Newtonian damping term into the adjoint equations. A set of observation system simulation experiments shows that landfast parameter distributions can be reconstructed after 5-10 iterations of the minimization procedure. Optimization of sea ice initial conditions and spatially varying parameters in the stress tensor equation requires more computation, but provides a better hindcast of the sea ice state and the internal stress tensor. Analysis of inaccuracy in the wind forcing and errors in sea ice thickness observations show reasonable robustness of the variational DA approach and the feasibility of its application to available and incoming observations.

## 1 Introduction

Due to the significant decline of sea ice volume and the increase of maritime activity in the Arctic Ocean over the past decades, an accurate sea ice hindcast/forecast has become an important component of the data assimilation systems for the region. Currently, there are several community sea ice models broadly used for modeling and/or reconstruction of the Arctic Ocean state through various Data Assimilation (DA) algorithms. Many of these models (e.g. Heimbach et al., 2010; Zhang and Rothrock, 2003; Vancoppenolle et al., 2009; Massonnet et al., 2015) are based on the visco-plastic (VP) rheology proposed

by Hibler (1979). In the last decades, several numerical approaches have been proposed for solving the VP problem (Hibler, 1979; Zhang and Hibler, 1997; Lemieux et al., 2008). These approaches are based on implicit solvers and require a significant number of iterations to achieve full convergence. Application of more efficient quasi-Newtonian solvers suffers from the lack of robustness and these are usually applied in the sea ice models of intermediate resolution (Lemieux et al., 2012; Losch et al., 2014). In addition, implicit VP solvers are less convenient for implementing on massively parallel supercomputer architectures. Despite these inconveniences, the currently known four-dimensional variational (4dVar) sea ice models employ an implicit VP solver for time integration of the tangent linear and adjoint (TLA) models (Kauker et al., 2009; Heimbach et al., 2010).

The Elastic-VP (EVP) rheology was proposed as an alternative explicit method which can be easily adopted for supercomputer architectures (Hunke and Duckowicz, 1997). In addition to the VP parameterization of the internal stress tensor, EVP includes an additional elastic term which requires internal subcycling to damp elastic oscillations in order to achieve the VP solution. The most popular sea ice model with EVP rheology is the Community sea ICE model (CICE, Hunke et al., 2010), which is currently maintained and developed by a group of institutions in North America and Europe known as the CICE Consortium. This model is widely used in sea ice and coupled sea-ice modeling (e.g. Posey et al., 2010, Metzger et al., 2014, Yaremchuk et al., 2019), and there are multiple examples of 3dVar, ensemble-based DA systems utilizing the CICE model (e.g., Zhang and Bitz, 2018). However, to our knowledge, there is no 4dVar DA system based on the CICE model yet.

CICE6 includes several parameterizations of the sea ice rheology including the formulation of Hibler (1979). This parameterization includes three major parameters ($P^*$, $e$, and $\alpha$), describing, respectively, the dimensional maximum ice strength per unit thickness, the ratio of yield ellipse major axes, and the non-dimensional scaling of ice strength with its compactness. While this parameterization is not the default option in CICE5/6, it is still widely used in sea ice modeling and DA applications at the Naval Research Laboratory.

For modeling landfast ice near the coast and in straits (the location of a so-called arching phenomena), an additional parameter $k_T$ has been introduced to model the tensile strength (Konig Beatty and Holland, 2010). This parameter is absent in the traditional (i.e. Hibler 1979) elliptical yield curve formulation. Lemieux et al., (2015) proposed a number of additional parameters $k_1, k_2, \alpha_b$, for better parameterization of the landfast ice grounded in the shallow regions. Formally, these land fast ice parameters are not related to the sea ice rheology. To simplify the presentation, we shall refer the entire set $\{P^*, e, k_t, k_1, k_2\}$ as rheological parameters (RPs), suggesting that all of them influence sea ice dynamics.

Typically, the above mentioned rheological parameters are constants and their values are defined empirically from multiple numerical experiments. RPs such as $P^*$ and $e$ reflect the model parameterization rather than physics and are not directly observable (Kreyscher et al., 2000), but are nevertheless known to range within certain limits (Harder and Fischer 1999). As a few examples, the typical values of $P^*$ determined from sea ice drift were diagnosed to vary within 27.5 kN/m$^2$ (Hibler III and Walsh, 1982), 15–20 kN/m$^2$ (Kreyscher et al., 1997, 2000), and 30–45 kN/m$^2$ (Tremblay and Hakakian, 2006). These studies indicate the existence of significant variations of $P^*$ estimates, which may be attributed to both non-physical considerations (such as spatially variable model resolution), and spatio-temporal variations of Arctic sea ice.

The numerical experiments of Lemieux et al. (2016) using a coarse resolution pan-Arctic CICE-NEMO model have shown that $k_T = 0.2$ provides the best agreement with landfast strength observations in the Kara Sea, when the ellipse axes ratio ranges

within 1.2–1.4. The most sensitive parameters for sea ice grounding are the critical thickness parameter $k_1$ and maximum basal stress parameter $k_2$. The optimal values were found to be 8 and 15 N/m$^3$, respectively, with higher sensitivity with respect to $k_1$ (Lemieux et al., 2015). However, fixed values of $k_T, k_1$ and $k_2$ cannot provide a universally good performance of landfast modeling for different parts of the Arctic Ocean, suggesting that these parameters are a function of local environmental conditions. The physical properties of sea ice also strongly depend on sea ice salinity, temperature, and ice age (e.g. Anderson and Weeks, 1958), indicating that rheological properties may vary between different sea ice categories.

Thus, numerous modeling experiments and sea ice observations (e.g., Juricke et al., 2013; Toyota and Kimura, 2018) indicate that spatially varying and properly optimized RPs should significantly improve the sea ice model performance. There were multiple attempts to define sea ice model parameters in an optimal way. The early attempts followed a traditional "trial-and-error" approach, involving multiple runs of a sea ice model with different sets of RPs (e.g. Miller et al., 2006, Uotila et al., 2012), while others utilized more advanced methods based on the Green's function approach (Nguyen et al., 2011), ensemble Kalman filtering (Massonnet et al., 2014) and genetic algorithms (Sumata et al., 2019). However, all of these attempts sought a relatively small set of spatially invariant sea ice model parameters in order to provide an optimal sea ice model solution for a period of several years or decades. The application of these algorithms for optimizing spatially varying RPs was not considered and, from our point of view, is not straightforward due to high computational overhead. Also note, that the above listed algorithms are not suitable for simultaneous optimization of other model parameters such as initial and open boundary conditions, or external forcing fields.

The major objective of our study is to find a numerically feasible method for simultaneous optimization of multiple model parameters including the spatially varying RPs, initial conditions and forcing fields in the sea ice models based on EVP solvers (e.g. CICE model). The framework of an NRL research project to identify spatially varying Land Fast Ice Parameters in the CICE6 model guided the priorities and objectives of this study. We suggest that, if successful, this approach can be adopted to optimize RPs in operational sea ice models (e.g. Posey et al 2010, Metzger et al 2014) and provide a more accurate short term (3–7 days) sea ice forecast.

The feasibility of reconstructing spatially varying fields of $P^*$ and $e$ (as well as other model parameters) through the variational assimilation of synthetic observations of Sea Ice Velocity/Concentration/Thickness (SIV/SIC/SIT) was recently analyzed by Stroh et al. (2019) in the framework of a 1d (zonal) VP sea ice model. It was found that variational DA allows for a reasonable reconstruction of spatially varying $P^*$ and $e$ in regions with strong convergence and significantly improves short range hindcast/forecast of the sea ice state. In particular, it was shown that optimization of spatially varying $P^*$ and $e$ provides more accurate reconstruction of ridging areas, which cannot be achieved by optimizing the initial sea ice state only.

Note that optimization of RPs through the 4Dvar DA approach allows us to efficiently use all available sea ice observations, including observations of sea ice velocity, which are rarely used for assimilation in sea ice DA systems controlled by the initial conditions only. This is due to weak sensitivity of the sea ice state with respect to sea ice velocity initial conditions (e.g. Kauker, et al., 2009). Augmenting the 4dVar control vector with RPs allows us to use sea ice velocity observations for consistent adjustment of the RPs and/or atmospheric forcing, providing a better sea ice forecast (Stroh et al., 2019).

In this study, we extend the investigation to analyze the feasibility of RP optimization within a more advanced 2d sea ice model based on the EVP rheology formulation of Lemieux et al. (2016). Our analysis is based on application of the 4dVar and Observing System Simulation Experiment (OSSEs) (e.g.Nitta, 1975; Arnold and Dey, 1986; Nichols 2003, 2010) and follows the conventional twin-data experiment approach (e.g. Goldberg and Heimbach, 2013).

Similarly to Stroh et al., (2019), we developed the corresponding EVP TLA models and analyzed the feasibility of optimizing spatially varying ellipse ratio and sea ice compressive strength. In addition, we also analyzed the effects of optimizing two of the landfast sea ice parameters introduced by Lemieux et al. (2016). Through multiple OSSEs, we evaluate the quality of the RP reconstruction and analyze the impact of spatially varying RPs on the sea ice state. A similar approach was recently proposed for the optimization of the basal stress parameters in an ice sheet model (Goldberg and Heimbach, 2013).

Currently, satellite sea ice observations are typically available daily with a reasonably dense spatial resolution. Analysis of SAR images (e.g. Panteleev et al., 2019) indicates that in the marginal sea ice zone, pancake/cake ice with floe sizes of ∼1-20 m may be easily replaced by floes exceeding 1 km in size in one week. As a consequence, we configured the OSSEs with a 3-day DA window assuming that sea ice features do not move very far from their initial position during this period. This assumption suggests that such an approach should have more impact on the short term sea ice forecast.

The paper is organized as follows: Section 2 describes the implemented sea ice model, the details of the TLA codes and generation of synthetic observations and the first guess solution used in OSSEs. Results of these experiments are described in Sections 3 (optimization of landfast sea ice parameters) and 4 (optimization of compressive strength and ellipse ratio), with a special focus on the feasibility of optimizing spatially varying RPs in the context of present and future observational coverage of sea ice at high latitudes. Section 5 summarizes the work and discusses directions of future research.

## 2 Sea ice model and its 4dVar implementation

This section provides details of the sea ice model formulation, its associated linearizations, outlines the variational data assimilation system used for optimizing model parameters, and describes synthetic observations used to do so. To distinguish between the parameter fields spatially varying in 2d and the fixed parameter values that were not subject to optimization, the latter are marked by tildes. The major parameters of the model are listed in Table 1.

### 2.1 EVP sea ice model

#### 2.1.1 Formulation

In the present study, we employed the sea ice model formulation of Lemieux et al. (2016) with the basal stress parameterization and generalized Hibler (1979) yield curve. Equations of the model describe EVP ice physics coupled with sea ice dynamics

which is forced by the stresses $\boldsymbol{\tau}$ exerted on ice through its interaction with the bottom $\boldsymbol{\tau}_b$, atmosphere $\boldsymbol{\tau}_a$, and the ocean $\boldsymbol{\tau}_w$:

$$\tilde{\rho} h A (\partial_t + \tilde{f} \boldsymbol{k} \times) \boldsymbol{u} = \mathrm{div} \boldsymbol{\sigma} + \boldsymbol{\tau}_b + \boldsymbol{\tau}_a + \boldsymbol{\tau}_w \tag{1}$$

$$\tilde{T}_d \, \partial_t \boldsymbol{\sigma} + e^2 \boldsymbol{\sigma} + \tfrac{1}{2} \mathrm{tr} \boldsymbol{\sigma} (1 - e^2) \, \boldsymbol{I} = P_p \left[ (1 + k_T) \dot{\boldsymbol{\epsilon}} - \tfrac{\Delta}{2} (1 - k_T) \, \boldsymbol{I} \right] / \Delta^* \tag{2}$$

$$\partial_t h = \mathrm{div}(h \boldsymbol{u}) \tag{3}$$

$$\partial_t A = \mathrm{div}(A \boldsymbol{u}) \tag{4}$$

Here div, tr are the divergence and trace operators, $\boldsymbol{k}$ is the vertical unit vector, $\boldsymbol{I}$ is the $2 \times 2$ identity matrix, $h, A$, and $\boldsymbol{u} = \{u, v\}$ are the 2d fields of ice effective thickness, concentration and velocity, and $\boldsymbol{\sigma}$ and $\dot{\boldsymbol{\epsilon}}$ are the 2d fields of ice stress and the deformation rate tensors:

$$\boldsymbol{\sigma} = \begin{bmatrix} \sigma_{xx} & \sigma_{xy} \\ \sigma_{xy} & \sigma_{yy} \end{bmatrix}; \quad \dot{\boldsymbol{\epsilon}} = \frac{1}{2} \begin{bmatrix} 2 \partial_x u & \partial_x v + \partial_y u \\ \partial_x v + \partial_y u & 2 \partial_y v \end{bmatrix} \tag{5}$$

The scalar field $\Delta$ used for normalizing the rhs in (2) is computed using the following expression (e.g., Hunke, 2001):

$$\Delta(\dot{\boldsymbol{\epsilon}}) = \frac{1}{e} \left[ (e^2 - 1)(\mathrm{tr} \dot{\boldsymbol{\epsilon}})^2 + 2 \mathrm{tr}(\dot{\boldsymbol{\epsilon}}^2) \right]^{1/2}. \tag{6}$$

To avoid numerical singularities at $\dot{\boldsymbol{\epsilon}} = 0$, the values of $\Delta$ are limited from below by the additional parameter $\tilde{\Delta}^* = 10^{-10} \ \mathrm{s}^{-1}$, so that $\Delta^*(\dot{\boldsymbol{\epsilon}}) = \max(\Delta, \tilde{\Delta}^*)$.

The empirical parameters $\tilde{T}_d, P^*, k_T$ and $e$ in equations (1)–(4) define the elastic damping scale of sea ice, its internal pressure, isotropic tensile strength, and the yield curve axes ratio, respectively.

The internal ice pressure $P_p$ is related to ice thickness and concentration in accordance with the rheology of Hibler (1979):

$$P_p = P^* h A \exp[-\tilde{\alpha}(1 - A)] \tag{7}$$

The typical values of the ice strength parameter $P^*$ and $\tilde{\alpha}$ are listed in Table 1.

The bottom and ocean stresses in eq. (1) were parameterized in accordance with Lemieux et al. (2015, 2016) and Hunke and Lipscomb (2008):

$$\boldsymbol{\tau}_b = -C_b \boldsymbol{u} \equiv -\theta(Ah - Ah_b/\tilde{k}_1) \frac{k_2 \boldsymbol{u}}{|\boldsymbol{u}| + \tilde{u}_0} \exp[-\tilde{\alpha}_b(1 - A)] \tag{8}$$

$$\boldsymbol{\tau}_w = -\tilde{C}_w \tilde{\rho}_w A |\boldsymbol{u} - \boldsymbol{u}_w| \boldsymbol{R}_\Theta (\boldsymbol{u} - \boldsymbol{u}_w) \tag{9}$$

where $\theta$ is the Heaviside step function, $h_b$ is the ocean depth, $\boldsymbol{R}_\Theta$ is the $2 \times 2$ matrix rotating the velocity vector by the turning angle $\Theta$ counterclockwise, $\tilde{u}_0 = 10^{-5}$ m/s, and $\boldsymbol{u}_w$ is the water velocity (set to zero in the present study). The values of other parameters ($\tilde{C}_w, \rho_w, \tilde{\alpha}_b, \tilde{k}_1$ and $k_2$) are listed in Table 1.

In contrast to the previous studies, where the free empirical parameters $P^*, e, k_T$ and $k_2$ were assumed to be constant, the present study attempts to retrieve their spatial variability from synthetic (satellite) observations of the sea ice state vector $\boldsymbol{C} \equiv \{h, A, \boldsymbol{u}\}$ using the variational DA technique.

**Table 1.** Model and assimilation system configuration parameters.

| Constant parameters | | |
|---|---|---|
| **Name** | **Symbol** | **Value** |
| Coriolis parameter | $\tilde{f}$ | $10^{-4}$ s$^{-1}$ |
| Ice density | $\tilde{\rho}$ | 900 kg m$^{-3}$ |
| Water density | $\tilde{\rho}_w$ | 1026 kg m$^{-3}$ |
| Water drag coefficient | $\tilde{C}_w$ | 0.0055 |
| Turning angle | $\Theta$ | 0.4363323 rad |
| Advective time step | $\delta t$ | 600 s |
| Subcycling time step | $\delta t_s$ | 1.5 s |
| Elastic damping scale | $\tilde{T}_d$ | $0.72\delta t$ |
| Creep limit | $\tilde{\Delta}^*$ | $10^{-10}$ s$^{-1}$ |
| Compactness strength parameter | $\tilde{\alpha}$ | 20 |
| Compactness basal stress parameter | $\tilde{\alpha}_b$ | 20 |
| Critical thickness parameter | $\tilde{k}_1$ | 8 |
| **Controlled parameter fields** | | |
| **Name** | **Symbol** | **Range** |
| Base Strength Parameter | $P^*$ | $22 - 33$ kN/m$^2$ |
| Yield curve axes ratio | $e$ | $1.0 - 3.1$ |
| Tensile/Compressive Strength ratio | $k_T$ | $0 - 0.8$ |
| Maximum basal stress parameter | $k_2$ | $0 - 20$ N/m$^3$ |

### 2.1.2 Numerical scheme

Numerical formulation of the model closely follows the EVP numerics given in Lemieux et al. (2016).

150  Introducing notations $\sigma_1 = \sigma_{xx} + \sigma_{yy}$, $\sigma_2 = \sigma_{xx} - \sigma_{yy}$, $\sigma_3 = \sigma_{xy}$, $\dot{\epsilon}_1 = \partial_x u + \partial_y v$, $\dot{\epsilon}_2 = \partial_x u - \partial_y v$, $\dot{\epsilon}_3 = \partial_y u + \partial_x v$, bulk viscosity $\zeta = P_p(1 + k_T)/2\Delta^*$, and replacement pressure $P = P_p \Delta/\Delta^*$, equation (2) can be split into three decoupled relationships:

$$\partial_t \sigma_1 + \quad \sigma_1/\tilde{T}_d \quad = \quad 2\zeta \dot{\epsilon}_1/\tilde{T}_d - P(1 - k_T)/\tilde{T}_d \tag{10}$$

$$\partial_t \sigma_2 + e^2 \sigma_2/\tilde{T}_d \quad = \quad 2\zeta \dot{\epsilon}_2/\tilde{T}_d \tag{11}$$

155 $\partial_t \sigma_3 + e^2 \sigma_3/\tilde{T}_d \quad = \quad \zeta \dot{\epsilon}_3/\tilde{T}_d$                     (12)

The first equation is obtained by taking the trace of (2). Subtracting the equation for $\sigma_{yy}$ from the one for $\sigma_{xx}$ in the set of four relationships (2) yields (11), while (12) is just the equation for $\sigma_{xy}$ extracted from the same set. Equations (10-12) are

advanced in time $s$ using the Euler scheme with the subcycling time step $\delta t_s = \varepsilon \tilde{T}_d$ ($\varepsilon = 0.00347$, see Table 1):

$$\sigma_1^s \quad -(1+\quad\varepsilon)^{-1}[\sigma_1 + \varepsilon P_p (1+k_T)\dot{\epsilon}_1/\Delta^* - \varepsilon P(1-k_T)] = 0 \tag{13}$$

$$\sigma_2^s \quad -(1+e^2\varepsilon)^{-1}[\sigma_2 + \varepsilon P_p(1+k_T)\dot{\epsilon}_2/\Delta^*] = 0 \tag{14}$$

$$\sigma_3^s \quad -(1+e^2\varepsilon)^{-1}[\sigma_3 + \;\varepsilon P_p(1+k_T)\dot{\epsilon}_3/2\Delta^*] = 0 \tag{15}$$

Hereinafter, all the fields without superscripts are taken from the previous time steps. After that, velocity components in equation (1) are advanced in time:

$$mu^s/\delta t_s - mfv^s + C_b u^s + \tilde{C}_w \tilde{\rho}_w A|\boldsymbol{u}|(u^s\cos\Theta - v^s\sin\Theta) \quad - \quad mu/\delta t^s - \partial_x\sigma_{xx}^s - \partial_y\sigma_3^s - \tau_{ax} = 0 \tag{16}$$

$$mv^s/\delta t_s + mfu^s + C_b v^s + \tilde{C}_w \tilde{\rho}_w A|\boldsymbol{u}|(u^s\sin\Theta + v^s\cos\Theta) \quad - \quad mv/\delta t^s - \partial_y\sigma_{yy}^s - \partial_x\sigma_3^s - \tau_{ay} = 0 \tag{17}$$

Here $m = \tilde{\rho}hA$ and $\boldsymbol{\tau}_a$ is the atmospheric forcing. After the equations (13)–(17) are advanced in time for 400 time steps, the ice thickness and concentration fields are updated using the obtained velocity $\boldsymbol{u}_n$ and a simplified Lax-Wendroff scheme:

$$h^n \quad - \quad h + \delta t\,[\mathrm{div}(\boldsymbol{u}_n h) + \nu\hat{D}h] = 0 \tag{18}$$

$$A^n \quad - \quad A + \delta t\,[\mathrm{div}(\boldsymbol{u}_n A) + \nu\hat{D}A] = 0 \tag{19}$$

where $\hat{D}$ is the Laplacian operator and $\nu = \delta t \boldsymbol{u}_n^2/2$ is the stabilizing diffusion coefficient. All spatial derivatives present in (13-19) were discretized by finite differences on the Arakawa B-grid. At the rigid boundaries, we used the condition $\boldsymbol{u} = 0$. Initial conditions for $\boldsymbol{u}$ were set either to zero or defined through the model integration for 1 hour. Initial conditions for $A$ and $h$ were specified as arbitrary functions.

The two-stage time stepping EVP procedure, described above, differs from the VP formulation in that iterations of the implicit solver of the VP formulation are replaced by the explicit adjustment of the stress tensor components (eqns. (13)–(15)) in the internal time loop.

## 2.2 Variational DA with EVP sea ice model

### 2.2.1 Strong constraint formulation

In the variational DA experiments we used the so-called strong constraint state-space formulation of the problem, which minimizes a user-defined cost function on the manifold whose structure is specified by the equations of the model. The cost function $\mathcal{J}$ was defined by

$$\mathcal{J} = \frac{1}{2}\sum_{\Omega}\left[W_h(h - h')^2 + W_A(A - A')^2 + W_u(\boldsymbol{u} - \boldsymbol{u}')^2 + \tilde{W}_h(\hat{D}h)^2 + \tilde{W}_A(\hat{D}A)^2 + \tilde{W}_u(\hat{D}\boldsymbol{u})^2\right] \tag{20}$$

Here, $W, \tilde{W}$ denotes the non-zero elements of the user-defined (diagonal) inverse error covariance matrices of the fields in the round brackets, simulated observations are denoted by primes, and summation is made over the entire space-time computational grid $\Omega$. Note, that the first three terms attract the optimized solution to the data, while the last three tend to penalize grid-scale components and enforce smoothness of the optimized fields.

In order to constrain minimization process to the manifold $M$ defined by the model equations (13-19), we introduce notation $X$ for a vector of state variables in all the grid points of $\Omega$ and define the vector of control variables $C = [C_0, C_p]$ which includes the initial state of the model $C_0 \equiv X|_{t=0}$, and other control fields $C_p$ which contain rheological parameters, and atmospheric

forcing errors. Note, that the vector of model trajectory $X$ is a non-linear function of the control vector $C$, whose constituent $C_p$ was defined on a sparser 2d grid (in every 15th node of the computational grid for most of the conducted experiments) and then spatially interpolated on the model grid using the bilinear interpolation operator $\mathcal{I}$.

To constrain the minimization to $M(X, C)$, we introduce the vector of Lagrangian multipliers $\hat{X}$ (adjoints of the state variables $X$) in $\Omega$ (e.g., Le Dimet and Talagrand, 1986) and minimize the modified cost function

$$\mathcal{J}_m = \mathcal{J} + \hat{\mathcal{J}} \equiv \mathcal{J}(X) + M(X, C)^{\mathsf{T}} \hat{X}, \tag{21}$$

with respect to $X, \hat{X}$ and $C$, where $^{\mathsf{T}}$ denotes the transposition. The minimum point is defined by setting the gradients of $\mathcal{J}_m$ (i.e. variations of $\mathcal{J}_m$ that are linear in $\delta X$, $\delta \hat{X}$ and $\delta C$) to zero:

$$\frac{\delta \mathcal{J}_m}{\delta \hat{X}} = M(X, C) \quad = \quad 0 \tag{22}$$

$$\frac{\delta \mathcal{J}_m}{\delta X} = \frac{\delta \mathcal{J}}{\delta X} + \frac{\delta}{\delta X}(M_X \delta X)^{\mathsf{T}} \hat{X} = \frac{\delta \mathcal{J}}{\delta X} + M_X^{\mathsf{T}} \hat{X} \quad = \quad 0 \tag{23}$$

$$\frac{\delta \mathcal{J}_m}{\delta C} = \frac{\delta \hat{\mathcal{J}}}{\delta M} \frac{\delta M}{\delta C} \quad = \quad 0 \tag{24}$$

As it is seen, eq. (22) simply presents the non-linear model trajectory specified by a given set of control variables. The second relationship contains the derivatives of $\mathcal{J}$ and $\hat{\mathcal{J}}$ with respect to $X$ and involves the product of the model operator $M_X$ linearized in the vicinity of $X$ (the tangent space to $M$ at $X$, or the "TL model") by the vector of respective perturbations $\delta X$. As soon as the current iterate of $X$ is determined by running the non-linear model, equation (23) can be solved by running

the adjoint model (transpose of $M_X$) forced by the derivatives of $\mathcal{J}$ to obtain the values of the adjoint variables $\hat{X}$. Finally, the derivatives of $\mathcal{J}_m$ are computed from equation (24) using the chain rule and the values of $X$ and $\hat{X}$ derived from solving eqns. ((22)–(23)).

The minimization of the cost function with respect to $C$ was performed using the limited-memory quasi-Newtonian (M1QN3) method of Gilbert and Le Marechal (1989) with the additional range constraints for the selected control fields (Section 2.3) per-

formed after each iteration. The M1QN3 algorithm employs the above procedure for computation of the cost function gradient $\delta \mathcal{J}/\delta C$ with respect to $C$ for a given value of the control fields.

To sum up, the minimization procedure can be outlined as follows:

1. specify a control vector $C = \{u(0, x), A(0, x), h(0, x), C_p(t, x)\}$ at the $n$th iteration

2. run the forward model (eqns. (13)–(19)) to compute $X_n$, the derivatives $\delta \mathcal{J}/\delta X$ and the value of $\mathcal{J}_n$ (eq. 20)

3. run the adjoint model forced by $-\delta \mathcal{J}/\delta X$ (cf. eq. (23)) to obtain $\hat{X}_n$

4. compute $\mathcal{J}_m/\delta C$ (eq. 24)

5. update the control variables using the M1QN3 software

6. repeat steps (1)-(5) until convergence of the M1QN3 minimization procedure.

Technically, apart from developing the model code ((13)–(19)), the outlined optimization requires development of the routines for computing the gradients as well as the tangent linear model $\boldsymbol{M}_X$ and its adjoint $\boldsymbol{M}_X^\mathsf{T}$. The machinery of deriving these codes is based on the rules of differentiation and was realized in multiple software packages (e.g., Giering and Kaminski, 1998; AutoDiff http://autodiff.com/tamc); OpenAD https://www.mcs.anl.gov/OpenAD; Goldberg and Heimbach, 2013).

More details on the variational techniques of data assimilation in different geophysical applications can be found in numerous publications (e.g. Penenko, 1981; Le Dimet, 1982; Lewis and Derber, 1985; Le Dimet and Talagrand, 1986; Wunsch, 1995; Errico, 1997). Note that both finite-difference TL and adjoint models are completely defined by the finite difference scheme of the forward model, thus allowing application of the above mentioned (semi-)automatic TL/adjoint compilers. An alternative is used in our implementation: the code for $\boldsymbol{M}_X$ is derived by an analytical linearization of the discretized forward model, while the adjoint code (eq. 23) is obtained by analytical differentiation of $\mathcal{J}_m$ with respect to the argument of the TL code. However, numerical stability of the non-linear forward model does not guarantee stability of the respective TL and adjoint models, and requires proper regularization (e.g. Hoteit et al., 2005) to move the eigenvalues of unstable eigenmodes of $\boldsymbol{M}_X$ inside the unit circle. This numerical issue is addressed in the following section.

### 2.2.2 Adjoint and tangent linear models

The TL code was derived by analytic differentiation of the above mentioned numerical scheme in the vicinity of the finite difference model trajectory. The adjoint code was obtained by implicit transposition of the sparse matrix in the code simulating the action of the TL operator $\boldsymbol{M}_X$ on a perturbed state vector $\delta\boldsymbol{X}$. Similar to the TL derivation, this procedure was performed by analytic differentiation with respect to $\delta\boldsymbol{X}$ of the code for computing $\hat{\boldsymbol{X}}^\mathsf{T}\boldsymbol{M}_X\delta\boldsymbol{X}$ (cf. (23). More detailed description of the TLA codes and the gradients with respect to the control variables can be found in Appendix A.

The most laborious part of deriving the TLA codes was associated with linearizing the rhs of eq. (2) with respect to ice velocities and RPs. Note, that the first term in the rhs of the linearized eq. (2) is proportional to the first derivatives of the velocity perturbations $\delta\boldsymbol{u}$. As a consequence, the components of $\boldsymbol{\sigma}$ are linear in the first derivatives of $\delta\boldsymbol{u}$ after taking the explicit time step $\delta t_s$ in the linearized eq. (2). Moreover, the first-order derivatives in $\boldsymbol{u}$ keep their presence in the rhs of the linearized eq. (1) due to spatial variability of the background fields in equations (6) and (7).

This property of the TL equations of the subsystem (1)-(2) may require additional care when specifying the subcycling time step $\delta t_s$ because the gradients of the background fields of $h$ and $A$ may invoke considerably larger propagation speeds of the effective elastic waves than those present in the original non-linear model. Consequently, the TL code could be constrained by a more stringent stability criterion and require even smaller subcycling time steps than those used in the integration of the full non-linear model. In particular, the non-linear stability criterion could be violated, for example, in areas of strong ice convergence. In such regions, eq. (7) implies that large SIT gradients may generate large coefficients of first-order derivatives of $\delta\boldsymbol{u}$ in the TL code for the second term in the rhs of eq. (2).

Preliminary numerical experimentation with the TL code exposed a necessity to reduce $\delta t_s$ as the TL solutions demonstrated uncontrollable amplification of velocity perturbations over the areas of strong sea ice convergence in the background fields. Our attempts to reduce $\delta t_s$ by an order in magnitude reduced this type of amplification with a limited success. A similar instability of the TL EVP solver has been observed in the MITgcm sea ice model (M. Losch, personal communication).

Instability of the linearized codes in the strongly non-linear regimes of the parent model is a well-known phenomenon in 255 the ocean general circulation models (OGCMs). A heuristic solution to this problem was proposed by Hoteit et al. (2004), who added an extra diffusion in TLA codes to suppress unstable small-scale harmonics. This kind of treatment is achieved, however, at the expense of reducing the TLA code accuracy (e.g., Yaremchuk et al., 2009). Later, Yaremchuk and Martin (2014) established a connection between the length of the DA window and the magnitude of the diffusion tensor in the TLA regularization terms.

However, in the sea ice model considered, this type of regularization did not work even when the contribution of the stabilization term was comparable in magnitude to the contributions from other terms in the TLA codes. We attribute this phenomenon to the specific structure of the unstable modes in the TL equations, which often take the form of strongly anisotropic ridge-like structures (i.e. having a wide spatial spectrum of the SIT component) in the areas of ice convergence. As a consequence, the unstable modes cannot be efficiently damped using isotropic diffusion added to the linearized equations for the $\sigma$ and/or ice 265 velocity components located in the respective rows of $\boldsymbol{M_X}$.

A straightforward solution is to introduce a spatially inhomogeneous diffusion tensor field (e.g. Yaremchuk and Nechaev, 2013), with local anisotropy derived from the background solution. However, this requires a considerable reduction of $\delta t_s$ due to very large diffusion along the ridges. As a simple alternative, we employed Newtonian friction in the TL version of eqns. (13)–(15), which homogeneously damps the entire spectrum of small perturbations. With this regularization, additional terms 270 $\varepsilon_N \sigma_i$, $i = 1, 2, 3$ appear inside the square brackets of the linearized equations (13)–(15), where $\varepsilon_N$ is the ratio of $\delta t_s$ to the Newtonian damping time scale $T_N$ (see Appendix A for details). Numerical experimentation has shown that this approach worked generally well using the Newtonian damping time scale $T_N$ of $7\delta t_s$.

Additional experiments have shown that $T_N$ must decrease to $3\delta t_s$ in the case of stronger sea ice convergence in the regions with thick ($h > 3$ m) ice.

Testing the validity of the stabilized TLA codes was done in a way similar to Yaremchuk et al. (2009). The initial conditions for the model thickness and concentration fields $\boldsymbol{C} = \{h(\boldsymbol{x}, 0), A(\boldsymbol{x}, 0)\}$ were slightly perturbed by the realizations of a random function $R$ (viz. $\boldsymbol{C}(\boldsymbol{x}, 0) \rightarrow \boldsymbol{C}(\boldsymbol{x}, 0) + \epsilon R(\boldsymbol{x}) \equiv \boldsymbol{C} + \delta \boldsymbol{C}$), and the model and its TL version were integrated for $t = 5\delta t$. After that, the dependence of the normalized difference between the non-linear solution and its TL approximation was checked by computing the following quantity:

$$280 \quad \Phi(\epsilon) = \frac{|\boldsymbol{X}(\boldsymbol{C} + \delta \boldsymbol{C}) - \boldsymbol{X}(\boldsymbol{C}) - TL(\delta \boldsymbol{C})|}{|\boldsymbol{X}(\boldsymbol{C})|} \tag{25}$$

by the initial conditions listed in the argument, and $|\cdot|$ is the euclidean norm. As it is evident from Fig. 1, the stabilized version of the TL code is characterized by $\Phi(\epsilon) \propto \epsilon$, while the correct TL code should provide the decay proportional to the square of

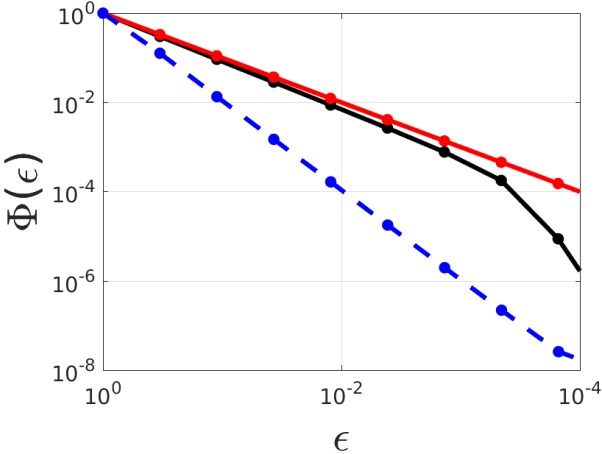

**Figure 1.** TL approximation errors $\Phi(\varepsilon)$ of the EVP (black), regularized EVP (red), and 1d VP ice model (blue dashed) solutions.

$\epsilon$. Decay of $\Phi$ for the unstable TL code is slightly faster than the stabilized one, but the respective solutions produce very noisy patterns causing much earlier stagnation of the descent process as compared to the stabilized code.

It is noteworthy that behavior of $\Phi(\epsilon)$ in similar experiments with the 1d VP sea ice model and corresponding TLA models (Stroh et al., 2019) agreed well with the $\epsilon^2$ dependence of the Taylor expansion (blue line in Fig. 1). We speculate that, similar to the 1d case, stability criteria of the 2d VP system may also be weakly affected by the TL transition due to the specific nature of the linearization in the implicit momentum equation solver (see Appendix B). These considerations may indicate an additional attractive feature of the VP rheology in the practical 4dVar applications. Note also, that to the best of our knowledge, the existing 4dVar sea ice models are based on VP solvers (MITgcm, Heimbach et al., 2010; NAOSIM, Kauker et al., 2009) and have never reported instability of their TLA codes. Our experiments with a larger number of internal iterations (up to 2000) did not reveal any substantial difference in either non-linear model solutions or in stability of the TLA models. More detailed description of the TLA codes are given in Appendix A.

In the OSSEs described below, control variables include initial conditions for sea ice velocity, thickness, and concentration; the wind stress; and the spatially varying RP fields of $P^*$, $e$, $k_T$, and $k_2$. For other RPs, we utilized constant values adopted from Lemieux et al. (2015, 2016) and listed in Table 1.

### 2.3 Simulated observations and cost functions

In all OSSEs we used three types of simulated sea ice observations, trying to keep the magnitude of sea ice observational errors close to realistic values.

The first type of data are accurate SIC observations, of which there are currently multiple gridded products based on various remote sensing instruments with different spatial resolutions. After additional pre-processing, these observations are routinely

used in data assimilation systems (e.g. GOFS 3.1 DA system of Cummings and Smedstad (2013)), with a nominal spatial resolution of 5 km and regionally low SIC representation errors (5%, Yaremchuk et al., 2019).

The second data type are SIT observations which contain moderate errors. Currently, the primary source of such data is CryoSat-2, with 1 km and 5 km gridded 2-day averaged observations available from the Center for Polar Observations and Modeling (*http://www.cpom.ucl.ac.uk/csopr/seaice.html*). Currently, the error estimates of CryoSat-2 SIT observations range between 0.34–0.74 m (Alexandrov et al., 2010; Laxon et al., 2013; Tilling et al., 2018). The recently launched Icesat-2 satellite provides high resolution (40m) freeboard estimates (*https://icesat-2.gsfc.nasa.gov/*) with higher accuracy. So, in the future, novel observational platforms and methods of analysis will likely provide better spatial coverage (i.e. over the entire Arctic) and improved accuracy. In our experiments, we set SIT observation errors to 0.3 m, having in mind future improvements of the SIT data accuracy. A similar error level was adopted by Stroh et al. (2019), which studied RP retrievals in a 1d sea ice model.

Accurate observations of sea ice velocities compose the third data type. An example product is the daily 25 km SIV analysis of various satellite and *in situ* sources (Tschudi et al., 2019). The respective uncertainties were established at 0.01–0.02 m/s (Schwegmann et al., 2011; Sumata et al., 2015). New methods of sequential SAR image comparison can resolve high-resolution SIV with an accuracy of 0.005 m/s (Komarov and Barber, 2014), suggesting a possibility of high-precision SIV observations in the near future. In the OSSEs reported below, inaccuracy of SIV is set to 0.01-0.025 m/s. Simulated SIC, SIV and SIT observations were derived from the "true" solution by adding the above-mentioned errors with a spatial decorrelation scale of 150 km and a temporal decorrelation scale of 7 days. Taking into account that high resolution satellite SIC, SIV are currently available on a daily basis over the entire Arctic Ocean and assuming they can be interpolated within each daily time frame, we set the synthetic observations to be available in all the space-time grid points of the model domain.

The standard state-space 4dVar DA approach of Le Dimet and Talagrand (1986) was utilized: the optimal vector $C$ of control variables was sought to ensure that observations of the model states lie close to assimilated observations within the prescribed time interval (assimilation window) which was set to 3 days in all the experiments. The DA procedure was performed by minimizing the quadratic cost function $\mathcal{J}_m(C)$ which included simulated data and regularization (smoothness) terms both characterized by the diagonal error covariance matrices (Appendix A). In addition, we applied bounding constraints on the field values of ice concentration ($0 \leq A \leq 1$), and the control fields of the rheological parameters (listed in the bottom of the left column in Table 1).

## 2.4 OSSEs

The major goal of the conducted OSSEs was to evaluate the feasibility of reconstructing the RPs through assimilation of the SIV, SIC and SIT observations. The first group OSSEs (named KT and K2) analyze the feasibility of optimizing landfast ice parameters $k_2$ and $k_T$. The control vector in these experiments included only parameters $k_2$ and $k_T$, and first guess initial conditions for SIT and SIC were assumed to be error-free and were not adjusted. Observations of the sea ice velocities, thickness and concentration were assimilated. In the second group of experiments forced by gyre-shaped winds (GYRE-0, GYRE-W), we analyzed the feasibility of optimizing $P^*$ and $e$ in the regions with spatially and temporally varying SIT and SIC. In the third group, we analyzed the feasibility of optimizing $P^*$ and $e$ in the Pack Ice Zone (PIZ) where SIT varies in space and SIC is

**Table 2.** List of the performed experiments

| Experiment | Grid size and resolution $\delta x$ | Description | Objective | Control fields |
|---|---|---|---|---|
| KT | 70×7, 15 km | zonal wind forcing, true $k_T$=0.6 | Evaluate feasibility of optimizing $k_T$ in narrow straits. | $k_T$, $h_0$, $A_0$ |
| K2 | 75×30, 15 km | zonal wind forcing, true $k_2$=15, sloping bottom topography | Evaluate feasibility of optimizing $k_2$ in shallow seas | $k_2$ |
| GYRE-0, GYRE-W | 75×30, 30 km | True solution with cyclonic wind forcing and spatially varying $P^*$, $e$ | Evaluate feasibility of optimizing $P^*$, $e$ in central Arctic under correct and biased wind forcing shallow sea. | $P^*$, $e$, $h_0$ $A_0$, |
| PIZ | 75×30, 30 km | True solution with convergent wind forcing and spatially varying $P^*$, $e$ | Evaluate feasibility of optimizing $P^*$, $e$ in PIZ areas under convergent winds | $P^*$, $e$, $h_0$, $\boldsymbol{u}_0$, |

equal to 1. We also explore the impact of optimizing $P^*$ and $e$ on the hindcast of ice thickness and internal stress distributions. In the second and third groups of the experiments, both the RPs and initial conditions were optimized, and the first guess sea ice initial conditions were, accordingly, disturbed. A list of OSSEs and their short descriptions are assembled in Table 2. The maximum number of control variables associated with the initial conditions (the number of ice model grid points occupied by the SIT, SIC and SIV fields) was about 9000. As mentioned in Section 2.2.1, the RP control fields were defined on coarser ($\delta x_p$=15$\delta x$) grids and bilinearily interpolated on the model grid of the respective OSSEs. Thus, the maximum dimension of the RP control vector never exceeded 36=(75/15+1)(30/15+1) elements, where 75 and 30 are the grid dimensions in Table 1. In all the experiments, we assumed that SIT, SIC and SIV observations were available at all the space-time grid points of the model domain.

## 3   Optimization of the landfast parameters

### 3.1   Arching: optimization of $k_T$

Formation of landfast ice in the deep narrow straits and between islands is a well known phenomenon in the Canadian Archipelagos and in the Kara Sea (e.g. Lemieux et al., 2016). In the Nares Strait, landfast ice is observed periodically and typically its boundary has an arching shape (e.g. Ryan and Münchow, 2017).

To mimic this phenomenon, the sea ice model was configured in a narrow zonal domain forced by steady zonal wind for 3 days (Figure 2a). The initial distributions of SIT/SIC were zonally symmetric and SIT/SIC fields were set  with values of 2 m/1.0 and 0.2 m/0.7 values in the western and eastern parts of the domain, respectively, while initial velocities were set to zero (Figure 2a). Following Konig Beatty and Holland (2010), and Tremblay and Hakakian (2006), the true value of $k_T$ was set to 0.6. Figure 2b shows that after 3 days, the sea ice in the western part of the domain did not drift eastward due to internal tensile strength, which was strong enough to keep sea ice in place, i.e. forming landfast ice in the western part of the domain. In the

eastern part where the tensile strength was weaker due to thinner (0.2 m) ice and smaller SIC (0.7), the sea ice moved eastward with typical velocities of about 0.1 m/s forming a polynya between the landfast ice area and thinner sea ice. Due to the impact of the Coriolis force, ice moved slightly southward forming the polynya along the northern boundary and increasing SIC along the southern boundary.

The first guess solution was forced by the same wind but with $k_T$=0. Figure 2c shows the first guess state at the end of the assimilation window ($t$=3 days). It clearly shows that SI moved eastward with a speed of 0.1–0.15 m/s throughout the entire domain, with the sharp boundary between thick and thin ice (Fig. 2a) deteriorating and the 10km wide polynya developing at the western boundary. Since in this solution ice moves eastward over the entire domain, landfast ice is completely absent due to the absence of tensile strength in the ice ($k_T$=0).

The 4dVar optimization of only $k_T$ (initial distributions of SIC and SIT were not optimized) provides a significant improvement of the SIC and SIT (latter not shown) clearly seen in Figure 2d. The optimized $k_T$ (Figure 2f) is very close to the true (0.6) value almost everywhere in the western part of the domain, while in the eastern part, it is close to zero. Thus, the optimization of $k_T$ enabled formation of landfast ice in the western part of the model domain. Obviously, a similar effect could be achieved with much higher values of $k_T$. To remove this ambiguity, we added an additional term into the cost function which
is proportional to the integral of $k_T^2$ over the domain. By minimizing the magnitude of $k_T$, we find the minimum value of $k_T$ necessary for holding ice in place. Note, that optimization of $k_T$ was achieved in only four iterations (Figure 2e), which is a consequence of our sparse grid representation of RPs in the 4dVar experiments. If the initial SIT and SIC distributions are not error-free, it is also possible to optimize the initial SIT and SIC in addition to $k_T$. This kind of optimization provides better SIT/SIC hindcast but requires more iterations to find the optimal solution (dashed line in Figure 2e).

Lemeaux et al. (2016) conducted multiple experiments with different values of the $k_T$ specified over the entire Arctic Ocean and found that $k_T$ should be smaller than 0.6, the value originally proposed by Konig Beatty and Holland (2010). Because of this, we conducted an additional experiment in which the value of $k_T$ was set to 0.2 everywhere and a weaker wind of 6 m/sec was specified. Sea ice thickness and concentration were the same.

The optimized solution after 3 days (Figure 2g) is very similar to the solution in the experiment with $k_T = 0.6$ but velocities
are smaller and sea ice concentration in the polynyas are higher due to weaker wind forcing. However, the spatial pattern of the optimized $k_T$ distribution is different: it has a clear meridianal maxima of 0.18-0.21 between 450 and 600km and sharply decreases to nearly zero in the other parts of the domain. The largest maximum of the optimized $k_T$ values is very close to the true value of $k_T$=0.2 utilized for this experiment. Note, that ice velocity in the entire western part is still equal to zero because the optimized tensile strength is sufficiently strong to keep all ice in place, while wind is not strong enough to deform the sea ice
in the western part of the domain. This result suggests that accurate land fast ice modeling can be achieved by specifying non-zero $k_T$ only in the key regions and thus, there is no need to specify uniform $k_T$ as it was done in the experiments conducted by Lemieux et al. (2016). In operational practice, such arching regions can be identified by analyzing SAR and/or SST images (e.g. Ryan and Münchow, 2017), or from historical sea ice maps available from sea ice data centers.

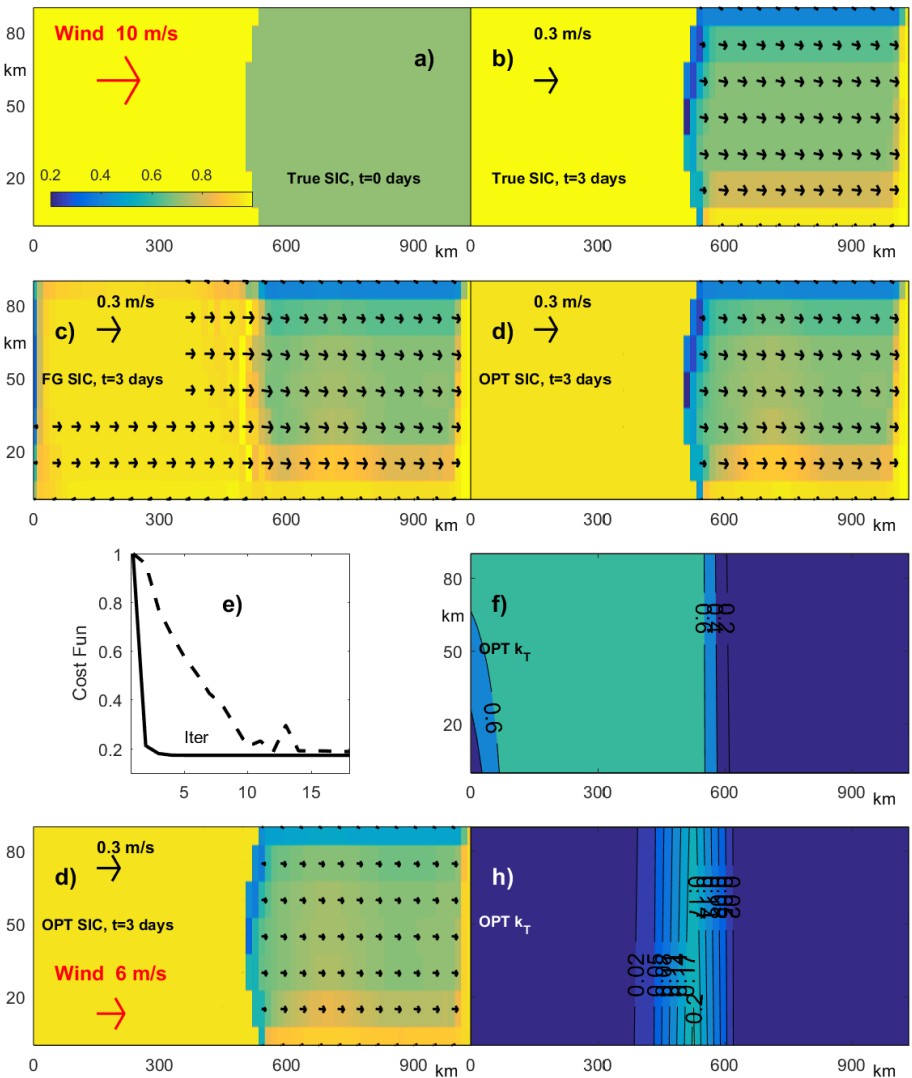

**Figure 2.** Results of the OSSEs optimizing $k_T$: *Top panels:* True SIC and SIV with $k_T$=0.6 at $t$=0 (a) and $t$=3 days (b). *Upper middle panels:* The first guess (c) and optimized (d) SIC and SIV fields at $t$=3 days; *Lower middle panels:* evolution of the normalized cost function for the OSSE with optimization of $k_T$ only (solid line) and with the joint optimization of $k_T$, $h$, and $A$ (dashed line) (e). Left panel (f) shows optimized $k_T$ for the experiment with true first guess SIC/SIT distributions at $t = 0$. *Bottom panels:* Results of the OSSEs optimizing $k_T$ with true $k_T$=0.2 and zonal wind of 6 m/sec: optimized velocity field and concentration at $t$=3 days (g); optimized $k_T$ (h).

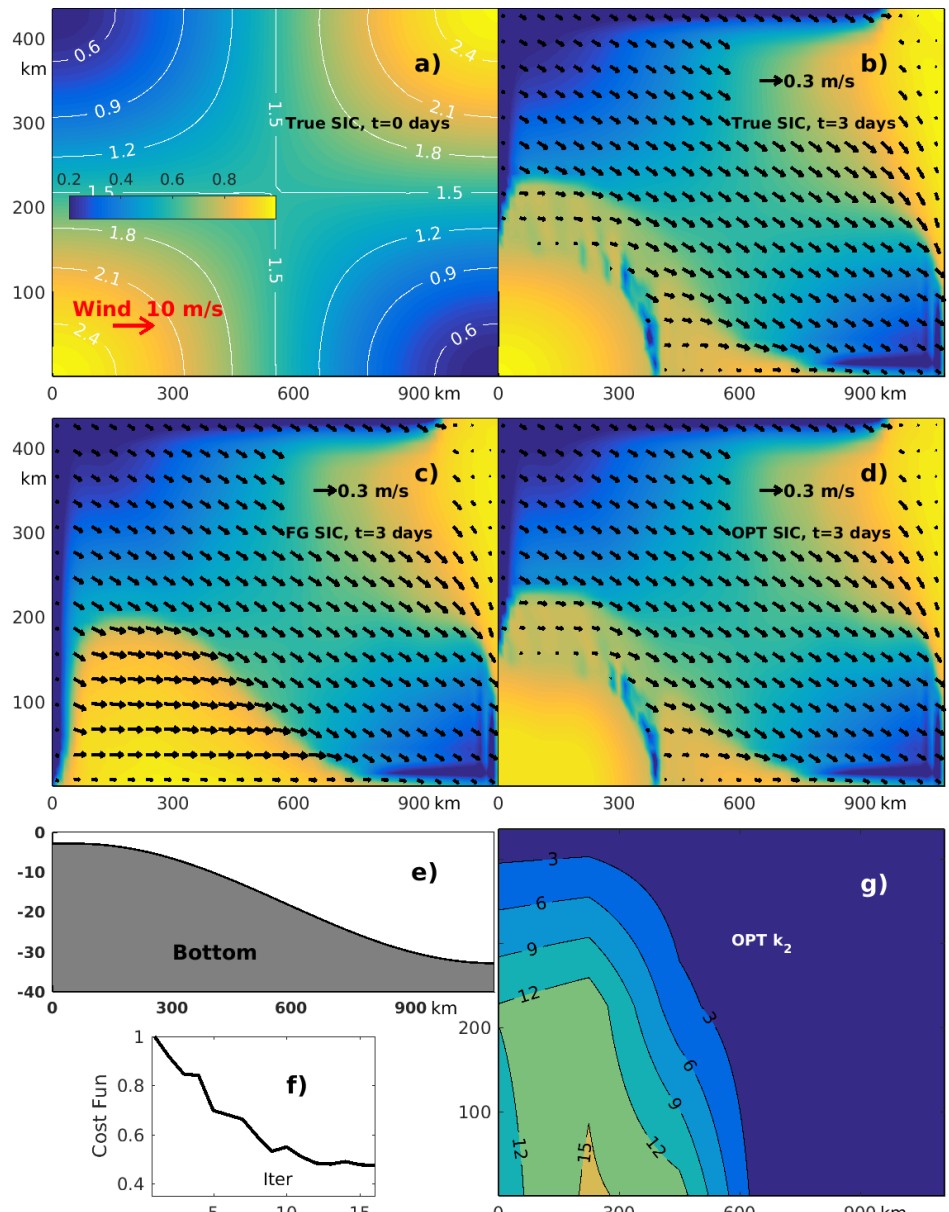

**Figure 3.** Results of the $k_2$ optimization with $\tilde{k}_1 = 0.8$: *Upper panels (a-:* True SIC and SIV with $k_2$=15 at $t$=0 and $t$=3 days respectively. SIT distribution (meters) is shown by white contours in the left panel; *Middle panels:* The first guess SIC and SIV with $k_2$=0 at $t$=3 days (c) and (d) optimized SIC and SIV at $t$=3 days. *Bottom panels:* zonal topography profile (e), evolution of the normalized cost function for the OSSE with optimization $k_2$ (f), and the optimized $k_2$ distribution (g).

## 3.2 Grounding effect: optimization of $k_2$

Grounding on the shallows is another mechanism of landfast ice formation. This kind of landfast ice is typically observed in the Laptev, Chukchi and East Siberian Seas and along the northern Alaskan coast (e.g. Lemieux et al., 2015). To mimic this phenomenon, the model was configured in the rectangular $1125 \times 450$ km domain (Figure 3) with zonally varying depth ranging between 3 m at the western boundary and 33 m at the eastern boundary (Figure 3e). The model was forced by the uniform zonal 10 m/s wind. The true solution was specified as follows. The initial SIC ranged between 0.2 and 1 while the initial SIT was proportional to the SIC and ranged between 0.25 m and 2.5 m as shown in Figure 3a. Initial velocities and the tensile/compressive strength ratio $k_T$ were set to zero, while the following true values of the RPs (Lemieux et al., 2016) were used: the critical thickness parameter $\tilde{k}_1$=8, basal stress parameter $\tilde{\alpha}_b$=20, and the maximum basal stress parameter $k_2$=15 N/m$^3$. Figure 3b shows that after 3 days, the sea ice moved eastward in most of the domain with a typical speed of 0.2–0.4 m/s; only in the south-western corner was the combination of SIT, SIC, and bottom topography sufficient to keep sea ice in place, thus forming a region of grounded landfast ice. Another interesting feature is the elongated polynya visible along the eastern boundary of the landfast ice region approximately 400km from the western coastline (Figure 3b). The first guess solution had the same initial conditions, wind forcing and RP distribution with the exception that $k_2$ was set to zero.

Initial SIT and SIC fields were set to the true values. Despite the perfect initial condition, it is clearly seen that wind moves sea ice eastward with a speed of 0.3–0.4 m/s, and forms a polynya along the western boundary (Figure 3c) which does not exist in the true solution (Figure 3b). The landfast ice region (i.e. the area with no SIV) in the southwestern corner is completely absent. This polynya separating the landfast ice from the moving ice in the south (bottom of Figure 3b) is also absent in the first guess solution.

The variational assimilation of SIV, SIT and SIC observations, targeted at optimization of $k_2$, demonstrated a significant improvement of SIT (not shown) and SIC, clearly seen in Figure 3d. In particular, the optimized solution includes a landfast ice region and a polynya which are nearly identical to those in the true solution (Figure 3b).

The optimized field of $k_2$ is shown in Figure 3g. The maximum values of $k_2$ are very close to the true $k_2$=15 N/m$^3$. Note that the true $k_2$ value was specified ad hoc and the grounding effect formally could be reached with smaller values of $k_2$. This is clearly demonstrated in Figure 3g, where $k_2$ is about 12 N/m$^3$ over the major part of the landfast ice area in true solution (cf. Figure 3b,g).

Similar to the KT experiment, an additional term penalizing the magnitude of $k_2$ was added to the cost function, and the optimized field of $k_2$ was obtained after a relatively small number of iterations (10–12) (Figure 3f). Note also, that according to equations (1) and (8), sea ice acceleration is directly proportional to $k_2$ which should invoke faster convergence of the minimization procedure.

To demonstrate the robustness of the $k_2$ optimization with respect to possible variations in $\tilde{k}_1$, we conducted a similar experiment with a smaller true $\tilde{k}_1$=2.5. As seen in Figure 4a,b, the decrease of $\tilde{k}_1$ causes a considerable decrease of the landfast area in the southwestern corner. The landfast ice polynya has also moved approximately 100 km closer to the western coastline and is now confined to the shallower region. The first guess solution for this experiment was the same as in Figure 3c with

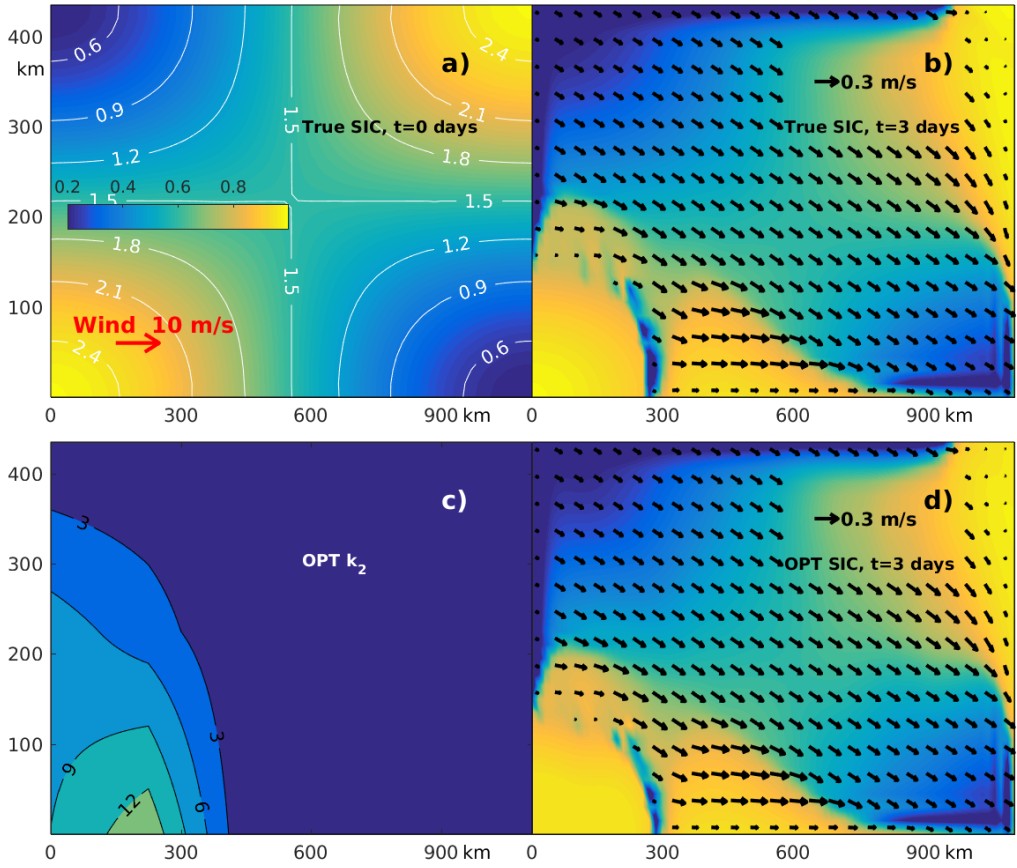

**Figure 4.** Results of $k_2$ optimization with $\tilde{k}_1 = 2.5$: Upper panels: True SIC (color) and SIV distributions with $k_2 = 15$ at $t=0$ (a) and $t=3$ days (b). SIT distribution (meters) is shown by white contours in the left panel. Bottom panels: optimized $k_2$ distribution (c) and optimized SIC and SIV at $t=3$ days (d).

$k_2$=0, i.e. all ice moving eastward. However, after 4dVar DA and optimization of $k_2$, the reconstructed solution (Figure 4d) is nearly identical to the true solution (Figure 4b). This optimized $k_2$ (Figure 4c) has the same spatial structure but a slightly smaller ($\sim 14\ \text{N/m}^2$) maximum value compared to the experiment where $\tilde{k}_1$=8.

The parameterization of grounding landfast ice also includes the critical thickness parameter $\tilde{k}_1$, which was kept fixed in the described experiments. According to multiple numerical simulations, the total landfast ice area is more sensitive to variations of $\tilde{k}_1$ than $k_2$ (Lemieux et al., 2015) because $\tilde{k}_1$ can be interpreted as a scaling coefficient in the definition of the critical ice thickness $\tilde{k}_1 h_c = A h_b$ (cf. eq. 8). It, therefore, acts as a "switch" which defines the areas of potential landfast ice generation. However, the discontinuity of the Heaviside step function present in equation (8) significantly complicates $k_1$ optimization through the gradient-based variational method. Formally, this problem can be regularized (e.g., Nicolsky et al., 2009), but such an approach requires the additional optimization of a regularization parameter, which may ultimately

be less computationally efficient in practice. In light of this consideration, we limit our feasibility analysis of landfast ice parameterizations to optimizing $k_T$ and $k_2$.

## 4  Optimization of the ice strength and axes ratio fields

### 4.1  Cyclonic gyre experiments (GYRE-0/W)

The rheological parameters $P^*$ and $e$ are the most important set of parameters responsible for proper sea ice modeling in the deep part of the Arctic Ocean. To evaluate the feasibility of optimizing $P^*$ and $e$, the EVP model was configured for a 2250 x 900 km rectangular domain with initial true values of the SIC and SIT fields as shown in Figure 5a. The true values of $P^*$ and $e$ varied as shown in Figure 5e,f within the following ranges: 22.5 kN/m$^2$ $\leq P^* \leq$ 32.5 kN/m$^2$ and $1 \leq e \leq 3$. These ranges were adopted from various studies (e.g. Hibler and Walsh, 1982; Kreyscher et al., 1997, 2000; Tremblay and Hakakian, 2006; Lemieux et al., 2016). The true wind forcing had a form of a Gaussian-shaped cyclone with a stationary position whose strength gradually increased by 1.5 times during the 3-day assimilation window. The resulting wind stress at $t$=0 is shown in Figure 5c and had a maximum value of 0.7 N/m$^2$. Initial SIV conditions were determined by a 100-minute model integration starting from rest, with the all other initial variables and parameters being the same. The initial internal stress was small, but significantly increased after 3 days, under the applied atmospheric forcing. Figures 5c,d show the trace of the stress tensor $\sigma_I \equiv P_{tr} = -\mathrm{tr}\boldsymbol{\sigma}/2$ at the beginning of the true state integration and after 3 days. The $P_{tr}$ distribution has a clear maximum near the location with the coordinates (500 km, 500 km), which corresponds to the maximum pressure ($\sim$40 kN/m$^2$) in the sea ice field (e.g. Tremblay and Mysak, 1997). This maximum is due to strong convergence of the relatively thick ($\sim$ 2.5m) sea ice in this region. In the eastern part of the domain, $P_{tr}$ is typically very low due to the divergence of the SIV and considerably thinner ($\sim$ 0.5-1.5m) sea ice.

Noisy SIC, SIT and SIV observations were generated by adding spatially and temporally correlated noise (with the correlation scales of 150 km and 7 days) to each of the state variables of the true solution at every time step. The simulated data mimics realistic observations such as those obtained from sources discussed in Section 2.3, i.e. they have similar absolute errors as most of the currently available observations. In the experiments, we did not introduce any bias to ice observations since the bias free observations are a common assumption in existing DA systems.

The magnitudes of the imposed noise correspond to errors of the respective observational data sets, with the amplitudes of 0.05, 0.25–0.35 m, 0.025 m/s, and 0.01–0.025 N/m$^2$ for SIC, SIT, SIV and wind stress, respectively. The initial conditions for the first guess solution were generated in a way similar to the true solution, with slightly larger decorrelation length scales for SIT, SIC and SIV and spatially uniform values of $P^*$=27.5 kN/m$^2$, $e$=2 and true wind forcing.

Despite the exact wind forcing, the first guess solution differs significantly from the true solution. Similarly to Stroh et al. (2019), the optimization was conducted in three steps. First, we optimized initial SIV, SIT and SIC conditions $\boldsymbol{C}_0 = \{\boldsymbol{u}_0, h_0, A_0\}$. Then we sequentially optimized rheological components of the control vector $\boldsymbol{C}_{rh} = \{P^*, e\}$ and finally conducted an additional optimization of the full control vector $\boldsymbol{C} = \{\boldsymbol{C}_0, \boldsymbol{C}_{rh}\}$. Note that available SIV, SIT and SIC observations efficiently constrain the respective initial conditions and thus provide a more accurate first guess for the final optimization of

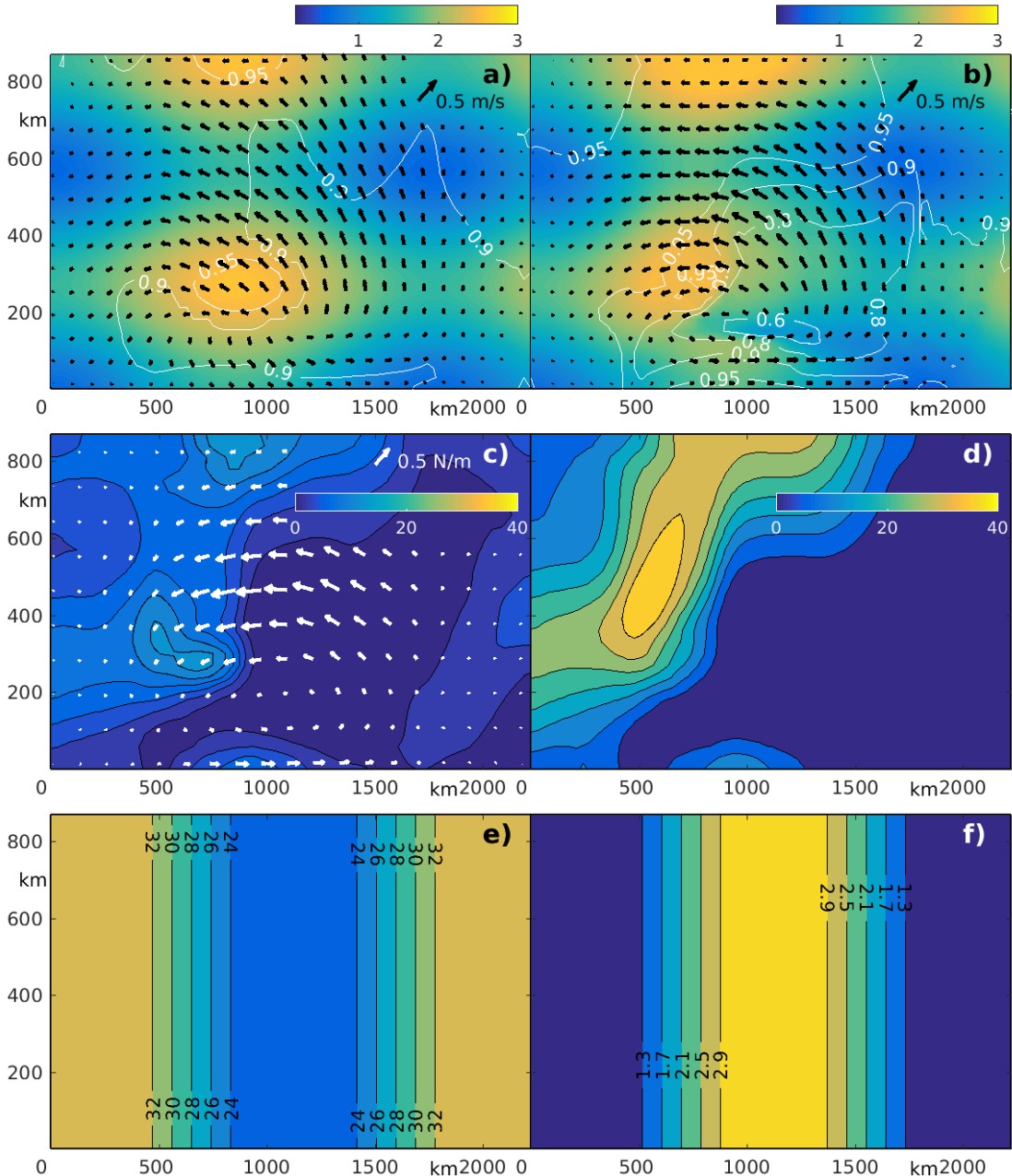

**Figure 5.** True solution in GYRE-0/W-OSSEs: (a-b) evolution of the SIT, SIC (white contours) and SIV (black arrows) at $t$=0 (left panel) and 3 days; (c-d) evolution of the trace $P_{tr}$ of the internal stress tensor at $t$=0 (left). White arrows show the initial wind stress caused by specified cyclonic wind forcing; (e-f) true distributions of $P^*$ (kN/m $^2$, left panel) and $e$.

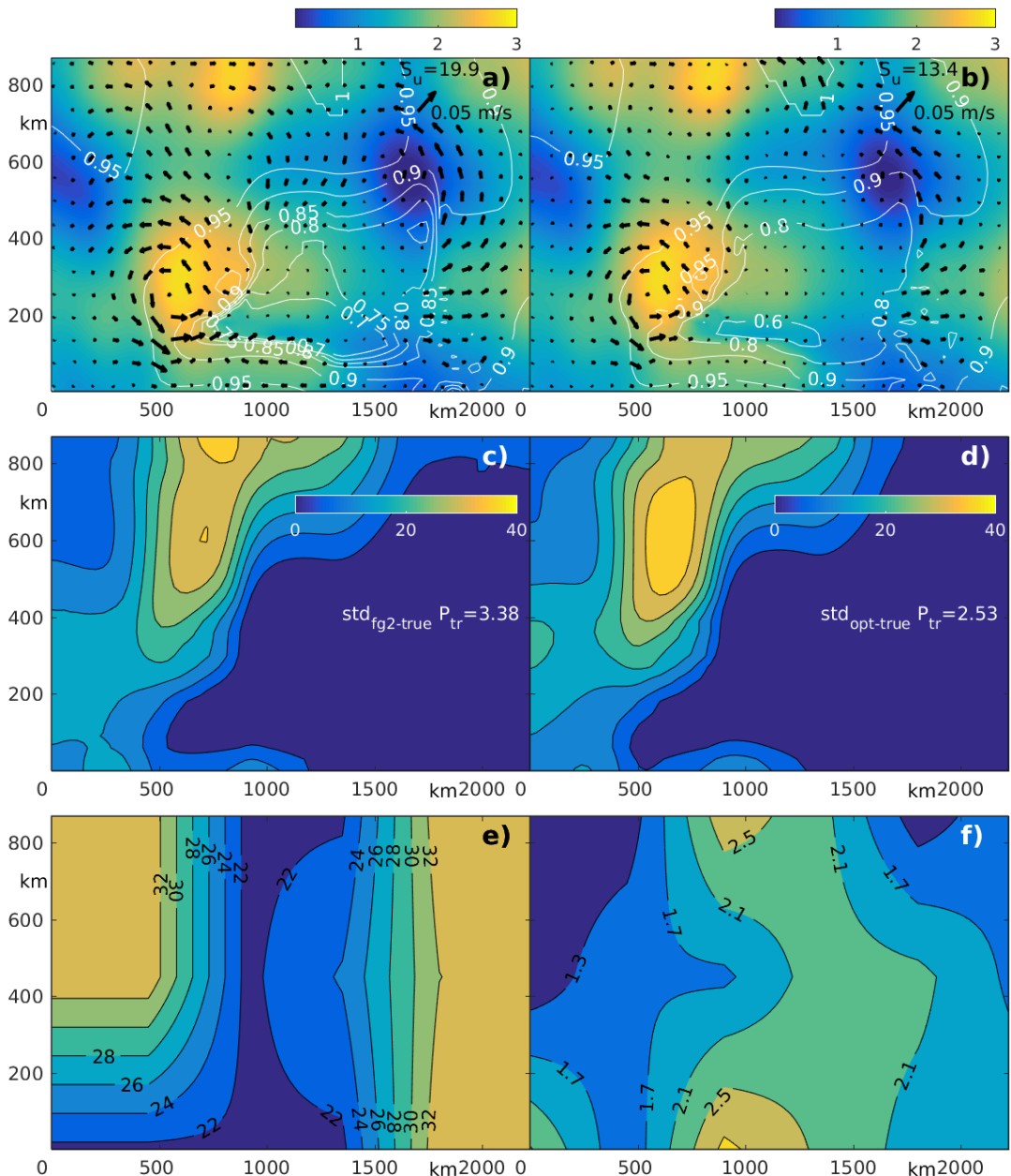

**Figure 6.** Results of the GYRE-0-OSSE: sub-optimal distribution of SIT, SIC (a) and $P_{tr}$ (c) for $t$=3 days after optimization of the initial conditions $\boldsymbol{u}_0$, $h_0$, $A_0$ using the first guess values of $P^*$=27 kN/m$^2$ and $e$=2. The deviation norms $S_{\boldsymbol{u}}$ and $S_{tr}$ are shown. (b,d) is the same as (a,c) but after additional optimization of $P^*$ and $e$; (e,f) - optimized distributions of $P^*$ (kN/m$^2$) and $e$. Black arrows in (a,b) show the difference between optimized and true SIV.

the entire control vector. This is important for highly non-linear optimization problems whose cost functions may have multiple minima.

Figure 6a-d compares the model states after optimization of the initial conditions $C_0$ with fixed $P^*$ and $e$ (left panels) and after additional spatial optimization of the $P^*$ and $e$ (right panels). The major result of optimizing $C_{rh}$ is an improvement of the SIV fields. The formal quantitative measure of the SIV improvement was evaluated by the function

$$S_{\boldsymbol{u}} = \sum_{\Omega,k} |\boldsymbol{u}_{opt}^k - \boldsymbol{u}_{true}|$$

where $\Omega$ is the model domain and index $k$=1,2 enumerates sub-optimal optimization stages: $k = 1$ using the initial conditions control vector $C_0$ only, and $k = 2$ employing the full control vector $C = \{C_0, C_{rh}\}$. It was found that $S_{\boldsymbol{u}}$ reduced almost 1.5 times after the additional optimization of the rheological parameters $C_{rh}$ (Figure 6a,b). Visual comparison of the sub-optimal and fully optimized SIC shows a certain improvement after $C_{rh}$, as well. For example, the local minimum of the sub-optimal SIC in the region with coordinates 700-1500 km and 180 km is about 0.7 (white contours in Figure 6a), while the fully optimized SIC has a minimum of 0.6 and agrees perfectly with the true SIC distribution (white contours in Figure 5b).

In this experiment, we found that optimization of $C_{rh}$ yields only a marginal improvement of the SIT distribution. In particular, $std(h_{opt}^k - h_{true})$ decreased from 0.23 m, after optimization of the initial conditions, to 0.2 m, after additional $C_{rh}$ optimization. The minor impact of $C_{rh}$ optimization on the SIT is probably due to relatively high SIT errors and a substantial difference between the first guess and observed SITs. Another possible reason is that the initial SIT distribution is not well controlled by $C_{rh}$ over the relatively short time scale of the 3-day assimilation window.

As expected, optimization of the rheological parameters $C_{rh}$ provides a major impact on the correction to the internal stress tensor. Figures 6c,d show that standard deviation of the differences between the true and the optimized values of $P_{tr}$ decreased by $\sim 35\%$, after the additional RP optimization. Note, also, that the fully optimized $P_{tr}$ maximum (Figure 6d) is located in close proximity to the true $P_{tr}$ maximum (Figure 5d), while the maximum in the sub-optimal $P_{tr}$ is shifted almost 400 km northward. Comparing optimized and true $P^*$ fields demonstrates agreement almost everywhere, with the only exception observed in the southwest corner of the domain (Figure 6e). This is caused by the substantial SIV divergence (Figure 6a) which diminishes the role of rheological forcing in the region.

The reconstruction of $e$ is not as accurate as $P^*$; the optimized $e$ ranges between 1.2 and 2.8, while the respective true values are between 1 and 3. However, the spatial locations of the extrema are in strong agreement. The exception is the local minimum in the south-western corner, where both $e$ and $P^*$ disagree with true values and are close to their first guess values. Note, that significant improvement of $P_{tr}$ and the internal stress components discussed above are directly related to the more accurate optimization of the RP and SIV fields because both $P^*$ and $e$ control the structure of the stress tensor in eq. (2).

To analyze the impact of wind forcing inaccuracy, we conducted an additional experiment where the center of the cyclonic disturbance was displaced 90 km westward, mimicking a systematic error in the hypothetical atmospheric forecast. The results obtained after full optimization of the control vector $C = \{C_0, C_{rh}\}$ are shown in Figure 7. It is worthy to note that the inaccurate position of the cyclone causes significant errors (up to 0.2 N/m$^2$, or $\sim$25%), in the wind stress forcing, in the

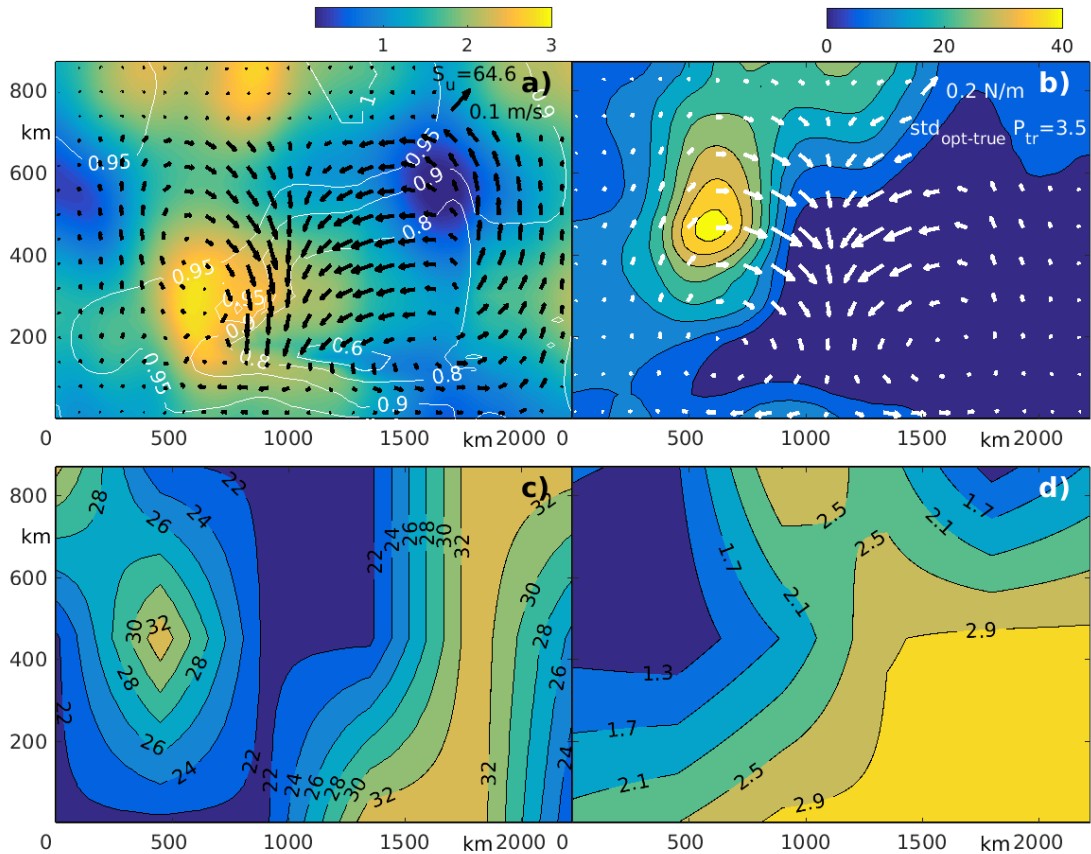

**Figure 7.** Results of the GYRE-W-OSSE: optimized distribution of SIT, SIC (a) and $P_{tr}$ (b) at $t=3$ days. Black/white arrows show the differences between true and optimized SIV (a) and wind stress (b) respectively; (c,d) – optimized distributions of $P^*$ (kN/m$^2$, left) and $e$.

central part of the domain (Figure 7b). As a result, the optimized SIV fields have essential ($\sim$ 0.1 m/s) errors (Figure 7a) and the integral measure of the SIV inaccuracy $S_u$ increased five times up to 0.64 m/s.

At the same time, degradation of the SIT retrieval was not as significant, with $std(h_{opt}^k - h_{true})$ increasing up to 0.25 m, i.e. by only 25% as compared to the previous experiment with exact wind forcing. Similarly, the optimized SIC distribution remained largely unchanged. The integral quality of the reconstruction of $P_{tr}$ is 3.5kN, i.e. about 40–50% worse than in the experiment with exact forcing, but it is important to note that maximum of $P_{tr}$ is remains in very good agreement with the true solution.

Although inaccurate wind forcing has a profound impact on the accuracy of $P^*$ and $e$ retrievals, there is still an essential level of similarity between the reconstructed and true rheological fields. For example, spatial distribution of the optimized $P^*$ still has its maxima in the western and eastern parts of the region and a minimum in the center of the domain (Figure 7c), while

the minimum of the $e$, in the western part, demonstrates a certain agreement with true $e$ distribution (Figure 5e,f). Note, that inaccurate wind forcing affects the accuracy of $P^*$ retrievals to a lesser degree than that of $e$.

## 4.2 PIZ-OSSE

In both experiments, described in the previous section, the initial true SIC was rather close to 1 and decreased below 0.8 in some regions after 3 days of integration (Figure 5a,b). Due to the exponential dependence of $P$ on $1 - A$ (eq. 7), the internal stress decreases $\exp(4) \approx 50$ times, and therefore has a minor rheological impact on the sea ice dynamics in these regions. At the same time in winter, most of the Arctic Ocean is almost completely covered by pack sea ice with SIC ranging between 0.98 and 1. To mimic these conditions, we conducted another OSSE with spatially and temporally invariant sea ice concentration $A = 1$. Numerically, this was achieved by removing the advection equation from eq. (4), and removing initial $A_0$ from the control vector $\boldsymbol{C}_0$. Note, that setting $A = 1$ can be interpreted as the introduction of a very fast cooling, which immediately removes areas with $A < 1$.

The model domain, initial SIT, and $P^*$ and $e$ distributions were the same as in GYRE-0/W-OSSEs. However, unlike the cyclonic wind from the previous experiment, the applied atmospheric forcing is a 16 m/s eastward wind at the western boundary which reverses in zonal direction across the breadth of the domain (Figure 8c). In time, the wind speed was linearly increased up to 20 m/s by the end of the DA window. The resulting wind stress at $t$=3 days (Figure 7c) has a maximum amplitude of about 0.5 N/m$^2$. This relatively strong wind corresponds to the category of strong Arctic Cyclones, a rather typical phenomenon in the Arctic Ocean, which may persist for periods of up to two weeks (Simmonds and Rudneva, 2012).

The temporal evolution of the true SIT and SIV is shown at Figure 8a,b. Under the relatively strong applied wind forcing, sea ice converges and SIT increases almost everywhere, with the exception of the narrow bands along the western and northern boundaries, caused by the joint effect of the Coriolis force and coastal boundary conditions. The true SIV has a maximum of about 0.4 m/s. The distribution of $P_{tr}$, shown in the lower panels of Fig. 8, has two clear maxima in the south, with the magnitudes of about 70 and 50 kN/m$^2$. Note, that due to ice convergence causing SIT growth, both $P_{tr}$ maxima increase in magnitude and slightly ($\sim 60$ km) move towards each other.

The first guess initial SIT/SIV conditions and data for the PIZ OSSE were derived from the true solution in a similar way as for the GYRE-0 OSSE. Similarly, we specify the first guess with $P^*$=27.5 kN/m$^2$, $e$=2, and the exact wind forcing. The optimization was conducted in three steps, by first optimizing $\boldsymbol{C}_0$, then $\boldsymbol{C}_{rh}$, and, finally, simultaneous optimization of $\boldsymbol{C}_0$ and $\boldsymbol{C}_{rh}$.

Figure 9a,c shows SIT and the difference between optimized and true SIV at $t$=3d after optimization of the initial conditions only. Interestingly, despite less spatial variations in wind forcing, optimization of the initial conditions does not allow for accurate reconstruction of SIV as in the GYRE-0/W OSSEs. The maximum errors in the eastern part of the domain are about 0.1 m/s, being comparable in magnitude with regional velocities. The relative accuracy of the SIV reconstruction, in the western part, is slightly better, but is still considerably worse than in the previous OSSEs.

The sub-optimal distribution of $P_{tr}$ (Figure 9c) differs significantly from the true $P_{tr}$ (Figure 8d), both quantitatively and qualitatively. For example, $std(P_{tr}^1(\text{opt})\text{-}P_{tr}(\text{true}))$ is 11.6 kN/m$^2$, or about 30-35% of the domain-average value of $P_{tr}$. The

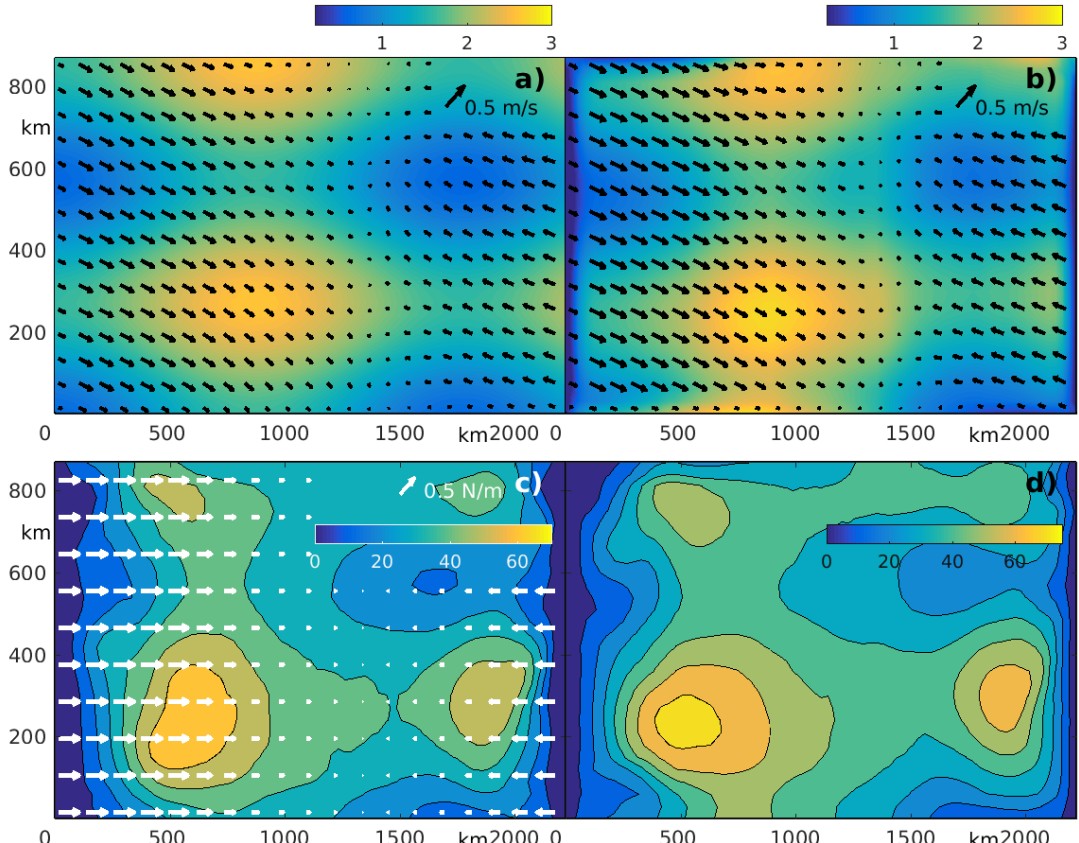

**Figure 8.** True solution in PIZ-OSSEs: a-b) Evolution of the SIT and SIV (black arrows) at $t$=0 (left) and 3 days; c-d) evolution of the $P_{tr}$ of the internal stress tensor. White arrows show the wind stress forcing at $t = 0$; The distributions of $P^*$ (kN/m$^2$) and $e$ are the same as for GYRE-0/W experiments (Fig. 5e,f).

qualitative difference is probably more important because the sub-optimal $P_{tr}$ distribution fails to provide the two maxima discussed above. Instead, the sub-optimal distribution of $P_{tr}$ has only one maximum located in the center (Figure 8d).

Additional optimization of the rheological parameters, $\mathbf{C}_{rh}$, significantly improves the reconstructed SIV practically everywhere, with the formal measure of the uncertainty $S_u$ decreasing by almost one-half from 56 m/s to 30 m/s (Figure 9b). Similar improvements are visible in the $P_{tr}$ distribution (Figure 9d). The $std[P_{tr}(opt) - P_{tr}(true)]$ decreased to 7.4 kN/m$^2$ (by 40%) and the fully optimized $P_{tr}$ has two maxima as in the true $P_{tr}$ distribution (Figure 8d).

The optimized $P^*$ and $e$ are shown in Figure 9e,f. In the eastern part of the model domain, the reconstructed $P^*$ and $e$ almost perfectly agree with the true distributions of $P^*$ and $e$, while there is some difference between optimized and true rheology in the western part. This is probably due to offshore sea ice transport which creates ice divergence along the rigid western boundary. As a reasult, the impact of rheology on ice dynamics becomes less significant here and rheological parameters are

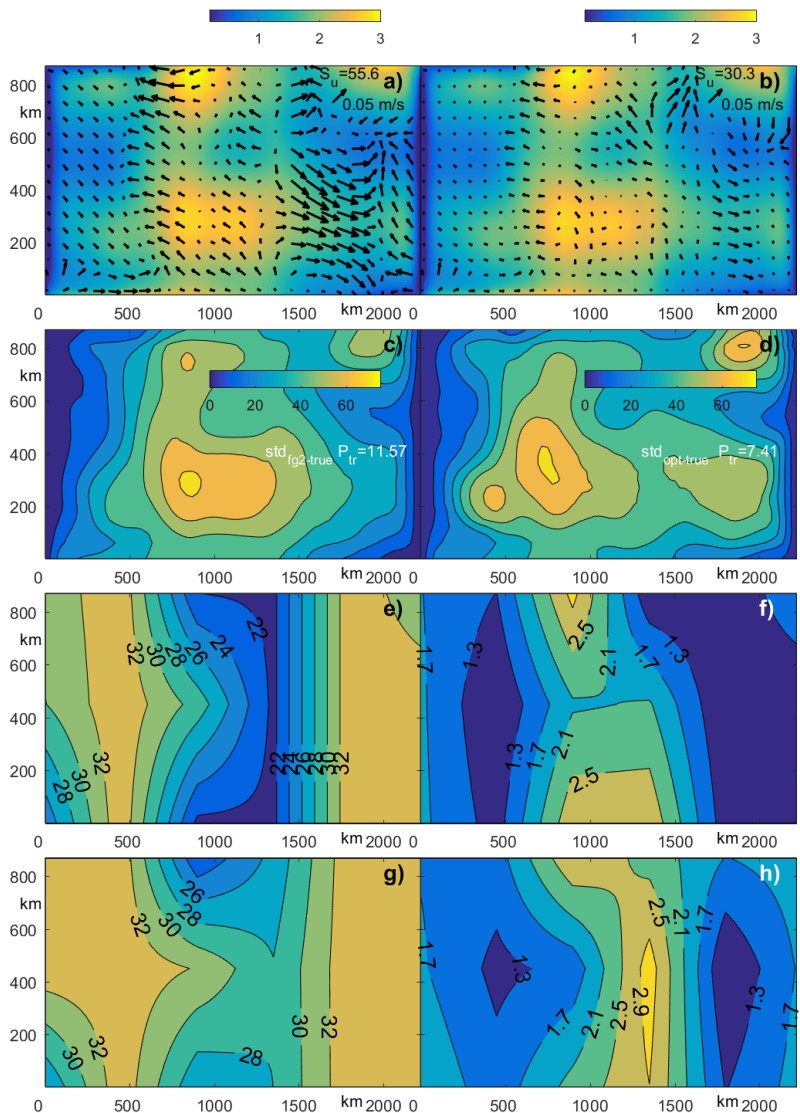

**Figure 9.** Results of the PIZ-OSSE: sub-optimal distributions of SIT (a) and $P_{tr}$ (c) at $t=3$ d after optimizing the initial conditions $\boldsymbol{u}_0$, $h_0$, $A_0$ and the first guess values of $P^*$=27.5 kN/m$^2$ and $e$=2. The discrepancies of $S_{\boldsymbol{u}}$ and $S_{tr}$, with the true solutions are shown. Black arrows show the difference between optimized and true SIV for each optimization stage. Panels b and d: same as a and c, but after additional optimization of $P^*$, $e$; Panels e and f show optimized distribution of $P^*$ (kN/m$^2$) and $e$. Bottom panel g and f show optimized distribution of $P^*$ (kN/m$^2$) and $e$ for the experiment with moderate wind of 10 m/sec.

harder to recover. There is also some quantitative difference between optimized and true $e$ in the central part of the model domain but, qualitatively, the reconstructed field of $e$ has all the features of the true distribution.

As mentioned above, PIZ experiments were conducted under a relatively high wind forcing. To assess the possibility of reconstructing $P^*$ and $e$ under weaker winds, we conducted an additional experiment with a typical wind of about 10 m/sec and a maximum wind stress about 0.15 N/m$^2$. The results, shown in Figures 9g,h, are similar to those of the experiments using stronger wind. In particular, both $P^*$ and $e$ are reconstructed better in the eastern part of the domain and somewhat worse in the western part due to ice divergence near the western boundary. Compared to the stronger wind case, the quality of $P^*$ reconstruction is diminished under weaker winds because lower wind stress generates smaller ice velocities, amplifying the impact of observation errors.

Note, however, that most of the inaccuracies in the reconstructed $P^*$ distribution are observed in the region with relatively thick (2.0–2.6 m) ice located meridionally between 700 km and 1200 km in the meridional direction. Under moderately weak wind, therefore, it is still possible to accurately reconstruct RPs in thin ice regions. Obviously, RPs are not reconstructable in regions where winds are too weak to influence ice motion. In this case, optimized $P^*$ may freely take any value above a critical threshold determined by ice strength.

## 5    Summary and discussion

The presented study continues our previous efforts (Stroh et al., 2019) and addresses the feasibility of retrieving spatially-varying RPs through the 4dVar assimilation the satellite observations of SIV, SIC and SIT in two dimensions. To perform this analysis, we developed TLA codes with respect to all rheological parameters (except $k_1$), initial conditions, and wind forcing for a single-category sea ice model recently proposed by Lemieux et al. (2016). The dynamical core of this model is based on the conventional formulation of the EVP rheology (Hunke and Dukowicz, 1997; Hunke and Lipscomb, 2008) and parameterizations of the grounding and arching land fast ice recently proposed by Lemieux et al. (2015, 2016) and Konig Beatty and Holland (2010). The model was configured in multiple rectangular domains and included several simplifications. In particular, we constrained ourselves to a single ice category, utilized a relatively simple but still widely used parameterization for the internal pressure (eq. 7), and employed a simplified non-linear Lax-Wendroff scheme for ice advection. We adopted these simplifications to reduce the complexity of the TLA codes, and this had negligible impact on the results at the 3-day time scale of conducted experiments.

It was found that TLA models for the EVP solver are unstable for the regions with high (>0.9) sea ice concentration and require stabilization. The standard stabilization technique, through the additional diffusion (Hoteit et al. 2005), widely used in the OGCM inverse modeling, was found to be inefficient, but a simpler stabilization, based on Newtonian friction, appeared to work well. Analysis of the TL approximation accuracy has shown that Newtonian stabilization has errors similar to the ones observed in the case of diffusion-based stabilization, and thus the Newtonian scheme can be successfully used in sea ice models based on the EVP solvers. In the last decade, several modifications of the EVP solvers were proposed to improve the convergence (e.g., Bouillon et al., 2013; Kimmritz et al., 2016; Koldunov et al., 2019). We assume that these ideas could benefit

the development of the respective TLA models. In particular, the Newtonian damping coefficient could be adjusted locally to account for the background sea ice pressure/tensile strength and thus provide a more efficient stabilization of the EVP TLA codes.

On the other hand, simple analysis (Figure 1, dashed line) indicates that the TLA model for the 1D VP sea ice solver (Stroh et al., 2019) is stable, provided that the forward model satisfies the stability criterion (which is always the case). Similar indications were obtained by Heimbach et al. (2010) and NAOSIM (Kauker et al., 2009), which did not observe instabilities in the TLA codes of the sea ice models with VP rheologies. We may, therefore, conclude that numerical models with VP rheology (e.g. Hibler et al., (1979); Lemieux et al., 2008) do not require additional stabilization of the TLA codes, and are formally more

suitable for development of sea ice 4dVar DA algorithms. Note, also, that due to their stability, VP TLA codes can be easily incorporated into the parent sea ice models to provide improved Jacobian-free Krylov solvers for VP rheology. This approach was employed in the ocean model by Nechaev et al. (2005), where the internal matrix-Free BiCG solver was constructed using the model's TLA codes in the IMEX algorithm, where the preconditioned Flexible GMRES algorithm was implemented in the Jacobian-free mode (Lemieux et al., 2014; Losch et al., 2014).

In a comprehensive series of OSSEs with a simplified EVP sea ice model, it was demonstrated that Newtonian stabilization of the TLA codes allows for a reasonable reconstruction of the RPs. The numerical experiments included two groups of 4dVar DA experiments.

    First, we analyzed the possibility of optimizing the RPs in two different landfast ice parameterization schemes incorporated in the CICE model (Hunke et al., 2010). In the current CICE6 version, all landfast ice parameters are treated as spatially

uniform variables, which degrades the accuracy of landfast ice simulations in various parts of the Arctic Ocean (Lemieux et al., 2015), especially in the shallow seas and narrow straits of the Canadian Archipelago. In this study, these parameters were specified by spatially variable functions with a reduced number (10-36, Section 2.4) of free parameters that were optimized to fit surface observations. The conducted OSSEs demonstrate that spatially varying landfast ice parameters $k_2$ and $k_T$, which are responsible for grounding and arching phenomena, can be optimized at a relatively low computational cost (5-12 iterations on

a sparse grid). We found that the impact of spatially uniform $k_T$ could be achieved by specifying $k_T$ only along a relatively narrow landfast ice boundary (Figure 2h), which works as a barrier to prevent ice drift under moderate wind conditions. This observation suggests that parameterizing landfast ice by spatially uniform $k_T$ can be reduced to specifying non-zero $k_T$ only within these localized regions of the domain. Interestingly, the optimization of both $k_2$ and $k_1$ requires a very small ($\sim 10$) number of iterations, which suggests the possibility of efficiently incorporating their optimization into a Pan Arctic operational

model only in regions of potential landfast ice arching or grounding phenomenon.

    Taking into account that landfast ice typically forms in shallow/coastal areas and in narrow straits, the landfast phenomena can be controlled more efficiently by adding to the control the critical thickness parameter $k_1$, which confines $k_2$ variability to shallow areas and narrow straits areas in a high-resolution pan-Arctic sea ice model. It should be noted that retrieval of $k_1$ is not straightforward because of non-differentiability of the grounding parameterization scheme (eq. 8). Optimizing $k_1$ in sea ice

models requires further development including more robust constrained minimization tools, such as genetic (Goldberg, 1989) or very fast simulation annealing (Ingber, 1989) algorithms. Another approach is to use parametric regularization of $k_1$, similar

to the one utilized by Nicolsky et al. (2009). In this case, TLA models require additional computation of the derivative with respect to the regularization parameter specifying a smooth approximation (e.g, Lemieux and Tremblay, 2009) of the Heaviside function in eq. (8). We are currently implementing this algorithm for the 2d VP solver.

In the second group of OSSEs, we analyzed the possibility of reconstructing spatially varying sea ice strength $P^*$ and ellipse axes ratio $e$ distributions. The respective OSSEs employed simulated observations of SIV and SIC characterized by the root-mean square errors of 0.025 m/s and 0.05, respectively, while SIT observations were simulated with uncertainties of 0.3 m, which is about two times smaller than the errors of the CryoSat-2 observations. The OSSEs also assessed sensitivity of the results with respect to systematic errors in wind forcing.

The OSSEs with accurate (exact) wind forcing demonstrated the feasibility of relatively accurate reconstruction of the $P^*$ distribution and less successful, but still reasonable, reconstruction of the axes ratio distribution. Similar results were recently obtained by Stroh et al. (2019), which used a 1d sea ice model featuring VP parameterization to find a higher impact of spatially varying $P^*$ than $e$ on the DA quality. We also observed that regions of less accurate $P^*$ reconstructions are typically co-located in the regions of strong sea ice divergence and/or SIC concentrations below 0.8, i.e. with the regions where rheology plays a lesser role in sea ice dynamics.

We also found that additional optimization of $P^*$ and $e$ (after optimizing the initial state of sea ice) provides a slightly more accurate reconstruction of the SIT and SIC distributions and a significant improvement of the SIV and $P_{tr}$ fields. Accurate forecasting of $P_{tr}$ is especially important for martime use, as it allows vessels to avoid regions with excessive compressive stress.

The OSSE with strong wind convergence in the pack ice ($A=1$) demonstrated even better quality reconstructions of $e$, and especially $P^*$. This can be attributed to the stronger role of internal stress on sea ice dynamics in pack ice. Similarly to the OSSEs with cyclonic wind pattern, we found that additional optimization of RPs provides smaller improvements in the SIT and SIC hindcasts as compared to SIV and $P_{tr}$. OSSEs with weaker winds ($\sim$ 0.0–0.15 N/m$^2$) demonstrated a slight reduction in the accuracy of reconstructed $P^*$ and $e$. This is because weaker winds generate smaller changes in the sea ice state and

observation errors contribute more to the results of assimilation. Note, that in the limiting case of zero (or very weak) wind and thick ice, the optimized $P^*$ is unconstrained and may take any value sufficient to keep ice in place.

Our 4dVar applications utilized a relatively small assimilation window (3 days) with an eye towards improving short-term sea ice forecasts in the ice pack and ice edge zones. In these regions, ice rheology typically changes at the time scales of several days (e.g. Panteleev et al., 2019) due to variations in the dynamic and/or thermodynamic forcing. In particular, wind variability

may cause profound changes of the ice edge. Meanwhile in the ice pack zones, wind forcing is the major driving factor of polynya formation and ridging processes, where rheological forces become important.

Experiments with cyclonic and zonal wind emphasized the importance of having accurate prior estimates of wind forcing: since sea ice dynamics is significantly controlled by winds (e.g., Thorndike and Colony, 1982), it is hard to expect a reasonable quality 4dVar reconstruction derived from a model driven by incorrect winds. In this case, both the model simulation and the

4dVar results will be inaccurate. However, a properly formulated 4dVar approach may still adjust wind forcing through the assimilation of the SIV/SIC/SIT observations (e.g. Stroh et al., 2019).

Finally, sea ice observations are typically available on a regular (daily) basis, and are expected to gain spatial coverage and accuracy in the near future for the SIC and SIV components that are directly observable from space, while SIT observations are still lacking accuracy due to the complex structure and uncertainties in its observation process. Ongoing developments in data acquisition and preprocessing will result in improved sea ice state observability, and this work has demonstrated the feasibility of reconstructing spatially varying RP fields on the basis of these data.

In the present study we utilized realistic observational errors for SIV and SIC, while SIT errors were somewhat smaller than the accuracy of currently available satellite observations. An additional set of experiments with more realistic SIT errors reveals their stronger impact on the reconstruction quality of $P^*$ and $e$, while the reconstruction accuracy of the landfast ice parameters $k_2$ and $k_T$ remained virtually unchanged. The incoming satellite platforms (e.g. ICEsat-2, *https://icesat-2.gsfc.nasa.gov*) with a better SIT observation capability may deliver sufficiently dense and accurate SIT observations required for reasonably accurate estimation of $P^*$ and $e$ in the internal regions of the Arctic Ocean.

Analysis of the potential impact of new observations, as well as more realistic inversions employing more complex rheological hypotheses (e.g., the Maxwell elasto-brittle rheology of Dansereau et al., (2016)), may be within the focus of our studies in the near future.

## Appendix A: Tangent linear numerics

The TL code was obtained by the subtracting a solution of the numerical model $X$ from the evolution equations of the perturbed state, $X + \delta X$, and keeping only linear terms in the expansions of all the nonlinearities. Note, that the easiest way to conduct this formal procedure is to apply a tangent liner and adjoint model compiler. In particular, this approach was used for the development of the MITgcm and NAOSIM sea ice models (e.g., Kauker et al., 2009) and basal sea ice model (Goldberg and Heimbach, 2013). Because of the availability of multiple automatic tools (e.g. TAMC (Giering and Kaminski, 1998; http://autodiff.com/tamc/), OpenAD (https://www.mcs.anl.gov/OpenAD/), we briefly outline the major details of the development of the EVP TLA models.

Perturbations of the auxiliary functions $\dot{\boldsymbol{\epsilon}}_{1,2,3}, m, \zeta, P_p, P, C_b$, of $X$ are given by

$$\delta\dot{\boldsymbol{\epsilon}}_1 = \partial_x\delta u + \partial_y\delta v; \quad \delta\dot{\boldsymbol{\epsilon}}_2 = \partial_x\delta u - \partial_y\delta v; \quad \delta\dot{\boldsymbol{\epsilon}}_3 = \partial_x\delta v + \partial_y\delta u; \quad \delta m = \tilde{\rho}(A\delta h + h\delta A) \tag{A1}$$

$$\delta\zeta = \delta P_p(1+k_T)/2\Delta^* + \delta k_T P_p/2\Delta^* - \delta\Delta^* P_p(1+k_T)/2\Delta^{*2} \tag{A2}$$

$$\delta P_p = (P_p^* h\delta A + P_p^* A\delta h + Ah\delta P_p^*)\exp(-\tilde{\alpha}(1-A)) + P_p\tilde{\alpha}\delta A \tag{A3}$$

$$\delta P = \frac{1}{\Delta^*}\left[\Delta\delta P_p + P_p\delta\Delta - P_p\frac{\Delta}{\Delta^*}\delta\Delta^*\right] \tag{A4}$$

$$\delta\Delta^* = \theta(\Delta - \tilde{\Delta}^*)\delta\Delta \tag{A5}$$

$$\delta\Delta = \frac{1}{e^2\Delta}\left[e^2\dot{\boldsymbol{\epsilon}}_1\delta\dot{\boldsymbol{\epsilon}}_1 + \dot{\boldsymbol{\epsilon}}_2\delta\dot{\boldsymbol{\epsilon}}_2 + \dot{\boldsymbol{\epsilon}}_3\delta\dot{\boldsymbol{\epsilon}}_3 - \frac{\delta e}{e}(\dot{\boldsymbol{\epsilon}}_2^2 + \dot{\boldsymbol{\epsilon}}_3^2)\right] \tag{A6}$$

$$\delta C_b = C_b\left\{\frac{\delta h}{h_b} - \frac{\delta A}{\tilde{k}_1} + \frac{\delta k_2}{k_2} - \frac{\boldsymbol{u}\cdot\delta\boldsymbol{u}}{\boldsymbol{u}^2 + \tilde{u}_0|\boldsymbol{u}|}\right\} \tag{A7}$$

Hereinafter, all the terms with variations of the RPs and atmospheric forcing and variations of the quantities that may contain those are underlined. Taking (A1-A7) into account, the TL equations of (13-19) are given by

$$\delta\sigma_1^s = (1+\varepsilon)^{-1}[\delta\sigma_1 - \varepsilon\underline{\delta P}(1-k_T) + \varepsilon P\delta\underline{k_T} + \varepsilon\underline{\delta P_p}(1+k_T)\dot{\epsilon}_1/\Delta^* + \varepsilon P_p\dot{\epsilon}_1\delta\underline{k_T}/\Delta^* +$$

$$+\varepsilon P_p(1+k_T)\delta\dot{\epsilon}_1/\Delta^* - \varepsilon P_p(1+k_T)\dot{\epsilon}_1\underline{\delta\Delta^*}/\Delta^{*2} - \boldsymbol{\varepsilon_N}\boldsymbol{\delta\sigma}_1] \tag{A8}$$

$$\delta\sigma_2^s = (1+e^2\varepsilon)^{-1}[\delta\sigma_2 + \varepsilon\underline{\delta P_p}(1+k_T)\dot{\epsilon}_2/\Delta^* + \varepsilon P_p\dot{\epsilon}_2\delta\underline{k_T}/\Delta^* + \varepsilon P_p(1+k_T)\delta\dot{\epsilon}_2/\Delta^* -$$

$$-\varepsilon P_p(1+k_T)\dot{\epsilon}_2\underline{\delta\Delta^*}/\Delta^{*2} - \boldsymbol{\varepsilon_N}\boldsymbol{\delta\sigma}_2] - 2\varepsilon e\underline{\delta e}(1+e^2\varepsilon)^{-2}(\sigma_2 + \varepsilon P_p(1+k_T)\dot{\epsilon}_2/\Delta^*) \tag{A9}$$

$$\delta\sigma_3^s = (1+e^2\varepsilon)^{-1}[\delta\sigma_3 + \varepsilon\underline{\delta P_p}(1+k_T)\dot{\epsilon}_3/2\Delta^* + \varepsilon P_p\delta\underline{k_T}\dot{\epsilon}_3/2\Delta^* + \varepsilon P_p(1+k_T)\delta\dot{\epsilon}_3/2\Delta^* -$$

$$-\varepsilon P_p(1+k_T)\dot{\epsilon}_3\underline{\delta\Delta^*}/2\Delta^{*2} - \boldsymbol{\varepsilon_N}\boldsymbol{\delta\sigma}_3] - 2\varepsilon e\underline{\delta e}(1+e^2\varepsilon)^{-2}(\sigma_3 + \varepsilon P_p(1+k_T)\dot{\epsilon}_3/2\Delta^*) \tag{A10}$$

$$(m\delta u^s + u^s\delta m)/\delta t_s - f(v^s\delta m + m\delta v^s) + \underline{\delta C_b}u^s + C_b\delta u^s + \tilde{C}_w\tilde{\rho}_w A|\boldsymbol{u}|(\delta u^s\cos\Theta - \delta v^s\sin\Theta) =$$

$$= \boldsymbol{\tau}_{wx}(u\delta u/\boldsymbol{u}^2 + \delta A/A) + (u\delta m + m\delta u)/\delta t_s + \underline{\delta\tau_{ax}} + \partial_x\delta\sigma_{xx}^s + \partial_y\delta\sigma_3^s \tag{A11}$$

$$(m\delta v^s + v^s\delta m)/\delta t_s + f(u^s\delta m + m\delta u^s) + \underline{\delta C_b}v^s + C_b\delta v^s + \tilde{C}_w\tilde{\rho}_w A|\boldsymbol{u}|(\delta u^s\sin\Theta + \delta v^s\cos\Theta) =$$

$$= \boldsymbol{\tau}_{wy}(v\delta v/\boldsymbol{u}^2 + \delta A/A) + (v\delta m + m\delta v)/\delta t_s + \underline{\delta\tau_{ay}} + \partial_y\delta\sigma_{yy}^s + \partial_x\delta\sigma_3^s \tag{A12}$$

$$\delta h^n = \delta t\,\mathrm{div}(\delta\boldsymbol{u}^n h + \boldsymbol{u}^n\delta h) + \delta t^2(\boldsymbol{u}^2\hat{D}\delta h + \delta(\boldsymbol{u}^2)\hat{D}h)/2 \tag{A13}$$

$$\delta A^n = \delta t\,\mathrm{div}(\delta\boldsymbol{u}^n A + \boldsymbol{u}^n\delta A) + \delta t^2(\boldsymbol{u}^2\hat{D}\delta A + \delta(\boldsymbol{u}^2)\hat{D}A)/2 \tag{A14}$$

The gradients with respect to RPs and atmospheric forcing control variables are given by

$$\frac{\delta\mathcal{J}_m}{\delta P_p^*} = -\varepsilon\mathcal{I}^\mathsf{T}\sum_t(1+k_T)Ah\exp(-\tilde{\alpha}(1-A))(\dot{\epsilon}_1\hat{\sigma}_1 + \dot{\epsilon}_2\hat{\sigma}_2 + \dot{\epsilon}_3\hat{\sigma}_3/2)/\Delta^* \tag{A15}$$

$$\frac{\delta\mathcal{J}_m}{\delta e} = \varepsilon\mathcal{I}^\mathsf{T}\sum_t\left\{\frac{2e}{(1+\varepsilon e^2)^2}[\sigma_2\hat{\sigma}_2 + \sigma_3\hat{\sigma}_3 + \varepsilon P_p(1+k_T)(\dot{\epsilon}_2\hat{\sigma}_2 + \dot{\epsilon}_3\hat{\sigma}_3/2)/\Delta^*] - \frac{P_p(1-k_T)}{e^3\Delta(1+\varepsilon)}(\dot{\epsilon}_2^2 + \dot{\epsilon}_3^2)\hat{\sigma}_1 +\right.$$

$$\left.+ \frac{P_p}{e^3\Delta^{*2}}\theta(\Delta - \tilde{\Delta}^*)(\dot{\epsilon}_2^2 + \dot{\epsilon}_3^2)\left[(1+k_T)\left(\frac{\dot{\epsilon}_1\hat{\sigma}_1}{1+\varepsilon} + \frac{\dot{\epsilon}_2\hat{\sigma}_2 + \dot{\epsilon}_2\hat{\sigma}_3/2}{1+e^2\varepsilon}\right) + (1-k_T)\frac{\Delta^*\hat{\sigma}_1}{1+\varepsilon}\right]\right\} \tag{A16}$$

$$\frac{\delta\mathcal{J}_m}{\delta k_T} = -\varepsilon\mathcal{I}^\mathsf{T}\sum_t P_p[(\dot{\epsilon}_1 + \Delta)\hat{\sigma}_1 + \dot{\epsilon}_2\hat{\sigma}_2 + \dot{\epsilon}_3\hat{\sigma}_3/2]/\Delta^* \tag{A17}$$

$$\frac{\delta\mathcal{J}_m}{\delta k_2} = \mathcal{I}^\mathsf{T}\sum_t C_b(\boldsymbol{u}\cdot\hat{\boldsymbol{u}})/k_2 \tag{A18}$$

$$\frac{\delta\mathcal{J}_m}{\delta\boldsymbol{\tau}_a} = -\mathcal{I}^\mathsf{T}\hat{\boldsymbol{u}} \tag{A19}$$

It is important to note that stability of the adjoint model, being strictly related to the stability of the TL model, contains the Newtonian damping given by the boldfaced terms $\boldsymbol{\varepsilon_N}\boldsymbol{\delta\sigma}_{1,2,3}$ in eq. (A8–A10). The adjoint model is integrated backward in time and requires the solution of the forward model for the each time step of the backward sub-cycling procedure. This can be achieved either through re-calculating the forward solution, or storing it, which includes storing all the intermediate states of the subcycling procedure, as well as space-time coordinates of the switches in the Heaviside functions. We elected the second option. Note that, formally, implementation of the 4dVar DA approach does not require the TL model (see eqns. 22–24). However, the development of the TL model cannot be avoided because of the necessity to check its tangent linear property and

validity of the respective Lagrangian identities for debugging the transposition procedure of the TL system matrix $\boldsymbol{M_X}$ whose action on the state perturbations is represented by equations (A8–A14).

## Appendix B:  On the stability of TL models with VP rheology

The VP system of equations is obtained by eliminating the time derivative in eq. (2), resolving the remainder with respect to $\boldsymbol{\sigma}$

$$\boldsymbol{\sigma} = \frac{1}{e^2}\left[\boldsymbol{P} - \boldsymbol{I}\frac{1-e^2}{2}\mathrm{tr}\boldsymbol{P}\right],$$       (B1)

and substituting (B1) into the momentum equation:

$$\tilde{\rho}hA(\partial_t + f\boldsymbol{k}\times)\boldsymbol{u} = \mathrm{div}\boldsymbol{\sigma} + \boldsymbol{\tau}_b + \boldsymbol{\tau}_a + \boldsymbol{\tau}_w$$       (B2)

$$\partial_t h = \mathrm{div}(h\boldsymbol{u})$$       (B3)

$$\partial_t A = \mathrm{div}(A\boldsymbol{u})$$       (B4)

where we used the notation $\boldsymbol{P}$ for the tensor in the rhs of (B1):

$$\boldsymbol{P} = P_p\left[(1+k_T)\dot{\boldsymbol{\epsilon}} - \frac{\Delta}{2}(1-k_T)\,\boldsymbol{I}\right]/\Delta^*$$       (B5)

The non-linear system of VP equations (B1-B4) is solved in two stages. First, equation (B2) is stepped forward using a semi-implicit scheme, which takes the fields of $P$ and $\Delta^*$ from the previous time step, treating only $\dot{\boldsymbol{\epsilon}}$ semi-implicitly. On the second stage, equations (B3-B4) are explicitly stepped forward in time using the velocity fields from the first stage (Hibler, 1979; Lemieux et al., 2008).

In application to TLA modeling, such procedure would not provide the exact derivative of the non-linear scheme with respect to $\boldsymbol{u}$ because equation (B2) is not linearized with respect to the variations of $\Delta(\boldsymbol{u})$ and will therefore exhibit behavior similar to the behavior of the regularized code (Fig. 1). Moreover, exact TLA code of the VP rheology, is intrinsically unstable in the regions of ice divergence (e.g., Gray and Killworth, 1995), especially in the areas, where tensile stresses associated with arching effects are important.

However, one may expect reasonable performance of the above mentioned "incomplete linearization" of the VP model in the 4dVar applications, as soon as the stability criterion of the non-linear model (B1-B4) is satisfied. We performed an additional experiment with 1d VP forward and TL models using the modified procedure featuring ten applications of the GMRES solver and ten $\Delta$ updates on every time step (Lemieux et al., 2008) and did not observe any instabilities in the TL model. In this

respect, we may conclude that VP rheology is less susceptible to the TLA instabilities than EVP, which requires introduction of the additional stabilization terms in the TLA code of the stress tensor evolution equation.

*Author contributions.*  All authors provided substantial contribution to the models' development, interpretation of the results, and writing the manuscript.

*Competing interests.* The authors declare that they have no conflict of interest.

*Acknowledgements.* This research was funded in part under the Naval Research Laboratory's 6.2 project "Modeling Arctic Landfast Ice (Program Element 62435N) and the 6.1 project "Optimization Rheology and Advancing Sea Ice Forecast" (Program Element 61153N). Oceana Francis was supported by the Coastal Hydraulics Engineering Resilience (CHER) Lab, the Civil and Environmental Engineering Department, and the Sea Grant College Program at the University of Hawaii at Manoa. Special thanks to the three anonymous reviewers for their comments which have significantly improved this manuscript.

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
