# Peer review of "Parameter Optimization in Sea Ice Models with Elastic-Viscoplastic"

_The Cryosphere, 2019_

## Referee Comment (RC1) · Anonymous Referee #1 · 15 Dec 2019

The manuscript "Parameter Optimization in Sea Ice Models with Elastic-Viscoplastic Rheology" by Panteleev et al describes assimilation/inversion experiments with a tangent linear and adjoint model of a dynamic sea ice model with an EVP solver. The assimilation window is short (3 days) and the experiments are designed (probably) with data assimilation for forecasts in mind. With this short window, the authors report some success in reconstructing unobserved parameters such as the ice strength P*, the ellipse ratio e, friction and tensile stress parameters in twin-experiments. The main results is that the authors managed to generated a stable approximate adjoint of the EVP solver, which has not been achieved (or published) so far. For that reason, the manuscript contains valuable material that should be published but the form of the manuscript requires work. Therefore I recommend major revisions.

[Figure]

Major comments:

The research questions behind this study are not entirely clear. The introduction describes in the last two paragraphs what is done (or will be described in the text), but not why. Instead the work is motivated by other work having done something similar. Based on the presented material, it's probably not difficult to phrase objectives and research questions, but the way the material is presented, it sounds a lot like a progress report without focus. Some of the experimental design choices (e.g. the very short assimilation window of 3days) could be easily motivated by central objectives/questions, but they are not. This makes the manuscript appear a somewhat random collection of experiments (I am exaggerating a little, but that's the impression I got).

This is not the first paper about sea-ice parameter optimisation. The results could have been discussed in the context of other published works, eg. Sumata et al (2019, DOI: 10.1175/MWR-D-18-0360.1), Kauker et al. (2009 doi:10.1029/2008GL036323), Massonnet et al (2014, doi:10.1002/2013JC009705), etc. even if their methods are not the same as here.

The main new technical achievement is the generation of a TLA of the EVP solver, but this work is only described in a very general way without paying attention to any detail. I do not think that this could be reproduced by a reader. A more detailed description of the EVP-adjoint (and regularisation) should be somewhere in the manuscript, maybe as an appendix.

The the presentation and the language of the manuscript is sloppy and th manuscript is sometimes difficult to read (unclear sentences, many small grammar and spelling errors). The list of authors contains at least one native speaker (I am guessing from the names), so that I would have expected an easier read. I marked a few smaller problems (see below), but since I am not a native speaker, either, I left many errors, inconsistencies and inaccuracies untouched (and I don't think that correcting this is my primary job as a reviewer), especially in the second half of the manuscript.

Further, the authors chose to use, in part, non-standard language and expressions so that it took me some time to reconcile formulae with previously published (and cited) literature. It's not clear to me, why the authors want to make the manuscript overly difficult to read. Using many (unnecessary) abbreviations doesn't make it any easier.

In summary, these comments (and also further comments below) do not address the core of the science in this manuscript (which I believe is solid), but the presentation of the material and of the relation to previous work requires an extensive and careful overhaul before the manuscript is ready for publication (and, in fact, for a review).

Minor comments and suggestions, (incomplete list of) typos and grammar problems: page 1 l1 not "the key" but "a key"

l7 a Newtonian

l16 unclear language: "the sea ice component of the global climate change becomes a more important factor"

l19: these are the same systems: Menemenlis et al. 2008; Heimbach 2008; Fenty et al. 2017, proper reference would be Heimbach et al 2010 (ocean modelling) for the adjoint model

l20: a [or the] visco

page 2 l24: "are not well suited for implementing" I think that this is too strong, or include a reference. They are more difficult to implement than explicit solvers.

l26: Again, this should be Heimbach et al 2010, also this is not the only system, there's also NAOSIM: Kauker, F. , Kaminski, T. , Karcher, M. , Giering, R. , Gerdes, R. and Voßbeck, M. (2009) Adjoint analysis of the 2007 all time Arctic sea-ice minimum, Geophysical Research Letters. doi:https://doi.org/10.1029/2008GL036323

l29: dump -> damp

l34: upon -> to
l36: it's not the eccentricity of the ellipse but the ratio of the to semi-major axes a/b

l41: RP: I would avoid this abbreviation. There are already too many abbreviations in the text, which make it more difficult to read. In general, I would try to reduce the number abbreviations to a minimum.

l50: eccentricity, s.a.

page 3 l56: there's also work by Peter-Jan van Leeuwen about using P* as a spatially varying control parameter in data assimilation (with a SIRFilter), can't find the reference now, unfortunately.

l65: more accurate reconstructions or a more accurate reconstruction

l29 delete "the"

l73: A similar approach

page 4 equation 1+2: This form of EVP has been found to produce noisy solutions, see, e.g. Hunke 2001, Lemieux etal 2012, Losch and Danilov 2012, Boullion et al 2013, and simple solutions to the problem exist (Lemieux etal 2012, Boullion et al 2013, Kimmritz etal. 2015, 2016). This may also greatly help with the stability of the TLA model of your code.

also: eq(2) is probably correct (maybe except for a factor of two in the time scale Td), but it was not easy to manipulate it to arrive at the equations described in Lemieux et al (2016). Please check, or provide a form in a language that the community (TC readers) will easily understand. Otherwise it feels like there is something to hide (I don't think there is, it's just the feeling that one gets when reading this).

l96: non-standard notation: I am used to \dot{\epsilon} for strain/deformation rate tensor, which \epsilon being the strain tensor (not the rate).

eq(6), correct, but unusual representation

l98, convergence depends on this choice. Again, for TLA codes I would prefer using a smooth regularisation to avoid additional non-differentiable expressions.

l100: not eccentricity, but ellipse ratio.

eq7: this is not what CICE uses by default, so the comparison to CICE is a little out of place.

l110: "their spatial variability". This now raises a more general question. What does it mean to use spatially varying parameters? Probably, that the parameterisation of ice properties is not correct and requires refinement. If a parameter fluctuates in space (and potentially time), what sense does the parameterisation make? A discussion of this would in place, either introduction of conclusions/discussion section

page 5 l115: very likely this is not enough to reach convergence (see Bouillon et al 2013, Kimmritz etal 2015, 2016). Will this be a problem for the adjoint? What is the adjoint of an iterative process? What is the adjoint of a non-converged iterative process?

l117 was -> were

l117-130 The description of how the TLA codes are derived is very hand-wavy and hard to follow. Consider a more accurate and detailed description (maybe in an appendix).

page 6 l147 for reproducibility alone, one needs to know what this term looks like in the corresponding equation(s). It's not clear which of the equations needs to be damped, or maybe all of them?

l163: remove "the" (in the similar experiments)

page 7 l165: please clarify if the TL/TLA codes of the VP model are part of this work or that of Stroh et al. The appendix is not very helpful in this context, because it only shows the VP equations and then some words about stability without explicitly naming the responsible variables, thresholds etc.

l177: acronym SIT not explained. Previously this was called SIH (line 61)

page 8 Table2: kT =0.6 is already very high

Table2: kT =15 is that realistic? Or a typo?

Table2: Spelling: Truie -> True

l196: "which was set to 3 days", that's short

l198: diagonal error covariance matrices? But in lines190/191 there are decorrelation scales for 150km and 7 days. How can the prior error covariances be diagonal?

In general, the cost function should be made explicit, especially the regularisation terms. Otherwise there is no chance of reproducibility.

l203: the feasibility

l204: the feasibility

page 9 l207: (or 15)???

l215 by steady 10 m/s winds

l221 forming a polynya

l223: perturbed instead of disturbed initial SIT and SIC fields, but why make it harder at this point?

It's not clear which pseudo data are assimilated. Fig2 is strange, with noise-like stripes near x=600km, y < 20km after 3 days.

page 10 l245: why these choices and not the values suggested by Lemieux et al 2015/2016?

page 11 Fig3 caption says k2=15, but text says 16

page 12 l247: most of the domain

[Figure]

Section 3 What do we learn from the optimisation of k2? In the parameterisation, k1 determines where basal stress is increased, k2 scales the stress, so that for k2=0 the parameterisation is turned off.

Also the solution should depend linearly on k2, because just scales the friction/decceleration.

l275: GYRE-0/W, 0/W is not defined in the text anywhere

l277 the feasibility

page 14 l292: "The simulated data mimics realistic observations such as those obtained from sources discussed in section 2c" but without any possible bias

l299 why are these two steps required? Doesn't that work against the philosophy of an inversion? Is it not possible to optimise all control parameters at the same time?

page 16 l310: "The minor impact of Crh optimization on the SIT is probably due to relatively high SIT errors and substantial difference between the first guess and observed SITs." Maybe the ice thickness just does not depend so much on e and P* on these short timescales, with low ice concentration (when the ice is in free drift anyway), should be discussed somewhere (in the discussion/conclusions section?)

l312: "In contrast, ...." I'd rather say, "as expected"

l317: southwest!!

l318: remove "of the"

l317/318: sentence unclear, as a consequence, I don't understand the explanation

page 17 l335: what do we learn about "observability"/"controllability" of the solution? P and e can be tuned to make up for any systematic errors in the forcing? How will that improve the solution (e.g. with respect to predictability)? It's not clear to what extent the initial conditions of SI[C,T,V] are important in this experiment.

4.2 Section headline: what does PIZ stand for?

page 18 l355: there are no middle panels in Fig7, bottom panels?

page 20 l369-371: "This issue is important because in realistic sea ice forecasts, improper prediction of Ptr may result for mechanical damage of ships due to extensive sea ice compression." should be part of a discussion

l391: "OGCM inverse modeling was found to be inefficient, but a simpler stabilization based on Newtonian friction appeared to work well." It's not clear how this was done.

l394: where was this shown?

l396 (and acknowledgements): Lemeaux: do you mean Lemieux?

page 21 l405: (10-15): where does this range of numbers come from? I counted 7: initial conditions for u, h, A, kT, k2, e, P*

l425, algorithmically, assimilating ice drift should not have too much of an effect on the model drift, because the information is lost in the EVP iteration: the result of the EVP solver does not really depend on the initial conditions at the beginning of the iteration, but only on the forcing and solver parameter. That is why adjusting the solver parameters P* and e has such a large impact on the ice drift. I think the experiments at least provide some evidence for this interpretation.

page 22 l455: the solution technique outlined here is not what is usually done in implicit VP-solvers. P = P(h,A) is usually held fixed as the value of the previous timestep (although this is not a requirement, see IMEX in Lemieux et al. 2014, doi:10.1016/j.jcp.2014.01.010), but \Delta is updated in the non-linear Picard iteration making the entire iteration very stiff (hence, the attempts with JFNK, and their failure, that are also cited in this paper). If \Delta is held at t-1, then the entire problem is linearised and much simpler to solve, and I would agree with this assessment. But it refers to a system that is not used in practice, and would give very different results, too.

page 25 l532: in press JTECH, appears to be online: https://doi.org/10.1175/JTECH-D-18-0239.1, unfortunately I don't have access to this journal.

---

## Referee Comment (RC2) · Anonymous Referee #2 · 17 Dec 2019

Review of "Parameter Optimization in Sea Ice Models with Elastic-Viscoplastic Rheology" by Panteleev et al.

The manuscript presents parameter optimization of an EVP sea ice model using the tangent linear and adjoint method. Experiments in the manuscript generally show the capability of the inverse method to optimize the model initial states and also the parameters. Although current study is based on the ideal experiments, it does show promising future if it can be further extended to the real application with a long-term optimization.

The science is fine in current manuscript, and the authors reported that their tricky that adding Newtonian damping term in the adjoint equation stabilize the numerical performance, which should be highlighted but are missing in the text. Applying adjoint

methods especially in the sea ice model is a difficult work and the readers are keen to see if there really are some advances on this field. The analytic differentiation as reported by the authors, the damping term and even the codes should be publicly accessible at least from the appendix or the supplement materials, though there is something still not clear for me.

Apart from these points, I would like to say the manuscript is not well-prepared. It seems to be a draft on its first version. The context is little bit tedious on some unnecessary parts from my feeling and ignores too many details that, however, should be elaborated. I guess the co-authors even did not really go through the manuscript, let alone help to improve the text. Too many small grammar mistakes that, however, can be easily corrected by grammar check in MS word or spelling check if you use the Latex! All your citation styles in the text should be also taken care of. I would suggest the authors should really shoulder their own responsibility on their manuscript, not the reviewers! Overall, I suggest substantial revisions.

Comments:

L24: About abbreviations such as SIM, SI, LFI. . ., I indeed find it does not improve but reduce the readability of the text.

L33: As above, the citation style. Add '.' after 'al'.

L35: 'the sea ice rheology is defined by . . .' needs to be rephrased. I think the word 'defined' is not proper.

L56-58: Better to remove this paragraph. I did not find any connection with the context. The stochastic parameters are locally varied, but this is actually another story when stochastic effects are considered.

L66: spelling mistake: analyize

L90: what are the div and tr? Please state clearly in the text!

L95: det. the same as L90

L101: "SI rheology: for P:". remove one of the colons

L110: "simulated (satellite) observations". I did not really get the point what is the simulated observations. I believe you want to say 'synthetic observations'.

L117: Please break this long sentence into shorter ones and elaborate how did you deal with the analytic differentiation of the equation in the appendix. I also wonder how is the $\Delta$ , which is highly non-linear, be processed.

L130: About the 'TL code', since the model is not such complicated, please make all your 'TL code' publicly accessible for better reproducibility for the community.

L144: Regarding the 'spatial spectrum', it's not clear that what kind of spatial spectrum you refer to.

L148: Regarding the 'Newtonian friction term', please implicitly show the equation and the damping time scales that you used

Figure1: Please consider to use dotted line. it can obviously show how many experiments you did.

Section 2.3: I would significantly simplify this section, since only the observation errors are used. You do not need to introduce all these. When I read this section, I was thinking about the experiments are dealing with the real observations. But actually, I think for the ideal experiments, these observation errors only set a reference.

Table2: PIZ experiment. Spelling issue: 'Truie'

L194: Spelling issue: 'wass' Line 206: remove the second 'and'

L224: the assimilation window is really short. I just wonder if the experiment results show sensitivities on the assimilation window.

L232: what is '$\omega$'?

L235: what is 'DAS' ?!

Section 3.1: the configuration of the experiment is not clear. For example, is the initial SIC condition symmetric? It seems not from Figure 2a. How is the boundary condition? And in the text, the authors should explain why the spatial distribution of the polynya is not symmetric over the y-axis. Coriolis effects or the initial condition effects?

L253: spelling issue: 'separting'

L275: GYRE-0/W. Elaborate the means of the abbreviations

L278: It's not clear why you use such weird initial SIC distribution.

L292: time step

L293: Please say clearly how you mimic the realistic observations, just their magnitude?

L307: remove 'is'

Figure 5: What is 'SIH'????

Figure 6: In the caption, 't=3 d' should be 't= 3 days'

L310: I wonder whether you optimize Crh first then the initial conditions, you could get the same conclusion.

L318: remove one 'of the'

L338: 'the western part agrees well with true e distribution'. Actually, only part of. I think the authors could just say something like " show parts of agreement". L343: "and therefore has a minor rheological impact of the sea ice dynamics". That is not the case, as most parts in fig 5b still have SIC >= 0.95.

L368: Spelling, "maximuma"

L388: "Koning Beatty and Holland (2010)"???????????????????

[Figure]

L414: the authors never defined RMSE!

L431: I do not know why it is worth to address the realistic observation errors are used.

About the conclusions: it's currently too long. Please try to simplify what you really want to say.

———————————————

---

## Referee Comment (RC3) · Anonymous Referee #3 · 13 Jan 2020

Review of manuscript "Parameter Optimization in Sea Ice Models with Elastic-Viscoplastic Rheology", Gleb Panteleev, Max Yaremchuk, Jacob N. Stroh, Oceana P. Francis, and Richard Allard

General comments:

The paper reports result of sea ice data assimilation experiments using the 4D-var (adjoint gradient based) assimilation technique. The parameters being adjusted/optimized in the data assimilation procedure include highly non-linear sea ice rheology parameters as well as sea ice initial conditions. In terms of novelty, this research extends existing works on 1D sea ice dynamics to now fully 2D, which allows for more realistic dynamical treatment of the sea ice motions and deformations. The focus is on the EVP rheology which is used in CICE, the sea ice model widely used by the community. The

number of experiments cover a reasonable range of sea ice regimes. Overall, the scientific content of the paper has potentially good quality and should be considered for publications after what I'd consider moderate revisions to address some of the issues I'd like to outline below.

Specific comments on Scientific quality:

The derivation of the TL model and its adjoint is commendable. Sea ice rheology is highly non-linear, so it is not surprising that damping is required in the adjoint, and that the adjoint gradients and reconstructions appeared to be stable for only very short time-scale. The assimilation window used here is 3-day.

1. On the clarity and evidence provided to support the experiment results, for scientific reproducibility purpose, I think the derivation of the equations of the tangent linear and the adjoint models should be made available, likely in the Appendix, in addition to only descriptive wordings in the main text (Section 2.2), so the readers can assess the impact of the linearization and damping on the sensitivities and reconstructions.

2. On basing their development of the TL and adjoint codes on the EVP rheology, can the authors discuss the physical meaning of their results in the context of published works reporting issues on convergence with the EVP rheology, e.g., Lemieux et al., [2012], Losch and Danilov [2012]? In addition, can the authors discuss how relevant/applicable/adaptable their TL and adjoint code development would be in light of the availability of more recent modified EVP solvers, e.g., Koldunov 2019?

3. Specific to the short assimilation window (3 days), the purpose of the work is not clearly articulated, other than to point out that they are extending on previous works (e.g., of Stroh et al., 2019). Is the goal, given the expected non-linearity, to achieve short-term (days) forecasting?

4. On the same subject of the short assimilation window, I think the authors need to provide an assessment on the meaning of the "optimized" parameters. Specifically,

are the adjustments and optimized values reflect physical values relevant to various sea ice regimes or whether they are merely for the purpose of curve-fitting. In addition, due to the 3-day window, what does it mean if these optimized 2-D fields of the ice parameters change / are discontinuous every 3-day or so?

5. The adjoint gradients, where stable, are powerful in that they reflect dynamical connections, and thus allows one to extract meaningful physical connections relating the control space (rheology and ice dynamics parameters and initial condition) and the sea ice state (fast ice, seasonal/marginal ice, thick, thin, etc.). However, due to the damping/regularization, it is not clear if these adjoint gradients contain physics, or whether they are simply numeric for use in a misfit reduction procedure. For transparency purpose, it would be good if the authors can provide a couple of figures on the gradients.

6. The authors mentioned why the relatively highly important parameters k1 cannot be part of the control space due to the non-linearity. Due to this reason, I believe the results in this manuscript is incomplete: I think there should be a discussion, and perhaps at least 1 or 2 sets of additional experiments conducted identically to those for k2 and P*, but for different k1 values, to gauge how sensitive their optimized k2 and P* are to other important parameters. In other words, one would like to understand whether results presented in this manuscript are robust and physically meaningful (e.g., the adjoint gradients are physical, the optimized rheology parameters yield useful information about their dependence on ice regimes), or whether they contain no physical meanings beyond curve-fitting.

Technical corrections: There are many misspelled words, including misspelled authors in citations ("Lemiuex" line 105, "Zhnag" on line 555). Only a few I spotted are listed here, but the authors should run a spellcheck through this. Lines: 66, 85, 105, 169, 194, 368, 454, 455, 495, 506.

Extra commas should be removed on lines: 319, 460

Need an extra ")" on line 442.

"SIT" was first introduced without spelling out on line 129.

"SIH" and "SIT" are scattered through the article, and I believe are meant the same thing, the authors should settle on one after defining them.

Figure 2 caption: "Left panel shows..." should be "Right panel shows..".

Line 355: ".. in the middle panels.." should be ".. in the bottom panels.."

---

## Author Comment (AC1) · 4 Mar 2020

Manuscript TC-2019-219-RC1

Parameter Optimization in Sea Ice Models with Elastic-Viscoplastic Rheology

by Panteleev et al

Response to Reviewer 1

We would like to thank the Reviewer for useful comments that helped us to significantly improve the manuscript.

Reply for MAJOR COMMENTS:

[Figure]

Reviewer: The research questions behind this study are not entirely clear. The introduction describes in the last two paragraphs what is done (or will be described in the text), but not why. Instead the work is motivated by other work having done something similar. Based on the presented material, it's probably not difficult to phrase objectives and research questions, but the way the material is presented, it sounds a lot like a progress report without focus. Some of the experimental design choices (e.g. the very short assimilation window of 3days) could be easily motivated by central objectives/questions, but they are not. Reply: The major objective of our study was to find an appropriate way of optimizing an extended set of control variables in the sea ice models based on EVP rheology (e.g. CICE model). This set of control fields includes spatially varying rheological parameters, initial conditions and forcing fields. In addition to the above, we assume that novel aspects of the study include development and validation of the regularization algorithm for TL and adjoint models for ice models with EVP rheology and validation of the DA algorithm based on the EVP TL and ADJ models (4Dvar) through the multiple Observing System Simulation Experiments. Also, most of the operational sea ice observations are available daily and our study is specifically aimed at the short-term forecasts. Because of that, the 3-day data assimilation window appears reasonable. We assume that temporal variability of the sea ice has even smaller time scales in the MIZ zone where the pancake ice can be replaced with very different ice category in less than a week. From our point of view, it is natural to assume different P* and e values for 0.1-1 m floes and sea ice floes larger than 0.5-1 km. Therefore, we do not see a necessity to increase the DA time window for the period more than 1 week. In the revised version we put more emphasis on these novel features in the abstract (lines 6-8) and Introduction (lines 62-75).

Reviewer: This makes the manuscript appear a somewhat random collection of experiments (I am exaggerating a little, but that's the impression I got) Reply: We do not completely agree: in the four series of numerical experiments presented in sections 3-4 we consecutively focus on optimization of the four rheological parameter fields using data assimilation with simulated observations. This is a standard way to present

new data assimilation schemes (see, for example, Goldberg, D.N. and P. Heimbach, 2013). Since the 90s this methodology is typically called "twin-data experiments" (e.g. Vossepoel, and van Leeuven, 2007). Later the term "observing system simulation experiments" became more popular. Throughout the manuscript, we use both terms (e.g., lines 89-95))

Reviewer: This is not the first paper about sea-ice parameter optimization. The results could have been discussed in the context of other published works

Reply: Our paper is focused on simultaneous optimization of the initial conditions, external forcing (wind), and rheological parameters. Therefore, we assume that our study falls into the cathegory of variational data assimilation in ice modeling, which is (to the best of our knowledge) currently carried out using MIT and NAOSIM numerical models. We added a short discussion of other approaches to optimization of the rheological and other sea ice model parameters (lines 61-70, 83-86) Unfortunately, we failed to find a paper by Van Leeuwen related to P* optimization in his publication list (see our comments below). Also, we could not find the analysis of RP optimization in the paper of Kauker et al, 2010, However, we now use this reference in the discussion of weak sensitivity of the sea ice model solutions with respect of the initial sea ice velocities (Lines 84, 200, 464)

Reviewer: The main new technical achievement is the generation of a TLA of the EVP solver, but this work is only described in a very general way without paying attention to any detail. I do not think that this could be reproduced by a reader. A more detailed description of the EVP-adjoint (and regularisation) should be somewhere in the manuscript, maybe as an appendix. Reply: The details are now given in the new Appendix A. We also found some other inaccuracies in previous manuscript and corrected them.

Reviewer: The presentation and the language of the manuscript is sloppy and the manuscript is sometimes difficult to read (unclear sentences, many small grammar

and spelling errors). The list of authors contains at least one native speaker (I am guessing from the names), so that I would have expected an easier read. I marked a few smaller problems (see below), but since I am not a native speaker, either, I left many errors, inconsistencies and inaccuracies untouched Reply: Thanks for your help. Two native English speaker co-authors checked the gramma in the revise version of the manuscript.

Reviewer: The authors chose to use, in part, non-standard language and expressions so that it took me some time to reconcile formulae with previously published (and cited) literature. It's not clear to me, why the authors want to make the manuscript overly difficult to read. Using many (unnecessary) abbreviations doesn't make it any easier. Reply: We assume that eq. (1)-(4) concisely describe rheological and dynamical constraints of the modern ice models and provide a better insight on the stability properties of the respective linearized systems. Following the Reviewer's request, we added a detailed description of the numerical scheme, and the respective TL and adjoint codes in the Appendix. The number of abbreviations was reduced significantly, leaving only those that are used in the text more than 30 times.

MINOR COMMENTS:

Reviewer: l16 unclear language: "the sea ice component of the global climate change becomes a more important factor" Reply: This sentence was modified. See lines 18-20 of the revised version of the manuscript.

Reviewer: l19: these are the same systems: Menemenlis et al. 2008; Heimbach 2008; Fenty et al. 2017, proper reference would be Heimbach et al 2010 (ocean modelling) for the adjoint model Reply: This part of the text was modified accordingly (line 21-22).

Reviewer: page 2 l24: "are not well suited for implementing" I think that this is too strong, or include a reference. They are more difficult to implement than explicit solvers. Reply: We agree. The sentence was modified (line 27)

Reviewer: l26: Again, this should be Heimbach et al 2010, also this is not the only system, there's also NAOSIM: Kauker, F. , Kaminski, T. , Karcher, M. , Giering, R. , Gerdes, R. and Voßbeck, M. (2009) Adjoint analysis of the 2007 all time Arctic sea-ice minimum, Geophysical Research Letters. doi:https://doi.org/10.1029/2008GL036323 Reply: Corrected: (line 30).

Reviewer: l41: RP: I would avoid this abbreviation. There are already too many abbreviations in the text, which make it more difficult to read. In general, I would try to reduce the number abbreviations to a minimum. Reply: We reduced the number of abbreviations substantially. However, we kept the abbreviation RPs for rheological parameters because it is used more than 30 times in the paper. In our opinion, removing this abbreviation will increase the manuscript length and decrease its readability.

Reviewer: l56: there's also work by Peter-Jan van Leeuwen about using P* as a spatially varying control parameter in data assimilation (with a SIRFilter), Reply: Sorry, we failed to find a paper by Peter van Leeuwen in his list of the publications at https://research.reading.ac.uk/meteorology/people/peter_van_leeuwen/. The only relevant publication we managed to find as an abstract at the 2008 Ocean Sciences conference in Orlando. From this Abstract, it is not clear what are the actual results of applying the SIR filter to the sea ice dynamics. In our recent personal communication with Peter Van Leeuwen said that these results have never been published in the final form. He also mentioned that they found some seasonal variability of the P*.

Reviewer: page 4 equation 1+2: This form of EVP has been found to produce noisy solutions, see, e.g. Hunke 2001, Lemieux et al 2012, Losch and Danilov 2012, Boullion et al 2013, and simple solutions to the problem exist (Lemieux et al 2012, Boullion et al 2013, Kimmritz et al. 2015, 2016). This may also greatly help with the stability of the TLA model of your code. Reply: We do agree with the Reviewer that EVP models may require extremely large number of sub cycles for proper convergence, and that the cited papers provide (partial) solutions to the problem at the expense of certain increase of complexity of the EVP numerics. However, our major objective was to

develop a RP optimization method based on the CICE5 representation of ice rheology, which is consistent with eq. (2). Also, we do not think that stability of the TLA can be related to the potential noise in forward solution. Numerical instability of the TLA models is a well known problem with explicit numerical schemes and Martin Losch observed the similar instability in their experiments with MIT ice model featuring EVP solver. We now mention this in Line 200-203 of the manuscript.

Reviewer: eq(2) is probably correct (maybe except for a factor of two in the time scale Td), but it was not easy to manipulate it to arrive at the equations described in Lemieux et al (2016). Please check, or provide a form in a language that the community (TC readers) will easily understand. Reply: We now provide an extended overview of the equations and their finite-difference approximation in Appendix A.

Reviewer: l96: non-standard notation: I am used to ndot{nepsilon} for strain/deformation rate tensor, which nepsilon being the strain tensor (not the rate). Reply: Notation has been changed in accordance with the Reviewer's request (lines 115,123, 125)

Reviewer: l98, convergence depends on this choice. Again, for TLA codes I would prefer using a smooth regularization to avoid additional non-differentiable expressions. Reply: We are aware of such problems. See, for example Nicolsky et al, (2009), where a parametric smoothing regularization was applied for unfrozen water content in the heat transfer equation for permafrost. Note however, that unfrozen water content is a key parametric function because phase transition (freezing/thawing) causes discontinuities in the behavior of the control field (diffusion coefficient), resulting in the unbounded growth of the derivatives. For sea ice dynamics conventional regularization of max$(\Delta, \Delta^*)$ has a minor impact on convergence because $\Delta^*$ is small enough and regularization is required only to avoid occasional discontinuities in the derivatives. We added discussion of the subject to section 5 including a reference to Lemieux and Tremblay, 2009, who proposed approximating Heaviside functions in the definition of max$(\Delta, \Delta^*)$ by a tanh-like function (lines 487-489).

Reviewer: eq7: this is not what CICE uses by default, so the comparison to CICE is a little out of place. Reply: We do not completely agree: Hibler's 1979 parametrization is among the options in CICE5 model, and has been extensively used in many operational runs. There is also a certain evidence (e.g., Ungermann et al, 2017) that this part of the EVP model has little effect on the stability of the TL EVP solver. Respective discussion has been included in section 5: lines 449-450).

Reviewer: l110: "their spatial variability". This now raises a more general question. What does it mean to use spatially varying parameters? Probably, that the parameterization of ice properties is not correct and requires refinement. If a parameter fluctuates in space (and potentially time), what sense does the parameterization make? A discussion of this would in place, either introduction of conclusions/discussion section. Reply: We are not sure that we understand your remark correctly. Existing parametrizations (e.g. Hibler's) inherently suggest that $P^*$, $e$, are fixed (i.e. do not depend on environmental conditions). This is a reasonable initial hypothesis but there are many indications that $P^*$ and probably $e$ should be different in different regions for multiple reasons: e.g., ice age, different floe structure etc. Of course, it would be very useful to derive a new parametrization which treats, for example, $P^*$ as a local function of floes statistics and ice age, but this problem lies beyond the scope of our manuscript. From our point of view, there are multiple indication concerning why $P^*$ and $e$ should not be constant. For example: different thysical properties of the cake ice ($\sim$20m) and large floes sea ice (> 1km). Sea ice salinity/temperature also impact the sea ice strength. Additional discussion of the reasons why $P^*$ and $e$ should vary in the Arctic is provided. (Lines 96-99, 509-517 of the revised manuscript).

Reviewer: page 5 l115: very likely this is not enough to reach convergence (see Bouillon et al 2013, Kimmritz etal 2015, 2016). Will this be a problem for the adjoint? What is the adjoint of an iterative process? What is the adjoint of a non-converged iterative process? Reply: In several experiment we increased number of subsycling iterations up to 2000 and did not reveal substantial difference in the inverse problem solutions. Note however, we did not check full convergence of our solutions to the "true VP solution" in a way recently discussed by Lemieux and Dupont, 2020 (https://www.researchgate.net/publication/337288766). Note, that TLA models are built in the vicinity on a non-linear solution of the forward model, and it does matter whether that solution is "fully converged to the true forward solution" or not. Intuitively, this follows from simple considerations: Let us assume that H(x) and A(x) are constant in time and any changes in H(x) and A(x) are compensated by some "additional" thermodynamic processes, which can be easily included into advection equations (3-4). In that process of the integration of the system (1-4) will be equivalent to the increasing the number of the subcycles, and the correspondent TLA will blowup anyway. Lines 145-148.

Reviewer: l117-130 The description of how the TLA codes are derived is very hand-wavy and hard to follow. Consider a more accurate and detailed description (maybe in an appendix). Reply: We now provide a more detailed description in Appendix A. See also lines 623-630.

Reviewer: page 6 l147 for reproducibility alone, one needs to know what this term looks like in the corresponding equation(s). It's not clear which of the equations needs to be damped, or maybe all of them? Reply: Now these terms are explicitly given in the Appendix A (eqns A18-20)

Reviewer: page 7 l165: please clarify if the TL/TLA codes of the VP model are part of this work or that of Stroh et al. Reply: The TL approximation errors for VP rheology in Fig. 1 are shown for the 1d model of Stroh et al (2019). The figure caption has been updated to clarify the point. See also Line 196.

Reviewer: l177: acronym SIT not explained. Previously this was called SIH (line 61) Reply: Actually, the abbreviation is SIT for sea ice thickness everywhere. Now corrected throughout the text

Reviewer: page 8 Table2: kT =0.6 is already very high Reply: We do not think that

kT=0.6 is too high. As an example, Tremblay and Hakakian (2006) estimate values of 0.5 to 0.8 for kT from their analysis of satellite-derived sea ice drift maps. We modified this part of text (lines 260-261) providing a justification for the choice of the reconstructed field of kT.

Reviewer: Table2: kT =15 is that realistic? Or a typo? Reply: This is a typo. Corrected to k2.

Reviewer: l196: "which was set to 3 days", that's short Reply: As we now explicitly state in the Introduction, the primary objective of the study is to improve sea ice forecasts for the periods 3-7 days (lines 96-99, 509-517).

Reviewer: l198: diagonal error covariance matrices? But in lines190/191 there are decorrelation scales for 150km and 7 days. How can the prior error covariances be diagonal? Reply: We agree that, ideally, they should be characterized by non-diagonal inverse error covariance matrices. However, in real applications observation errors are assumed to be diagonal, mostly because confident information on the space-time variability of the decorrelation scales is rarely available. This uncertainty in the formulation of the cost function is partly compensated by the smoothness regularization terms (now explicitly shown in the Appendix A, (eq A25) whose magnitude implicitly introduce spatial scales in the variation of respective error fields.

Reviewer: In general, the cost function should be made explicit, especially the regularization terms. Otherwise there is no chance of reproducibility Reply: Now explicitly given in the Appendix A (eq. A25).

Reviewer: l223: perturbed instead of disturbed initial SIT and SIC fields, but why make it harder at this point? Reply: Corrected. In all the experiments exposed in Fig. 2 (except for the dashed line in Fig. 2e) the first guess fields of SIT and SIC were not perturbed.

Reviewer : It's not clear which pseudo data are assimilated. Fig2 is strange, with

noise-like stripes near x=600km, y < 20km after 3 days. Reply: In the KT experiments presented in Fig. 2 (with the exception of the dashed line in Fig. 2e) initial conditions for SIT and SIC were not optimized. If initial conditions for SIT, SIC are not perfect, we may use (dense) SIT/SIC observations and optimize them as well. Behavior of the cost function in this experiment is shown in Figure 2e by the dashed line. Emergence of the noise-like features after 3 days of integration in the previous version of the manuscript were due to several reasons: a) SIC/SIT initial conditions along the northern and southern boundaries had the form of a narrow tongue (1 grid point wide and 3 grid points long ). b) Dispersive properties of the Lax-Wendroff advection scheme. c) Effects of the Matlab function PCOLOR utilized for plotting. By default, this function uses cubic spline which tends to produce grid scale noise in the regions of sharp gradients. To diminish these effects, we slightly modified the shape of the initial condition along the northern and southern boundaries and utilized a different plotting procedure. Note that this feature is absent in K2 experiments because initial conditions were smooth. Note, also that new initial conditions result to more efficient minimization because the "true" solution is less noisy.

Figure 1

See Lines 239-243 , 279-281, 295-298 and modified Figure 2.

Reviewer: page 10 l245: why these choices and not the values suggested by Lemieux et al 2015/2016? Reply: There is some misspelling in specifying: b and k2 here: they were actually set to be equal to 20 and 15 (as in Table 1) in our experiments. We took these values from Table 1 of Lemieux et al., 2016, (k1=8, k2=15, b =20); the misspelled values and the respective citation added (lines 290-295).

Reviewer: page 11 Fig3 caption says k2=15, but text says 16 Reply: Corrected to k2 = 15.

Reviewer: Section 3 What do we learn from the optimisation of k2? In the parameterisation, k1 determines where basal stress is increased, k2 scales the stress, so that

for k2=0 the parameterisation is turned off. Reply: Our K2 OSSE shows that value of the k2 can be relatively easily and accurately retrieved from sea ice observations. This property creates the prerequisite for operational optimization of the K2 and improving short range sea ice forecast. We agree with the Reviewer that adding k1 to the control parameters would be beneficial. However, retrieving the value of k1 from observations by 4dVar method is not straightforward due to essential non-differentiability. We more discussion of the subject (lines 487-490).

Reviewer: Also the solution should depend linearly on k2, because just scales the friction/ decceleration. Reply: We agree. This statement (on linear dependence on k2) is given in lines 308-311. (lines 263-264 of the original manuscript)

Reviewer: l275: GYRE-0/W, 0/W is not defined in the text anywhere Reply: These experiments (as well as KT and K2) are now named in lines 239-243, 240-250

Reviewer: page 14 l292: "The simulated data mimics realistic observations such as those obtained from sources discussed in section 2c" but without any possible bias Reply: We added the respective comment (lines 346-349).

Reviewer: l299 why are these two steps required? Doesn't that work against the philosophy of an inversion? Is it not possible to optimize all control parameters at the same time? Reply: Actually, lines 298-301 in the original manuscript describe the three-step optimization. Simultaneous optimization of all the controls can be done only for "well-behaved" (e.g. quadratic) cost functions with unique minima. In our case, the non-linear minimization problem obviously appears to have multiple minima and finding a physically sensible first guess control vector was a necessity, which was realized in our case in the form of initial two-step minimization. At the third sweep, all control variables were optimized simultaneously. So in that sense, we do not see any controversy to the philosophy of the inversion. See line 355-360 in the original manuscript.

Reviewer: page 16 l310: "The minor impact of Crh optimization on the SIT is probably due to relatively high SIT errors and substantial difference between the first guess and

observed SITs." Maybe the ice thickness just does not depend so much on e and P* on these short timescales, with low ice concentration (when the ice is in free drift anyway), should be discussed somewhere (in the discussion/conclusions section?)

Reply: We agree. A remark was included in the text (lines 370-371).

Reviewer: l317/318: sentence unclear, as a consequence, I don't understand the explanation Reply: The sentence was rephrased (lines 378-379).

Reviewer: page 17 l335: what do we learn about "observability"/"controllability" of the solution? P and e can be tuned to make up for any systematic errors in the forcing? How will that improve the solution (e.g. with respect to predictability)? It's not clear to what extent the initial conditions of SI[C,T,V] are important in this experiment. Reply: To explore this issue, a comprehensive adjoint sensitivity analysis (e.g. Kauker et all, 2009) has to be conducted, but this goes beyond the scope of the present study, which has an objective to demonstrate feasibility of RP optimization. Note, that because of stability it is more reasonable to conduct adjoint sensitivity analysis using the VP solver similar to the one used in the MIT and NAOSIM models. We included this into the discussion (lines 461-465).

Reviewer 4.2 Section headline: what does PIZ stand for? Reply: PIZ = pack ice zone. The abbreviation PIZ was introduced at line 204 of the former version of the manuscript. The section headline remains the same.

Reviewer: page 18 l355: there are no middle panels in Fig7, bottom panels? Reply: Corrected. See line 417 of the current version of the manuscript.

Reviewer: page 20 l369-371: "This issue is important because in realistic sea ice forecasts, improper prediction of Ptr may result for mechanical damage of ships due to extensive sea ice compression." should be part of a discussion Reply: Actually the similar sentence was already in the Conclusion in the former version of the manuscript. See lines 425-426 of the former version. So, we removed it from section 4.2

Interactive
comment

Reviewer: l391: "OGCM inverse modeling was found to be inefficient, but a simpler stabilization based on Newtonian friction appeared to work well." It's not clear how this was done Reply: More details are now given in the Appendix. Equations A18-A24.

Reviewer: l394: where was this shown? Reply: To the best of our knowledge, there is no analytical proof of (conditional) stability of the linearized VP rheology in two dimensions. However, there are numerical indications of its stability in MIT and NAOSIM Models containing ice dynamics with VP rheology. The statement was expanded (lines 462-465).

Reviewer: page 21 l405: (10-15): where does this range of numbers come from? I counted 7: initial conditions for u, h, A, kT, k2, e, P* Reply: At the end of the subsection 2.4 we state that RP control fields were specified on a coarser resolution grid and bilinear interpolation was applied to project the RP values on the grid where SIT/SIC and SIV values were defined. So, the maximum number of unknowns (dimension for the of the control vector) associated with initial conditions was 75*30*4. The corresponding numbers of unknowns for kT, k2, e, and P* were respectively 6*3=18, so that the total part of the RP control did not exceed (e + P*) 18*2=36. We included more details on this issue in section 2.4 and referred to them in the appropriate place of Section 5 (lines 245-250)

Reviewer: l425, algorithmically, assimilating ice drift should not have too much of an effect on the model drift, because the information is lost in the EVP iteration: the result of the EVP solver does not really depend on the initial conditions at the beginning of the iteration, but only on the forcing and solver parameter. That is why adjusting the solver parameters P* and e has such a large impact on the ice drift. I think the experiments at least provide some evidence for this interpretation Reply: We agree with the Reviewer. A related discussion was included in introduction (lines 83-85). See also lines 503-505.

Reviewer: page 22 l455: the solution technique outlined here is not what is usually done in implicit VP-solvers. P = P(h,A) is usually held fixed as the value of the pre-
vious timestep (although this is not a requirement, see IMEX in Lemieux et al. 2014, doi:10.1016 /j.jcp.2014.01. 010), but  is updated in the non-linear Picard iteration making the entire iteration very stiff (hence, the attempts with JFNK, and their failure, that are also cited in this paper). If  is held at t-1, then the entire problem is linearised and much simpler to solve, and I would agree with this assessment. But it refers to a system that is not used in practice, and would give very different results, too.

Reply: We agree with the Reviewer. To explore the issue, we performed an experiment with 1D VP forward and TL models (only ice thickness field was disturbed, while the sea ice concentration was kept 100% everywhere) using the modified procedure featuring ten applications of the GMRES and ten  updates on every time step. This procedure is similar to Lemieux et al., (2008). Results are shown in the Figure below.

Figure 2: Solutions (velocity and thickness) of the 1D VP forward and TL model (normalized) derived with one (four top panels) and ten (four bottom panels) outer loop iterations (application of the GMRES with  updates). The sea ice with non-uniform thickness and 100% concentration was forced by converging wind schematically shown at the upper left pane

It is evident that the solutions of the forward model did not change significantly, probably due to relatively short period of the model integration. There are more changes in the corresponding TL solutions, but still solutions are similar to those obtained with simplified procedure (without updates). More importantly, the TL code does not reveal any instabilities implying that the adjoint model is stable with updates as well. We are aware that application of the GMRES is not very popular, but this simple experiment with 1D VP model still suggests that VP solver should be more suitable for the variational sea ice data assimilation applications. Note also, that MIT and NAOSIM sea ice models use VP solvers and (as far as we know) their authors did not report any instabilities in the TLA codes of the respective sea ice models. We are currently working on the 2d VP model planning to investigate stability of its TL code using numerical spectral analysis of the respective matrices. We modified the text in the appendix B

(lines: 651-654).

Reviewer: page 25 l532: in press JTECH, appears to be online: https://doi.org/10.1175/ JTECHD-18-0239.1 Reply: This manuscript was published. Reference corrected.

Reviewer: l7 a Newtonian, and 21 more spelling/language corrections (lines 20, 29, 34, 36, 50, 65, 73, 100, 117, 163, 203, 204, 215, 221, 223, 247, 277, 312, 317, 318, 396. Reply: All corrected

REFERENCES

Goldberg. and Heimbach, 2013: The Cryosphere, 7, 1659-1678. Lemieux and Tremblay 2009: JGR, 114, C05009. Lemieux, Tremblay, Thomas, Sedlacek, and LMysak, 2008: JGR Oceans, 113, C10004, doi:10.1029/2007JC004680. Lemieux and Dupont 2020: Geoscientific Model Development, https://doi.org/10.5194/gmd-2019-284 (in press) Nicolsky, Romanovsky and Panteleev, 2008: Cold Regions Science and Technology. 55, 120-129. Ungermann, Tremblay, Martin and M. Losch, 2017: JGR Oceans, 122, 2090–2107. Vossepoel, F.and Van Leeuwen,.2007, MWR, 135(3), DOI: 10.1175/MWR3328.1 Yaremchuk, Nechaev and Panteleev, 2009: Monthly Weather Review 137, 2966-2978.

Please also note the supplement to this comment:
https://www.the-cryosphere-discuss.net/tc-2019-219/tc-2019-219-AC1-supplement.pdf

[Figure]

**_Reviewer :_**  _It's not clear which pseudo data are assimilated. Fig2 is strange, with noise-like stripes near x=600km, y < 20km after 3 days._

**Reply:**  In the KT experiments presented in Fig. 2 (with the exception of the dashed line in Fig. 2e) initial conditions for SIT and SIC were not optimized. If initial conditions for SIT, SIC are not perfect, we may use (dense) SIT/SIC observations and optimize them as well. Behavior of the cost function in this experiment is shown in Figure 2e by the dashed line. Emergence of the noise-like features after 3 days of integration in the previous version of the manuscript were due to several reasons:

   a) SIC/SIT initial conditions along the northern and southern boundaries had the form of a narrow tongue (1 grid point wide and  3 grid points long ).
   b) Dispersive properties of the Lax-Wendroff advection scheme.
   c) Effects of the Matlab function PCOLOR utilized for plotting. By default, this function uses cubic spline which tends to produce grid scale noise in the regions of sharp gradients.

To diminish these effects, we slightly modified the shape of the initial condition along the northern and southern boundaries and utilized a different plotting procedure. Note that this feature is absent in K2 experiments because initial conditions were smooth.  Note, also that new initial conditions result to more efficient minimization because the "true" solution is less noisy.

[Figure]

See **Lines 239-243 , 279-281, 295-298 and modified Figure 2.**

**Fig. 1.**

**_Reviewer:_** *page 22 l455: the solution technique outlined here is not what is usually done in implicit VP-solvers. P = P(h,A) is usually held fixed as the value of the previous timestep (although this is not a requirement, see IMEX in Lemieux et al. 2014, doi:10.1016 /j.jcp.2014.01. 010), but Δ is updated in the non-linear Picard iteration making the entire iteration very stiff (hence, the attempts with JFNK, and their failure, that are also cited in this paper). If Δ is held at t-1, then the entire problem is linearised and much simpler to solve, and I would agree with this assessment. But it refers to a system that is not used in practice, and would give very different results, too.*

**Reply:** We agree with the Reviewer. To explore the issue, we performed an experiment with 1D VP forward and TL models (only ice thickness field was disturbed, while the sea ice concentration was kept 100% everywhere) using the modified procedure featuring ten applications of the GMRES and ten Δ updates on every time step. This procedure is similar to Lemieux et al., (2008). Results are shown in the Figure below.

[Figure]

Figure: Solutions (velocity and thickness) of the 1D VP forward and TL model (normalized) derived with one (four top panels) and ten (four bottom panels) outer loop iterations (application of the GMRES with Δ updates). The sea ice with non-uniform thickness and 100% concentration was forced by converging wind schematically shown at the upper left pane

It is evident that the solutions of the forward model did not change significantly, probably due to relatively short period of the model integration. There are more changes in the corresponding TL solutions, but still solutions are similar to those obtained with simplified procedure (without updates). More importantly, the TL code does not reveal any instabilities implying that the adjoint model is stable with Δ updates as well. We are aware that application of the GMRES is not very popular, but this simple experiment with 1D VP model still suggests that VP solver should be more suitable for the variational sea ice data assimilation applications. Note also, that MIT and NAOSIM sea ice models use VP solvers and (as far as we know) their authors did not report any instabilities in the TLA codes of the respective sea ice models. We are currently working on the 2d VP model planning to investigate stability of its TL code using numerical spectral analysis of the respective matrices. We modified the text in the appendix B (lines: **651-654**).

**Fig. 2.**

---

## Author Comment (AC2) · 4 Mar 2020

Manuscript TC-2019-219-RC1

Parameter Optimization in Sea Ice Models with Elastic-Viscoplastic Rheology by Panteleev et al

Response to Reviewer 2

We would like to thank the Reviewer for useful comments that helped us to significantly improve the manuscript.

MAJOR COMMENTS:

Reviewer:Applying adjoint methods especially in the sea ice model is a difficult work

and the readers are keen to see if there really are some advances on this field. The analytic differentiation as reported by the authors, the damping term and even the codes should be publicly accessible at least from the appendix or the supplement materials, though there is something still not clear for me, but they are not. Reply: Following the Reviewer's request, we now provide a detailed description of the TL model in the Appendix A. We provide the description of the part of the ADJ model as well. The full adjoint model operator is the transposed to the TL model operator and can be derived easily.

Reviewer: the manuscript is not well-prepared. It seems to be a draft on its first version. The context is little bit tedious on some unnecessary parts from my feeling and ignores too many details that, however, should be elaborated. I guess the co-authors even did not really go through the manuscript, let alone help to improve the text. Too many small grammar mistakes that, however, can be easily corrected by grammar check in MS word or spelling check if you use the Latex! All your citation styles in the text should be also taken care of. Reply: The revised manuscript is thoroughly corrected. We also strongly apologize for the misspelling issue. We current version of the manuscript was checked by two of our native English co-authors.

MINOR COMMENTS:

Reviewer: L24: About abbreviations such as SIM, SI, LFI:, I indeed find it does not improve but reduce the readability of the text. Reply: The number of abbreviations was reduced significantly. In particular we removed abbreviations SIM, SI and LFI from the revised version of the manuscript. .

Reviewer: L33: As above, the citation style. Add '.' after 'al'. Reply: Corrected.

Reviewer: L35: 'the sea ice rheology is defined by . . .' needs to be rephrased. I think the word 'defined' is not proper Reply: Corrected. Line 39

Reviewer: L56-58: Better to remove this paragraph. I did not find any connection with

the context. The stochastic parameters are locally varied, but this is actually another story when stochastic effects are considered. Reply: The paragraph was removed.

Reviewer: L90, 95: what are the div and tr? Please state clearly in the text!. Reply: The notation is clarified (line 118 of the revised manuscript). Eq. (6) was reformulated in terms of the trace operator only to remove the necessity of introducing the determinant.

Reviewer: L117: Please break this long sentence into shorter ones and elaborate how did you deal with the analytic differentiation of the equation in the appendix. I also wonder how is the ïĄĎ , which is highly non-linear, be processed. Reply: The sentence is rephrased. The new Appendix A describes the numerical scheme and the TLA codes structure in much more detail.

Reviewer: L130: About the 'TL code', since the model is not such complicated, please make all your 'TL code' publicly accessible for better reproducibility for the community. Reply: The current version of the manuscript includes a more detailed description of the TLA models. Full adjoint code can be easily derived by transposition of the operator of the TL model. We now make a detailed outline of the respective procedure in the Appendix. Regarding the public access, the NRL regulations imply that the codes could be obtained only after filing an official request in the NRL security system.

Reviewer: L144: Regarding the 'spatial spectrum', it's not clear that what kind of spatial spectrum you refer to. Reply: We meant the local spectrum of the sea ice thickness (SIT) component of the state vector in the direction orthogonal to the ridges. The clarifying correction has been made (line 168-180).

Reviewer: L148: Regarding the 'Newtonian friction term', please implicitly show the equation and the damping time scales that you used Reply: Now described in the Appendix A (equations A18-A20). The damping scales are given can be found in lines 182-183 of the current version of the manuscript.

Reviewer: Figure 1: Please consider to use dotted line. it can obviously show how

many experiments you did. Reply: Done.

Reviewer: Section 2.3: I would significantly simplify this section, since only the observation errors are used. You do not need to introduce all these. When I read this section, I was thinking about the experiments are dealing with the real observations. But actually, I think for the ideal experiments, these observation errors only set a reference.

Reply: We do not completely agree with the Reviewer, because specifying observational errors is critical for correct formulation of the OSSEs. In particular, the weights of various model-data misfit terms in the cost function are inversely proportional to the errors of the respective observations. Therefore, we specify error levels similar to those in the real observations and provide detailed estimates of the errors in satellite products that are widely used in sea ice data assimilation systems. We underline this on the first two lines of this section (Line 206-207).

Reviewer: L224: the assimilation window is really short. I just wonder if the experiment results show sensitivities on the assimilation window. Reply: We now put more emphasis in articulating the objectives of the study focused on the improvement of the short-term ice forecasts (lines 96-100, 485-494) in the ice pack and near the ice edge. These regions are subject to variability at time scales of several days, so 3-day data assimilation window looks reasonable. Additional experiments with longer (5-day) window demonstrated similar results.

Reviewer: L232: what is 'ïĄů'? Reply: The sentence text has been changed to remove ïĄů and improve the clarity of presentation.

Reviewer: L235: what is 'DAS' ? Reply: Abbreviation removed.

Reviewer: Section 3.1: the configuration of the experiment is not clear. For example, is the initial SIC condition symmetric? It seems not from Figure 2a. How is the boundary condition? And in the text, the authors should explain why the spatial distribution of the polynya is not symmetric over the y-axis. Coriolis effects or the initial condition

effects? Reply: The initial conditions for SIC and SIT were symmetric. Emergence of the noise-like features after 3 days of integration and some asymmetry for the day 0 in the previous version of the manuscript were due to several reasons: a) SIC/SIT initial conditions along the northern and southern boundaries had the form of a narrow tongue (1 grid point wide and 3 grid points long ). b) Dispersive properties of the Lax-Wendorff advection scheme. c) Effects of the Matlab function PCOLOR utilized for plotting. This function inherently use cubic spline which produce some artificial "noisy" features. The polynya is non-symmetric due to the Coriolis force. We added respective comments (see lines 259-256, 265-266) in the revised version of the manuscript and slightly modified initial SIC and SIT in this experiment: Figure 1 (new figure 4)

Reviewer: L275: GYRE-0/W. Elaborate the meanings of the abbreviations Reply: The meaning of these abbreviation is explained now in lines 239-242.

Reviewer: L278: It's not clear why you use such weird initial SIC distribution Reply: We use the same kind of the sin function as we used in our previous publication Stroh et al, (2019), but two–dimensional. It allows to have regions with high/low concentration and thickness simultaneously.

Reviewer: L293: Please say clearly how you mimic the realistic observations, just their magnitude? Reply: Our "synthetic" observations is a sum of the observations derived from "true" solution plus some noise. The magnitude of noise had the realistic values discussed in Section 2.3. Some additional clarification was added to the manuscript (Lines 345-349)

Reviewer: L310: I wonder whether you optimize Crh first then the initial conditions, you could get the same conclusion. Reply: In strongly nonlinear inversions uniqueness of the solution cannot be guaranteed, because the cost function may have multiple minima and the optimized solution in this case depends on the first guess values of the control variables and the initial descent direction. In our case, finding a physically sensible first guess control vector is a necessity, which was realized in the form of

three-step minimization.

Reviewer: L338: 'the western part agrees well with true e distribution'. Actually, only part of. I think the authors could just say something like " show parts of agreement". L343: "and therefore has a minor rheological impact of the sea ice dynamics". That is not the case, as most parts in fig 5b still have SIC >= 0.95. Reply: We meant a decrease of the impact in the regions with SIC<0.8 where the RPs are very difficult to reconstruct from surface observations of SIC, SIT and SIV. Design of the PIZ experiments had the major incentive to have a closer examination of observability of RPs in pack ice. The respective clarification has been made (lines 395-400).

Reviewer: L414: the authors never defined RMSE Reply: Corrected (lines 493-494)

Reviewer: L431: I do not know why it is worth to address the realistic observation errors are used. Reply: We consider that our final goal is to develop the 4Dvar data assimilation system for the sea ice model which will be capable to retrieve rheological parameters from realistic observations. Because of that we are trying to underline, that using available SIV/SIC observations with realistic error bars and SIT observations with twice smaller (0.3m) errors than currently available, can be successfully utilized for this purpose. We suggest that that accurate SIT observation are already available for some moorings and will be available from MOSAiC and from the future satellites.

Reviewer: Conclusions: it's currently too long. Please try to simplify what you really want to say. Reply: Actually, this section contained both "Conclusions and Discussion", so we changed the title accordingly. Other reviewers recommended to include more discussion here, so the section was expanded.

Reviewer: Spelling/language corrections (lines 66,101, 110, 194, 292, 307, 318, 368, 388). Reply: All corrected.

Please also note the supplement to this comment:
https://www.the-cryosphere-discuss.net/tc-2019-219/tc-2019-219-AC2-

supplement.pdf

**_Reviewer:_** _Section 3.1: the configuration of the experiment is not clear. For example, is the_
_initial SIC condition symmetric? It seems not from Figure 2a. How is the boundary condition?_
_And in the text, the authors should explain why the spatial distribution of the polynya is not_
_symmetric over the y-axis. Coriolis effects or the initial condition effects?_

**Reply:** The initial conditions for SIC and SIT were symmetric. Emergence of the noise-like
features after 3 days of integration and some asymmetry for the day 0 in the previous version of
the manuscript were due to several reasons:

  a) SIC/SIT initial conditions along the northern and southern boundaries had the form of a
     narrow tongue (1 grid point wide and 3 grid points long ).
  b) Dispersive properties of the Lax-Wendorff advection scheme.
  c) Effects of the Matlab function PCOLOR utilized for plotting. This function inherently use
     cubic spline which produce some artificial "noisy" features.

The polynya is non-symmetric due to the Coriolis force. We added respective comments (see
**lines 259-256, 265-266)** in the revised version of the manuscript and slightly modified initial SIC
and SIT in this experiment:

[Figure]

Figure 1 (new figure 4)

**Fig. 1.**

---

## Author Comment (AC3) · 4 Mar 2020

Manuscript TC-2019-219-RC3

Parameter Optimization in Sea Ice Models with Elastic-Viscoplastic Rheology

by Panteleev et al

Response to Reviewer 3

We would like to thank the Reviewer for useful comments that helped us to significantly improve the manuscript.

Specific comments on scientific quality:

[Figure]

Reviewer:. On the clarity and evidence provided to support the experiment results, for scientific reproducibility purpose, I think the derivation of the equations of the tangent linear and the adjoint models should be made available, likely in the Appendix, in addition to only descriptive wordings in the main text (Section 2.2), so the readers can assess the impact of the linearization and damping on the sensitivities and reconstructions. Reply: In the revised version of the manuscript, we provided a detailed description of the TL and adjoint models in the Appendix A.

Reviewer: On basing their development of the TL and adjoint codes on the EVP rheology, can the authors discuss the physical meaning of their results in the context of published works reporting issues on convergence with the EVP rheology, e.g., Lemieux et al., [2012], Losch and Danilov [2012]? In addition, can the authors discuss how relevant/applicable/adaptable their TL and adjoint code development would be in light of the availability of more recent modified EVP solvers, e.g., Koldunov 2019? Reply: Following the Reviewer's request, we added discussion of the subject to section 5 (lines 456-461, 467-471).

Reviewer: Specific to the short assimilation window (3 days), the purpose of the work is not clearly articulated, other than to point out that they are extending on previous works (e.g., of Stroh et al., 2019). Is the goal, given the expected non-linearity, to achieve short-term (days) forecasting? Reply: We now put more emphasis in articulating the objectives of the study in sections 1 and 5 (lines 95-100, 509-515).

Reviewer. On the same subject of the short assimilation window, I think the authors need to provide an assessment on the meaning of the "optimized" parameters. Specifically, are the adjustments and optimized values reflect physical values relevant to various sea ice regimes or whether they are merely for the purpose of curve-fitting. In addition, due to the 3-day window, what does it mean if these optimized 2-D fields of the ice parameters change / are discontinuous every 3-day or so? Reply: We assume that RP fields do reflect the relevant changes in sea ice regimes. In a recent personal discussion, Peter Van Leuven mentioned that according to his preliminary results P*

have a strong seasonal variability, but these results were never published. Temporal variability of the sea ice has even smaller time scales in the MIZ zone where the pancake ice can be replaced with large floe ice in less than a week (e.g. Panteleev et al., 2019). From our point of view it is natural to assume different P* and e for 0.1-1m floes and sea ice floes with spatial dimension larger than 0.5-1 km. Similar scales of temporal variability can be found from the analysis of the landfast ice maps: strong wind may still move the grounded floes offshore and the newly formed ice will be unable to form the keels needed for keeping landfast ice in place for some period of time, even in case of sufficient thickness. Also, currently most of the sea ice observations are available daily. Because of that, we do not see a necessity to increase the DA time window for the period more than 1 week. More discussion on the choice and possible impact of the 3-day assimilation window is given (lines 96-98, 509-517).

Reviewer The adjoint gradients, where stable, are powerful in that they reflect dynamical connections, and thus allows one to extract meaningful physical connections relating the control space (rheology and ice dynamics parameters and initial condition) and the sea ice state (fast ice, seasonal/marginal ice, thick, thin, etc.). However, due to the damping/regularization, it is not clear if these adjoint gradients contain physics, or whether they are simply numeric for use in a misfit reduction procedure. For transparency purpose, it would be good if the authors can provide a couple of figures on the gradients. Reply: Due to non-linearity of the cost function with respect to the control variables and the first guess, the gradient may not be physically meaningful on a given iteration. It also strongly depends on the utilized minimization algorithm (M1QN3), due to complicated nature of the cost function behavior near local minimum. As an example, below we provide the averaged gradient over the 10 minimization iterations with respect of the k2 control field from the experiment K2-OSSE. Comparison of the gradient distribution with Figure 3 from the manuscript reveals the region of negative gradient in the southwestern corner of the domain, which agrees well with the reconstructed k2 distribution. Note however, we do not know the magnitude and direction of the increment which M1QN3 applies to update the control vector using the gradient supplied to

M1QN3 on each iteration. In our opinion, this is a natural result, because otherwise it would be impossible to reconstruct the distribution of k2 starting from constant first guess and obtain the land fast ice region similar to the true solution.

Figure 1: The cost function gradient with respect of k2(x,y) averaged over 10 iterations. Note also, that this gradient map is in "full space" and it should be re-interpolated on sparse grid where the k2 control is defined.

Reviewer The authors mentioned why the relatively highly important parameters k1 cannot be part of the control space due to the non-linearity. Due to this reason, I believe the results in this manuscript is incomplete: I think there should be a discussion, and perhaps at least 1 or 2 sets of additional experiments conducted identically to those for k2 and P*, but for different k1 values, to gauge how sensitive their optimized k2 and P* are to other important parameters. In other words, one would like to understand whether results presented in this manuscript are robust and physically meaningful (e.g., the adjoint gradients are physical, the optimized rheology parameters yield useful information about their dependence on ice regimes), or whether they contain no physical meanings beyond curve-fitting.

Reply: We do not completely agree with the Reviewer: our statement was that optimization of k1 requires an additional parameter r , which controls the "steepness" of the approximation of the Heaviside function in eq(8). Typically that can be done through the arctangent of some oother smoothed vession of the Heaviside function. Currently, we are working of the 2D VP TLA 4Dvar approach and plan to investigate this option to optimize k1 in future. For your convenience we accomplished an OSSE with smaller k1true=2.5 and the similar k2=15 (see Figure below, or Figure 4 in the manuscript). As you can see, the decrease of the k1true does actually decrease the area where the landfast ice may be generated with given sea ice thickness and concentration. But, the value of the optimized k2 in the south-west corner is rather close (k2=12, and max(k2)=14) to the true value of the K2=15. The new figure and some discussion was included into the text. See lines 312-318, 488-490 and new figure 4 (below).

Figure 2: Results of the k2 optimization similar to the Figure 3 from the manuscript but with k1=0.25. Upper panels: True SIC and SIV with k2=15 at t=0 and t=3 days respectively. SIT distribution (meters) is shown by white contours in the left panel; Lower panels: The optimized k2 (c) and SIV and SIT at t=3 days (d).

Reviewer Technical corrections: There are many misspelled words, including misspelled authors in citations. Only a few I spotted are listed here, but the authors should run a spellcheck through this. Lines: 66, 85, 105, 169, 194, 368, 454, 455, 495, 506. Reply: We apologize. Corrections have been thoroughly made, two native English co-authors proofread the manuscript.

Reviewer Extra commas should be removed on lines: 319, 460. Need an extra ")" on line 442. Reply: Corrected.

Reviewer "SIT" was first introduced without spelling out on line 129. Reply: Corrected. SIT= Sea Ice Thickness is now defined in line 77

Reviewer "SIH" and "SIT" are scattered through the article, and I believe are meant the same thing, the authors should settle on one after defining them. Reply: Corrected throughout the text.

Reviewer Figure 2 caption: "Left panel shows..." should be "Right panel shows..". Reply: Corrected.

Reviewer Line 355: ".. in the middle panels.." should be ".. in the bottom panels.." Reply: Corrected.

Please also note the supplement to this comment:
https://www.the-cryosphere-discuss.net/tc-2019-219/tc-2019-219-AC3-supplement.pdf
* * *
*Reviewer* *The adjoint gradients, where stable, are powerful in that they reflect dynamical connections, and thus allows one to extract meaningful physical connections relating the control space (rheology and ice dynamics parameters and initial condition) and the sea ice state (fast ice, seasonal/marginal ice, thick, thin, etc.). However, due to the damping/regularization, it is not clear if these adjoint gradients contain physics, or whether they are simply numeric for use in a misfit reduction procedure. For transparency purpose, it would be good if the authors can provide a couple of figures on the gradients.*

**Reply:** Due to non-linearity of the cost function with respect to the control variables and the first guess, the gradient may not be physically meaningful on a given iteration. It also strongly depends on the utilized minimization algorithm (M1QN3), due to complicated nature of the cost function behavior near local minimum. As an example, below we provide the averaged gradient over the 10 minimization iterations with respect of the $k_2$ control field from the experiment K2-OSSE. Comparison of the gradient distribution with Figure 3 from the manuscript reveals the region of negative gradient in the southwestern corner of the domain, which agrees well with the reconstructed $k_2$ distribution. Note however, we do not know the magnitude and direction of the increment which M1QN3 applies to update the control vector using the gradient supplied to M1QN3 on each iteration. In our opinion, this is a natural result, because otherwise it would be impossible to reconstruct the distribution of $k_2$ starting from constant first guess and obtain the land fast ice region similar to the true solution.

[Figure]

**Figure 1:** The cost function gradient with respect of $k_2(x,y)$ averaged over 10 iterations. Note also, that this gradient map is in "full space" and it should be re-interpolated on sparse grid where the $k_2$ control is defined.

**Fig. 1.**

_**Reviewer**_ _The authors mentioned why the relatively highly important parameters k1 cannot be part of the control space due to the non-linearity. Due to this reason, I believe the results in this manuscript is incomplete: I think there should be a discussion, and perhaps at least 1 or 2 sets of additional experiments conducted identically to those for k2 and P\*, but for different k1 values, to gauge how sensitive their optimized k2 and P\* are to other important parameters. In other words, one would like to understand whether results presented in this manuscript are robust and physically meaningful (e.g., the adjoint gradients are physical, the optimized rheology parameters yield useful information about their dependence on ice regimes), or whether they contain no physical meanings beyond curve-fitting._

**Reply:** We do not completely agree with the Reviewer: our statement was that optimization of $k_1$ requires an additional parameter $r$, which controls the "steepness" of the approximation of the Heaviside function in eq(8). Typically that can be done through the arctangent of some oother smoothed vession of the Heaviside function. Currently, we are working of the 2D VP TLA 4Dvar approach and plan to investigate this option to optimize $k_1$ in future. For your convenience we accomplished an OSSE with smaller $k_1^{true}$=2.5 and the similar $k_2$=15 (see Figure below, or Figure 4 in the manuscript). As you can see, the decrease of the $k_1^{true}$ does actually decrease the area where the landfast ice may be generated with given sea ice thickness and concentration. But, the value of the optimized $k_2$ in the south-west corner is rather close ($k_2$=12, and max($k_2$)=14) to the true value of the $K_2$=15. The new figure and some discussion was included into the text. See _**lines 312-318, 488-490 and new figure 4 (below).**_

[Figure]

**Figure:** Results of the k2 optimization similar to the Figure 3 from the manuscript but with $k_1$=0.25. Upper panels: True SIC and SIV with $k_2$=15 at t=0 and t=3 days respectively. SIT distribution (meters) is shown by white contours in the left panel; Lower panels: The optimized $k_2$ (c) and SIV and SIT at t=3 days (d).

**Fig. 2.**

---

## Referee Report (RR1)

[referee-annotated manuscript omitted]

---

## Author Response (AR2)

Manuscript **TC-2019-219-RC1**

*Parameter Optimization in Sea Ice Models with Elastic-Viscoplastic Rheology*

by Panteleev et al.

**Response to Reviewer 1 (2nd revision)**

We took into account the comments provided by the Reviewer 1 and significantly modified our manuscript. In particular, two additional OSSEs (small $k_T$ and weaker wind) are conducted and discussed. Below are detailed replies for reviewer suggestions and comments:

**MAJOR COMMENTS:**

***Reviewer:*** *The motivation of the experiment choices is not clear (and when I said "somewhat random collection" I didn't mean the principle twin-experiment setups as the reply of the authors implies, but the choice of parameters, configurations etc):*
*The most important term in the sea ice momentum equations is the stress divergence term which includes the rheology formulation. Finding appropriate parameters $P^*$ and $e$ is a fundamental problem in sea ice modelling and has been the subject of many previous papers (many of which are now cited, others are missing, which could have been used for further motivation, btw I am sorry about the van Leuven comment, I didn't realise that this was never published) and future papers will follow. It is easy to motivate including these parameters into the control vector and the authors do so, but only after motivating extensively the inclusion of land-fast ice parameters. $P^*$ and $e$ are important and relevant for anyone working with sea ice models and should come first, in the motivation, in the presentation of the experiments, in the (basically absent) discussion, and in the conclusions.*

**Reply:**   We may agree, that $P^*$ and $e$ are the most important parameters for sea ice modeling, but our major objectives are prioritized by NRL goals, which currently are:
  a) Improve the sea ice forecast in the Land Fast areas.
  b) Create new data assimilation approaches and improve the sea ice hindcast/forecast.
These goals establish *our priorities,* so we first consider the optimization of the landfast sea ice parameter. As we understand, potential readers who are interested in *fundamental* problems may always skip sections related to the optimization of $k_2$ and $k_T$ and directly proceed to the section related to $P^*$ and $e$.  We would also like to note that there is a substantial difference between sea ice modeling and sea ice data assimilation. In particular, sea ice numerical modeling has the goal of developing an accurate model which simulates ice conditions for a given period under the assumption that model parameters are already known. Sea ice DA has the goal of reconstructing poorly-known model parameters from observations with the ultimate goal of improving the short/medium range forecast of a given model. As we wrote in lines 19, 72-75 (previous version of the manuscript) our goal is to find an appropriate way for the short-range forecast.  In the current version of the manuscript, we also described our priorities for the optimization of the land fast ice parameters. Our results indicate that 4Dvar LFI optimization can be achieved in 4-15 iterations. This can be easily realized for 4Dvar nested sea ice data assimilation models or even for Pan-Arctic high resolution sea ice models. The optimization of $P^*$ and $e$ requires more iterations (~50) and will require significantly more computational resources.

***Reviewer:*** *The discussion about land-fast ice parameterizations is relatively new (I am aware of some work in the early 2000, and then Rozman/Itkin around 2014/2015, and the cited papers by König-Beatty, Lemieux and the not cited Einar Olason, who was the first to do a realistic simulation of land fast ice following König-Beatty's idea of added tensile stress, the $k_T$ parameter, to a Mohr-Coulumbic yield curve, he, btw did not need any tensile stress for the elliptical yield curve to get fast ice in the Kara Sea), and to my knowledge CICE and the sea ice model of the MITgcm are the only sea-models with a larger user community to use the parameters. In this sense, it is totally unclear to me, why this parameterization receives so much attention in this paper, even before the more central rheology parameters P\* and e.*

**Reply:** We tried to include all (known to us) references related to the Land Fast ice parametrization, but, our manuscript is not about the developing a new approach on how to model land fast ice. We describe and validate how to optimize the spatially varying land fast ice parameters in the framework proposed by König-Beatty, Holland 2010 and Lemieaux et al., (2015, 2016) and the elliptical yield curve, so we do not understand why we should discuss the option of modeling Kara-Sea LFI in the framework of the Mohr-Coulomb yield curve? This has little (if any) relevance to the subject of the manuscript.
We would also like to note that *a)* the UAF ROMS-sea ice model does utilize the LFI parametrization proposed by Lemieux et al. (2015), and now they are working on including the LFI parametrization into the MOM6-sea ice model; *b)* your comments on various options to model LFI underline the difference between numerical modeling and data assimilation. Numerical modeling replies to the question: can we model "Kara ice" this way? For DA: "how can we improve the model state and/or forecast using a given model?"

***Reviewer:*** *Further, while the tensile strength parameter $k_T$ is part of the "rheology parameters", the other parameters $k_1$ and $k_2$ of Lemieux's scheme are definitely not "rheology", but parameters of an "ad-hoc" (and very effective) parameterisation of grounding. For comparison, I haven't heard anyone calling the wind stress or ocean stress term, which have the same form as the basal/bottomstress term for grounding, part of the rheology.*

**Reply:** This is a question of the terminology. In our understanding, the term "rheology" describes the behavior of non-Newtonian fluids or solid materials. We doubt that sea ice is completely equivalent to a non-Newtonian fluid, at least at small scales, but, nevertheless the term "rheology" is used for sea ice. To address your comment, we provide an explanation that $k_{1,2}$ are not really rheological parameters (lines 48-49), but for the convenience of presentation, we refer to all control parameters as rheological parameters or (RPs).

***Reviewer:*** *The generation of the TLA and regularization scheme is now described in the appendix, but the description is in an incomprehensible and unacceptable form. Instead of throwing most/all of the available information at the reader it would have been useful to sketch the derivation and the form of the TLA equations in a compact form. Then it would have also been possible to show and discuss the additional Newtonian damping term, which is now hidden somewhere in eq. A18-20 (I guess) with no discussion of the coefficient values etc. I would have tried to do this in a compact form, maybe more explicitly for a 1D example. In fact, if this were my paper, I would try to find a compact form with symbolic equations including the regularisation for the text in section 2 (the reason for this being the regularization term, not the TLA derivation, which one could in principle find elsewhere), and then maybe have everything spelled out in detail in the appendix. There's still text in section 2 that's unclear (i.e. the transpose of the sparse model matrix of the TL operator implies that this matrix is explicitly formed, which I don't believe is what the authors meant) and that is not supported by the*

*information in the appendix. The description of the adjoint of the Lax-Wendroff scheme is confusing, because it suddenly involves non-linear terms that are not introduced in A9 and 10. Again, I don't think that there's anything wrong with the presented material, but the presentation does not help much to understand the derivation of the model.*

**Reply:** The derivation of the TL and ADJ models is actually based on the differential of the Lagrangian function. So, given an explicit formulation of the finite difference forward model, the TL and ADJ models are defined *automatically* as soon as you specify the set of control variables and the cost function. The math explaining the adjoint is calculus 3 level (see, for example, https://tutorial.math.lamar.edu/classes/calciii/lagrangemultipliers.aspx). Actually, it is because of this "formal differentiation" property, that the TL and ADJ can be obtained by TAMC or other TL/adjoint compilers. Since 1970, there have been numerous publications on the subject. In a symbolic/ compact form, the concept of adjoint is best described in the publication by Le Dimet and Talagrand (1986), and in a more popular form by Errico (1997). So we refer a potential reader to these and others publications. We assume that the finite-difference form of the TL and ADJ equations, given in Appendix A, would be useful to a potential reader who has no hands-on experience in its derivation from the parent numerical model. To address all of these issues, we moved the finite-difference description of the model from Appendix A to Section 2.1, and, following the Reviewer's request, provided a short description of the assimilation method in symbolic form in Section 2.2.1. In this exposition, we tried to follow the notation used by Ungermann et al. (2017) in the explanation of the data assimilation approach using the Green function method. Taking into account that other reviewers already approved our manuscript and did not object to the detailed presentation of the TLA (Appendix A), we do not feel comfortable in removing it completely.

*Reviewer: The model in this work is an EVP model that does not have much to do with CICE, except for the EVP scheme and the B-grid. The default CICE strength parameteriZation is different from what is discussed here (and yes, CICE does have the option of using the P\* parameterization, but I have not seen many papers explicitly doing so except for the ECCO-group. Also, the "Cf" parameter in the Rothrock 1975 formulation is has a similar scaling function as P\*, but that needs to be argued for in the text, if you want to relate to CICE*

**Reply:** We mentioned the CICE6 model solely because all CICE applications at NRL use $P*$ parametrization. We do not think that an overview of all CICE parameterization options is a necessary requirement in better understanding the manuscript. To avoid possible confusion, in the revised version of the manuscript, we tried to minimize discussion relevant to the "similarity" between our simple model and CICE6.

*Reviewer: there is not nearly as much complexity here as there is in CICE in all aspects (the advection scheme in CICE is totally different from the Lax-Wendroff scheme used here.*
*Relating to CICE so often makes very little sense and should be dropped in most places. The presentation still requires work, mainly: (1) giving appropriate weight to different parts of the manuscript, (2) proper description of TLA generation and regularisation in a comprehensible compact form that has enough details so that it can be reproduced*

**Reply:** We commented on the use of CICE (lines 79-80) and removed excessive mentioning of the model from the text. The TLA generation and step by step description of the 4Dvar concept in symbolic terms are now given in the modified Section 2.2.

**MINOR COMMENTS:**

***Reviewer:*** *"Taking into account that sea ice observations are available daily, the experiments are configured for a 3-day data assimilation window in a rectangular basin" Unclear, why this is connected. I would think a 1 day window would be the natural choice if there's new data every data. Also Cryosat2 data is not available in gridded form every day. Needs a better explanation.*
**Reply:** The sentence was modified. See lines 7-8 of the current version of the manuscript.

***Reviewer:*** *line 21: (e.g. Heimbach et al, 2010; Zhang and Rothrock, 2003; Vancoppenolle et al. 2009; Massonnet et al. 2015) not clear if you refer to model or to models with DA capacities. Heimbach et al describes an adjoint sea ice model simulation (but not DA), Zhang+Rothrock, and Vancoppenolle describe a forward model, Massonnet et al describe solutions of models with some sort of DA or state estimation.*
**Reply:** We do not see any controversy in our text. See lines 20-25 of the manuscript after the first revision. From our point of view, it is obvious that on lines 21-22 we refer to both models and models with DA capabilities. To further clarify the point, we changed "and" to "and/or" on line 20 (current version).

***Reviewer:*** *l39: P\* and $\alpha$ are not part of a typical (default) CICE simulation. See major comment.*
**Reply:** The sentence is corrected and $P^*/e$ option is directly linked to the NRL applications of the CICE5/6 model (see lines 39-43, current version)

***Reviewer:*** *l45: (RPs) unnecessary abbreviation? I guess the authors have a different opinion.*
**Reply:** We agree with the Reviewer: we have a different opinion and prefer to use this abbreviation. A clarifying statement was made in the Introduction (lines 47-49.

***Reviewer:*** *l82: Note, that optimization of the RPs through the 4Dvar DA approach allows us to efficiently use all available sea ice observations including sea ice velocity, that are rarely used for assimilation in sea ice DA systems. The latter is due to weak sensitivity of the sea ice state with respect of the ice velocity (e.g. Kauker, et al., 2009). Roughly speaking, the 4dVar DA approach allows us to use sea ice velocity observations for adjustment of the RPs and/or atmospheric forcing in an appropriate manner resulting in a better sea ice forecast (Stroh et al, 2019). Not clear why the "weak sensitivity" can be brushed aside for this approach, which is essentially the same as that of Kauker et al 2009.*
**Reply:** We did not brush aside the *"weak sensitivity"*, we wanted to state that attempts to use velocities are rare because of the weak sensitivity of the model state with respect to the initial velocity conditions (Kauker et al., 2009). Augmenting the 4dVar control vector with RPs allows us to use sea ice velocity observations more efficiently. We modified the paragraph to improve the clarity of presentation (lines 89-91).

***Reviewer:*** *l89: I find OSSE a bit of an overstatement for the type of idealized experiments that are presented here. I would use this term OSSE only for realistic applications when the design of the observing systems is the subject, for example, resolution or accuracy requirements for a new satellite system to be designed, etc. This work is not about observations, but about TLA model development and testing.*
**Reply:** Indeed, the major goal of our manuscript is to investigate options on how to improve the short range sea ice forecast and we do that using the OSSEs with a simplified model. We consider the development of the TL and ADJ models is a technical goal, which formally can be resolved through automatic compilers in future. Note, that the proposed stabilization of the TLA models through the Newtonian dumping term is only a minor part of the study. We consider that

these idealized experiments are necessary because development of the TLA for the CICE is not easy and proper understanding of the problems/possibilities/advantages and disadvantages is a necessary step that should be made before proceeding in the development of the TL/ADJ code for CICE which may (or may not) include the respective RP controls. We do not see a principal difference between "OSSE" and the currently obsolete term "twin data experiment". Indeed, 10-15 years ago, we would be happy to use the term "twin data experiment". During the last decades, however, the term OSSEs became well established and, in our opinion, completely replaced the term "twin-data experiment". Note also, that the early OSSE references to "OSSEs" were made by Nitta (1975), and Arnold and Dey (1986) in studies which use simplified models. We have added these two references to the manuscript to clarify the issue of "inadequate terminology".

*Reviewer:* *ll96: Currently, satellite sea ice observations are typically available daily with a reasonably dense spatial resolution. Analysis of the SAR images (e.g. Panteleev et al., 2019) indicates that in the marginal sea ice zone, the pancake/cake ice with floe sizes of ~1-20 m may be easily replaced by floes exceeding 1 km in size in one week. As a consequence, we configured the OSSEs with a 3-day DA window assuming that such approach should have more impact on short therm sea ice forecast. Unclear reasoning.*
**Reply:** From our point of view, sea ice with floes 1-20 m and 1-2 km should have a different $P^*$ and $e$. In the OSSEs, we assume that $P^*$ and $e$ are time invariant, i.e. $P^*$ always has a maximum in the eastern and western parts of domain. But, indeed, $P^*$ and $e$ are the property of the sea ice, which moves. A 3-day DA window is reasonably "short enough" to assume that ice does not move "very far" from its original position. The DA window also cannot be very long for computational cost considerations. To prove this point, we conducted experiments with a 5-day long DA window. Results were similar but took almost twice as much computational time. We modified the sentence at lines 105-106. and added some discussion in Section 5.

*Reviewer:* *I agree that Eq(6) is correct, but it's unfortunate, because it hides the form of $\Delta^2$ as the sum of ice divergence + ice shear/e^2*
**Reply:** We do not think that this form is "unfortunate" as it emphasizes the invariance of $\Delta$ wrt to coordinate transformations.

*Reviewer:* *ll145: the explicit advective time step -> an explicit time stepping scheme why not say "a Lax-Wendroff time stepping scheme" here, as you don't say more in the appendix, either. Linear or non-linear?*
**Reply:** We moved the entire description of the finite difference scheme to Section 2.1. We used simplified Lax-Wendroff in order to simplify TL and adjoint code derivations. The applied isotropic diffusion produces a certain loss in accuracy in the approximation of advection, which has negligible impact on the results of the manuscript. Additional discussion of the advection scheme can be found at lines 569-570.

*Reviewer:* *I150: "The adjoint code was obtained by transposition of the sparse matrix in the code simulating the action of the TL operator on a perturbed state vector". In spite of the large amount of information in the appendix, this is still not clear. Is this matrix ever formed explicitly so that you can transpose it? Or is this done only symbolically by re-ordering the operations as in "automatic differentiation"?*
**Reply:** The matrix was neither formed explicitly, nor "symbolically obtained by re-ordering the operations". The code for computing its action on a vector was derived by taking the derivatives of the cost function (21) wrt to the state variables. We added a new Section 2.2.1 and provided more details in Section 2.2.2 (lines 234-236).

**Reviewer:** *Tab1: turning angle 0.4343 rad = 24.88 deg. Why such a number?*
**Reply:** The misprint is corrected to 0.436332 (25°). But we do not think this is essential from a scientific point of view.

**Reviewer:** *the proper way would be something like this: A similar instability of the TL EVP solver has been observed in the MITgcm sea ice model (M. Losch, personal communication).*
**Reply:** Corrected. Lines 251-252.

**Reviewer:** *l170: (e.g., Yaremchuk et al, (2009));*
**Reply:** Corrected.

**Reviewer:**
l180: more simple-> simpler? or just simple?
**Reply:** This part of the text was rewritten.

**Reviewer:** *l180: time scale T_d: is this the same as in eq(2)? That doesn't make sense*
**Reply:** Misprint corrected and more details describing regularization terms are added (lines 267-269).

**Reviewer:** *l182: a Newtonian*
**Reply:** This part of the text was rewritten. Lines 265-272.

**Reviewer:** *l200: MIT the model is called "MITgcm", not MIT, please correct everywhere*
**Reply:** Corrected.

**Reviewer:** *l216: missing parentheses around https://icesat-2.gsfc.nasa.gov/*
**Reply:** Corrected. Line 305.

**Reviewer :** *l219: A similar error level was …*
**Reply:** Corrected. Line 308.

**Reviewer:** *l225: "with spatial decorrelation scale of 150 km and temporal decorrelation scale of 7 days" two missing articles?*
**Reply:** The sentence was rephrased to improve clarity (lines 314-316 of the revised manuscript)

**Reviewer:** *l228: where do these numbers come from?*
**Reply:** These numbers were removed to prevent distraction and improve the conciseness.

**Reviewer:** *I maintain that the systematics between the first group and there rest remain unclear. These are two very different experiments and it's not clear why the LF scheme of Lemieux et al 2016 receives similar attention as the optimization of the more universal parameters P\* and e*
**Reply:** See our comment above. Attention to optimizing the parameterization of the LFI scheme is caused by current priorities of NRL research. Potential readers may always skip sections related to the LFI and proceed to *P\** and *e* experiments without any loss in understanding the material.

**Reviewer:** *ll247: The maximum number of control variables associated with the initial conditions (the number of ice model grid points occupied by the SIT, SIC and SIV fields) was*

*about 9000. The RP control fields were defined on coarser ($\delta xp=15$ (or 7)$\delta x$) grids with bilinear interpolation on the model grid of the respective OSSEs. Thus, the maximum dimension of the RP control vector never exceeded 36 elements. This remains very unclear.*

**Reply:** The paragraph was rewritten (see lines 337-340). Additional details are now given in the new subsection 2.1.2 (lines 193-194)

*__Reviewer:__ 261: $k_T$ was set to 0.6. I maintain that this is an unusually large value, except for explicit land fast ice simulations. You wouldn't use that universally in a Pan-Arctic simulations. (see Olason 2016, A dynamical model of Kara Sea land-fast ice, JGR, doi:10.1002/2016JC011638, where $k_T=0$ with the elliptical yield curve in spite of the focus on land fast ice). Tremblay and Hakakian (2006) report upper and lower bounds for compressive and tensile stress, I am not sure if this can be used to infer $k_T=0.6$. In König-Beatty and Holland, the noisy EVP solver prohibited land fast ice for smaller values of $k_T$, Lemieux et al (2016) discuss the value of $k_T$ and obtain better agreement with observations with small values of $k_T = O(0.1)$.*

**Reply:** You are correct, Tremblay and Hakakian (2006) reported estimates of maximum and medium tensile stress. However, as we understand, König-Beatty and Holland (2010), used Tremblay and Hakakian (2006) results and derived estimates $k_T \sim 0.5$-$0.6$ (page 188 from König-Beatty and Holland, 2010). In order to resolve this discussion, we conducted additional experiments with $k_T = 0.2$. See the new Figure 2 and discussion on lines 373-384 of the current version of the manuscript.

*__Reviewer:__ l270: "due to the absence of tensile strength in ice ($kT = 0$)" and due to the non-converged EVP solution, see König-Beatty and Holland*

**Reply:** We do not agree. We conducted the experiment with a much larger number (~2000) of sub-cycling iterations and got the same solution with ice moving eastward in the entire region. So, we attribute the absence of the LFI solely to the condition $k_T=0$.

*__Reviewer:__ Caption of Fig2 does not refer to (e) and and (f) explicitly (but to a-d)*
**Reply:** Corrected.

*__Reviewer:__ l284: 3m -> 3 m? In fact, the use of a space before a unit is inconsistent throughout the text*
**Reply:** Corrected.

*__Reviewer:__ l295: and SIV that's not different than the "true solution", where "initial velocities [...] were set to zero" (l288)*
**Reply:**
Corrected. Lines 397-398.

*__Reviewer:__ paragraph starting at line 319: Rather than using the numerical difficulties associated with the Heaviside function to waive the optimization of k1, one could use this analysis to argue for a smooth parameterization, that would also be more physical, because it is very unlikely that in a grid cell of a finite (usually large) extend of order (km) all ice ground at the same time and instantaneously (like in a cloud scheme not the entire grid cell of a AGCM will suddenly go through a phase transition). Instead a smooth transition is more appropriate as we expect the physics of sea ice to be smooth. I can see that this is beyond the scope of the paper, but I also think that the scope of the paper is fuzzy with these two types of experiments. It's not clear why the optimization of well established and very important parameters P\* and "e" is juxtaposed with two parameters of a relatively new parameterisations that are hardly used.*

**Reply:** As we outlined above, the structure of the paper reflects our (and NRL's) current research interests and goals. In the future, we will definitely address the problem of $k_1$ optimization. Also, it is not clear what you mean by "a smooth transition is more appropriate as we expect the physics of sea ice to be smooth." The LFI regions (with characteristic scales of 30-100 km) and polynyas (~10-20km), near the coastline, are typical phenomena observed in NRL's pan-Arctic CICE model runs at a 2-km resolution, that are characterized by discontinuities in ice thickness and concentration fields (i.e. cannot be considered as "smooth" in our opinion).

**Reviewer:** *Section 4 "eccentricity" is not what "e" is. It's has been defined correctly before.*
**Reply:** Corrected.

**Reviewer:** *l335: had the form of a Gaussian-shaped cyclone.*
**Reply:** Corrected.

**Reviewer:** *ll339: the trace of the stress tensor $P_{tr}$ = $-tr\sigma/2$ this is commonly called $\sigma_l$, the first invariant of the stress tensor, or divergence of stress. Not sure why this needs a new name here.*
**Reply:** We thought that the subscript "l" could be mixed with "1". An appropriate correction was made (lines 445)

**Reviewer:** *l348 "as most of the currently available observations". Rephrase so that it says what you mean (bias free observations are a common assumption in DA systems)*
**Reply:** Corrected. Lines 449-450.

**Reviewer:** *l364: definition of $S_u$: why did you use this norm and not the more common RMSE? Or is this implied? Unclear.*
**Reply:** We utilize different norms for scalars and vectors. Since only relative changes of these quantities are important, we do not see a problem here.

**Reviewer:** *l368: std($h_{opt}$ – $h_{true}$). In what sense is this form different from the definition of $S_u$? If it is different, why would you use different forms?*
**Reply:** We used STD for scalar values and 1-norm for the velocities (see previous comment).

**Reviewer:** *ll369: I find it much more plausible, that the impact on velocities is largest by optimizing rheology parameters that directly affect the momentum equations. The effect on the thickness field will take longer assimilation windows because the changed dynamics needs time to advect thickness (and concentration). [And later you say so, why not here?]*
**Reply:** We agree with the Reviewer: on lines 370-371 of the previous version of the manuscript, we actually made a similar statement: "Another possible reason is that $\mathbf{C}_{rh}$ is not well controlled by the initial SIT distribution at the time scale of the relatively short 3-day assimilation window." Note that under the simplifying assumptions of Kohl and Stammer (2004), (see also Panteleev et al., 2013) sensitivity is proportional to the accuracy of SIT observations and, therefore, will be large if the respective error variance is small enough.

**Reviewer:** *l380: "that spatial locations of extrema agrees well with true distribution" fix grammar*
**Reply:** Corrected.

**Reviewer:** *ll387: "that inaccurate position of the atmospheric cyclone" there's an article missing*

**Reply:** Corrected.

**_Reviewer:_** _l389: "( 0.1 m/s)" the extra space after "(" happens very often, but not always. I am not sure if this is intentional or not, but I guess, the space should be removed here and everywhere else._
**Reply:** Corrected.

**_Reviewer:_** _l396: the wind appears to dominate the optimization. Without proper wind, the optimization appears meaningless pointing to severe problems in the sea ice model parameterizations (why should they depend on the wind?). It thinks that's worth pointing out, maybe in the discussion._
**Reply:** We agree that wind is a dominant forcing in sea ice dynamics and, therefore, it is a major factor in both sea ice modeling and data assimilation. You may have perfect $P^*$ and $e$ but your sea ice model solution will be incorrect if it is forced by incorrect wind. Note, however, that without proper wind, any sea ice model solution becomes meaningless, but the DA solution may still be reasonable because wind forcing can be adjusted to the correct values through the assimilation of the SIV/SIC/SIT observations if they are accurate enough (e.g. Stroh et al., 2019). We added a short discussion on the importance of accurate wind forcing for obtaining reasonable results. See lines 640-644 of the revised manuscript.

**_Reviewer:_** _l400: to a less degree than e. Fix grammar_
**Reply:** Corrected.

**_Reviewer:_** _ll403: "To mimic these conditions, we conducted another OSSE with spatially and temporally invariant sea ice concentration A=1. Numerically this was achieved by removing the advection equation from eq. (4), and removing initial A0 from the control vector C0."That changes the model and makes it difficult to understand the generality of the results. Why not construct an experiment where the ice strength is strong enough (relative to the wind) so that the system does not move? That would be more "realistic"._
**Reply:** Actually, we do not see the problem here. $A=1$ may be interpreted as an additional very strong cooling, which removes areas where $A<1$. We actually tried different winds and always had some regions with $A<0.9$ somewhere. Your suggestion with the experiment with zero ice velocities is meaningless, because: a) If ice does not move from the wind, $P^*$ can take any value above some critical point which is actually defined by the wind amplitude; b) As sea ice data shows, ice is always in motion in the central Arctic. An extra sentence was added (line 511-512)

**_Reviewer:_** _l413 "about 0.5 N/m2" that's a lot!_
**Reply:** We agree. This is a rather strong wind corresponding to intense Arctic cyclones that are rather frequent phenomena in the Arctic Ocean. Note, this wind was applied only near the western and open boundaries. We also found a minor inaccuracy here: the wind actually increased from 16 m/s up to 20 m/s, which yields ~0.5 N/m$^2$ by the end of the data assimilation period. Thus _wind stress_ increased 1.5 times, _not wind_. To resolve the issue with wind stress amplitude, we conducted an additional experiment with a twice as small wind (~10 m/s) and stress about 0.15 N/m$^2$. See new Figure 9 and lines 513-518, 548-559- related to the results of the new experiment.

**_Reviewer:_** _l415: "SIT increases over almost everywhere" this raises the question of volume conservation. Does the model conserve volume, and is this important for the optimization._
**Reply:** As mentioned above, the condition $A=1$ can be physically interpreted as cooling followed by immediate sea ice production. So, formally, sea ice volume is not conserved. But we

do not understand how this is important for DA with a 3-day window. Theoretically, this problem can be important for very long DA windows, but again, this will be a problem with the model. Not the DA algorithm.

*Reviewer:* l418: maximuma-> maxima (a spell checker would have found this)
**Reply:** Corrected. Line 503.

*Reviewer:* l431: "because sub-optimal Ptr distribution fails" missing article
**Reply:** Corrected.

*Reviewer:* l431: maximums → maxima (used previously)
**Reply:** Corrected.

*Reviewer:* l435 "std[ P 2 (opt) − P (true)]" Ptr squared?
**Reply:** Corrected.

*Reviewer:* l437: in Figure 9e,f
**Reply:** Corrected.

*Reviewer:* ll439: "The effect could probably be attributed to the region with zero convergence along the western boundary where the rheology does not play a significant role." rephrase to be more specific.
**Reply:** The term **"**zero convergence" has been changed to "divergence": when the ice field diverges, internal stress becomes less important. A comment has been added to lines 544-545.

*Reviewer:* Section 5 Conclusions and discussion" promises the unusual order of conclusions at the beginning followed by a discussion (why not do it conventionally with discussion first, followed by conclusions?), but then starts with a long summary of the results where the actual conclusions are hidden in the details, and at the end there are two paragraphs of discussion. I suggest to rewrite this section for better clarity.
**Reply:** We do not see a problem here. Actually, the title of the section is "Conclusion and Discussion", not vice versa. For example, the first conclusion is that the common way to stabilize TL is not efficient, however the Newtonian scheme allows us to do this, and after, we provide some discussion that the VP solver allows us to avoid this problem. We do not understand, how this can be outlined in the opposite order. Similarly, the "conclusion" is related to the optimization of Land Fast ice and P*/e parameters. Perhaps the title of this section is confusing, and so we have re-titled the section as "Summary and Discussion". We also include some discussion related to the additional experiments.

*Reviewer:* l445 with respect to
**Reply:** Corrected.

*Reviewer:* l445: "all rheological parameters" k2 (and k1) is not a rheological parameter and "k_T" is not a parameter that is commonly used (i.e. different from zero)
**Reply:** We agree that $k_2$ and $k_T$ are not the rheological parameters, but from a physical point of view, sea ice rheology is also not "real rheology" which describes a Non-Newtonian liquid. See our explanation above.

*Reviewer:* l448: "Lemieux et al., (2015, 2016) and Konig Beatty and Holland, (2010)" (and elsewhere) remove "," before "("

**Reply:** Corrected.

**Reviewer:** l455: "Analysis of the TL approximation accuracy has shown that Newtonian stabilization has errors similar to the ones observed in the case of diffusion-based stabilization, and thus the Newtonian scheme can be successfully used in sea ice models based on the EVP solvers." This analysis/discussion would be very important to understand in detail, because it is potentially something that readers may want to apply themselves. However, the "analysis" is extremely short, leaving out important aspects, like a stability analysis based on the value of the damping coefficients, etc. It would be interesting to understand, why this term successfully stabilizes the system etc

**Reply:** We do agree that it would be interesting to analyze the effects of the proposed realization and we plan to work on this problem in future. Note, however, that the application of the conventional scale-selective Laplacian regularization was first published as an *ad hoc* method by Hoteit et al. (2005) (actually, it was applied earlier, but was not published). Some additional analysis was done by Hoteit et al. (2006), and the theoretical interpretation was done by Yaremchuk et al. (2015). So, it is typical when some technique is initially applied *ad hoc*, and after some time, it finds some theoretical background. We did provide some speculation why scale-selective regularization poorly works with ice (see lines 261-263 of the manuscript). From the viewpoint of applications, the Newtonian damping should be kept as small as possible and its value is usually defined by trial and error.

**Reviewer:** l471: Since you have cited Losch et al (2014) already, they could be cited here again, because there AD tools were already used to compute the matrix times vector operation necessary for the matrix free JFNK with FGMRES in a sea ice model, however without significant improvement over simpler finite difference schemes.

**Reply:** Corrected.

**Reviewer:** l480: "kT and k2, responsible for grounding and arching phenomena"  unless intended as a "chiasms", I would turn the order around so that the first symbol corresponds to the to the first explanation (k2: grounding, kT: arching)

**Reply:** Corrected.

**Reviewer:** ll491: "In the second group of OSSEs, we analyzed the possibility of reconstructing spatially varying sea ice strength P* and ellipse axes ratio e distributions." I would argue that this is the more important part of the paper and should come first.

**Reply:** As explained above, optimizing $k_2$ and $k_T$ was the top priority of the current research.

**Reviewer:** l500: collocated in the regions of strong… collocated with regions of strong … I am also not sure if "collocate" is the correct word here. I would use "co-locate"

**Reply:** Corrected.

**Reviewer:** l502: provides a slightly more accurate reconstruction

**Reply:** Corrected.

**Reviewer:** l504: "Accurate forecasting of Ptr is very important because it better informs avoidance of regions with excessive compressive stress." Not the right context.

**Reply:** Corrected. Moved to Summary and Conclusions section.

**Reviewer:** ll523: This is a totally generic and even inaccurate statement (in spite of its generality) that should be removed or adequately modified: On the one hand, it is not clear to me, how the MOSAiC observations that are very local should help to constrain a local model,

*where open boundaries and boundary conditions as control parameters should be an important aspect that is not even mentioned in this work. On the other hand, the MOSAiC observations are regionally too confined to serve as a serious data source for a large scale model. I do agree that MOSAiC will be substantial to future sea ice parameter studies, but mostly for slow thermodynamic process and for fast stress parameters, because local drift and deformation as well as stress measurements are obtained.*

**Reply:** The statement was removed at the Reviewer's request. However, from our point of view MOSAIC data can be helpful for $P^*/e$ analysis related to the 2-kilometer resolution in the NRL model.

***Reviewer:*** *l541: the slightly different form of \epsilon vs. \varepsilon is unfortunate, because both are "epsilon" and easy to confuse this with strain rate tensor.*
**Reply:** We modified the notation and put the dot over all the quantities related to the components of the strain rate tensor. So now, they are easily distinguished from time scale ratios.

***Reviewer:*** *Are Eq A7+8 solved implicitly? u^s and v^s appear in both equations on the left hand side*
**Reply:** Yes, they are. The respective matrices are block-diagonal with 2 x 2 cells and can be inverted explicitly.

***Reviewer:*** *l550: the Lax-Wendorff scheme the scheme is called "Lax-Wendroff"!!*
**Reply:** Misspelling corrected.

***Reviewer7:*** *l559: MIT -> MITgcm (Heimbach et al 2010)*
**Reply:** Corrected.

***Reviewer:*** *TAMS -> TAMC*
**Reply:** Corrected.

***Reviewer:*** *eq A18-24: I don't understand, why the time stepping algorithm has been used here to illustrate how the TLA works. It makes the entire presentation far more complicated and difficult to understand. The four of eq. A18-A24 is unacceptable. I don't think that any of the reviewers meant this, when asking for a better description of the TLA*
**Reply:** In your previous review, you requested of us to provide more details for potential reproducibility. From our understanding, reproducibility can be achieved either by accurate differentiation of the Fortran code as it is done in TAMC (or by hand) or by providing a relatively accurate description of the TLA models, as it is done in the Appendix. Because other reviewers considered the Appendix acceptable (had no objections against it), we do not feel comfortable in removing it. However, following your request, we added some schematic explanation in the new Section 2.2.1.

***Reviewer:*** *eq. A18-20 shouldn't the damping term be highlighted somehow? I am not even sure if -eps delta sig1 is the term*
**Reply:** It is one of the terms. A more detailed explanation is now given in the new Section 2.1.2. The terms are now highlighted in the Appendix A of the revised manuscript.

***Reviewer:*** *l584: "variational assimilation experiments we used the strong constraint state-space formulation of the problem, minimizing the cost function" doesn't the adjoint method start from the cost function defining the scalar product with respect to which the entire equations are*

*derived? The presentation appears backwards through the eyes of the practitioner, but for understanding the principles of what has been done it's not very useful.*
**Reply:** The basic principles of 4Dvar DA theory in geophysical applications has been extensively published since the 1970s. So, all details of the technique can be easily found. It takes significant space to explain everything and we do not feel that the format of the journal publication is the proper place for providing this information. For this reason, we outline only the basic principles and steps of the 4Dvar data assimilation in the revised Section 2, and provide additional references (Le Dimet and Talagrand, 1986 and Lewis and Derber, 1985).

*__Reviewer:__ l601: "with the additional range constraints for the selected control fields (Section 2.3)" If you are using bounds, a better choice would have been an L-BFGS algorithm (as in m1qn3) with bounds, e.g. L-BFGS-B*
**Reply:** We do not agree. We are familiar with the L-BFGS routine and actually tried it. We found that M1QN3, with bound constraints, was computationally more efficient and converged faster in most of the experiments.

*__Reviewer:__ l612: paralell -> parallel*
**Reply:** Corrected.

*__Reviewer:__ Eq A27-31: the symbol $\mathcal{I}^T$ is not introduced. I have no idea what this is. It should be the "adjoint" model, i.e. the transpose of the tangent linear model matrix, which is never explicitly formed*
**Reply:** The superscript $^T$ is a standard notation for the transposition of a matrix. The respective matrix $\mathcal{I}$ is actually introduced on line 599 of the original manuscript, so $\mathcal{I}^T$ denotes the transpose of the matrix $\mathcal{I}$ which performs bilinear grid-to-grid interpolation from the (coarse) control grid (where the control fields and their gradients are defined) onto the (finer) model grid.

*__Reviewer:__ l623: adjont -> adjoint*
**Reply:** Corrected.

*__Reviewer:__ l624: "the Newtonian damping given by the terms −εδσ1,2,3 in eq. (A18-A20)" this information should've come much earlier*
**Reply:** We agree. Now, these terms are discussed in the modified Section 2 (subsection 2.2.2). See lines 267-269.

*__Reviewer A77:__ l723: Lemieux,J.F., missing space*
**Reply:** Corrected.

**References:**

Ko˙hl and Stammer, D., 2004. Optimal observations for variational data assimilation. J. Phys. Oceanogr. 34, 529-541

Panteleev et al, 2014 Configuring high frequency radar observations in the Southern Chukchi Sea, Polar Science, (7), 2, P77-81.

Ungermann, M., L. B. Tremblay, T. Martin, and M. Losch (2017), Impact of the ice strength formulation on the performance of a sea ice thickness distribution model in the Arctic, J. Geophys. Res. Oceans, 122, 2090–2107, doi:10.1002/2016JC012128.

Manuscript **TC-2019-219-RC1**

*Parameter Optimization in Sea Ice Models with Elastic-Viscoplastic Rheology*
by Panteleev et al

**Response to Reviewer 2 (2ⁿᵈ revision)**

We would like to thank the Reviewer for useful comments that helped us to improve the manuscript.

**MINOR COMMENTS:**

***Reviewer: B1*** *L18,19: AO is specifically used for 'Arctic Oscillation', I would suggest not using this abbreviation..*
**Reply:** Corrected.

***Reviewer: B2*** *L42: reference for Konig Beatty and Holland is missing.*
**Reply:** Corrected.

***Reviewer: B3*** *Please check again all the citiation form, especially 'et al.'. For example, L65, L66*
**Reply:** Corrected

***Reviewer: B4*** *Eq(1,3), please confirm if h is the effective thickness*
**Reply:** Corrected

***Reviewer: B5*** *L200, using MITgcm instead of MIT*
**Reply:** Corrected

***Reviewer: B6*** *Eq (A25), extra plus '+'*
**Reply:** Corrected.

---

## Author Response (AR3)

We would like to thank Reviewer 1 for another thorough review the manuscript. Please find below our replies to your comments and critique.

**This is the 3rd review of "Parameter Optimization in Sea Ice Models with Elastic-Viscoplastic Rheology" by Panteleev et al.**

REVIEWER: The authors have addressed my concerns in their reply, but do not agree with my comments in many cases. I still think, that the work is an important contribution to the field, but I maintain that work needs a better presentation to receive the attention it deserves. Having said that the manuscript has improved further, in particular the description of the tangent linear model and its regularisation is now clearer than before. Many of my concerns and smaller comments have been rebutted. In my opinion, one can publish this now after minor revisions, but I believe this could be a much better paper if the authors would revise the aspects addressed in my previous reviews and in the following comments. I don't need to see this paper again, because I think that we have reached a point where I can not add anything new to the discussion.

In particular, in my opinion a research paper in a journal like "The Cryosphere" should not be motivated by the NRL goals or those of any other organisation. I understand that the funding and employer sets constraints for research, but the type of publication where the priorities of the funding organisation dominate is a report. In a research paper for the scientific community, the priorities should be set by scientific interest and not the funding interests.

The statement on page 3, l78: "The framework of an NRL research project to identify spatially varying Land Fast Ice Parameters in the CICE6 model defined the priorities and objectives of this study." sounds awfully awkward in this context. Instead of this statement the paper's presentation would improve dramatically by rearranging the "priorities", otherwise this paper will should like a report.

**Reply:** Optimization of the K2 and KT is our current research interest because it requires very limited number of iterations and thus can be easily implemented for CICE-6 model. Optimization of the e(x,y), and P(x,y) requires more iterations and probably cannot be applied for the regional and/or nesting sea ice models. Sea our additional comments at lines 606-608. Also, as stated in our previous reply, a potential reader may easily skip sections related to the K2, KT optimization so we do not think this should a big issue.

REVIEWER: New section on page 7 (2.2.1 Strong constraint formulation): Previous versions did not even show the cost function,

**Reply:** In the previous version, the cost function was presented in the Appendix A.

REVIEWER: so this section is definitely an improvement, but this new section is not what I had thought of. I agree with the authors, that it's not necessary to repeat textbook knowledge or previously published procedures in detail (and I think the author's make this sarcastically clear in their reply), unless there's something fundamentally different here. The new part is the regularisation by Newtonian damping. The description should be detailed enough to explain how and where this Newtonian damping is introduced.

**Reply:** In the previous version all details related to the Newtonian dumping were placed in the Appendix (eq.A8-A10) and the respective terms highlighted by bold font. In addition lines 270-271 explain that *"additional terms $\varepsilon_N$ $\sigma_i$, i=1,3, appear inside the square brackets of the linearized equations (13-15)…"* . We think that gives a clear understanding how the Newtonian dumping was used numerically.

REVIEWER: I think the first two paragraphs of section 2.2.1 are necessary (l180-l193), the rest only insofar, as it allows understanding the Newtonian damping. The discussion at the end is also good, but it still does not become clear if the authors used any of the automatic differentiation tools or not, if they first formulated the tangent linear model analytically and then discretised it, or used the rules of algorithmic differentiation to derive the tangent linear model from the discretised forward model, etc.

**Reply:** We are confused with this comment. Lines 232-234 (previous version) clearly say:

"The TL code was derived by **analytic differentiation** of the above mentioned **numerical scheme** in the vicinity of a background model trajectory. The adjoint code was obtained by implicit transposition of the sparse matrix in the code simulating the action of the TL operator $M_x$ on a perturbed state vector."

So, from our point of view, it is clear that we took the numerical formulation of the forward model and derived the TL analytically, i.e. by "hand". The words "implicit transposition" indicate that adjoint model was built as an operator, but not as "transposed matrix".

We think that the Reviewer could be confused by the sentence

Note, that both finite-difference TL and adjoint models are completely defined by the finite difference scheme of the forward model thus allowing application of the above mentioned (semi-)automatic TL/adjoint compilers.

We these sentences were modified in the revised version of the manuscript: lines 227-231, 233-234

REVIEWER: Using non-standard notation and terminology does not improve the readability, see comments in previous reviews, and the re-definition of "rheology parameters". (page 2 l48: "To simplify the presentation …" A term with a defined meaning is re-defined here in an unusual way to be more inclusive. Confusing the terminology doesn't make anything simpler. One could use a term that does not already have a defined meaning.) In general, terminology is often used in a "liberal" way.

**Reply:** We guess that you mean the confusing term "rheological parameters" which we applied for k1,k2,kt. As we mentioned in the previous revision, we formally agree: K1, k2, kt are not the "true" rheological parameters. Because of that, in lines 46-49 we now provide an explanation. Note also, that in the Abstract we clearly distinguish the LandFast Ice parameters and rheological parameters P* and e. We used a single term (RP) for the convenience of presentation and because from mathematical point of view there is no formal difference between the parameters k1,k2,kt and e, P. All of them are not prognostic variables and all of them somehow affect sea ice dynamics.

REVIEWER: Many (small) grammar mistakes remain (for example, missing or extra articles), and generally sloppy referencing (see below) still gives the impression of a hastily composed submission.

**Reply:** We asked our English native speakers put more attention for the English gramma. We hope that minimized the English gramma errors.

REVIEWER: more minor comments:
page 1
l5: "The feasibility of optimization of the and landfast sea ice and rheological parameters" something is missing in this sentence

**Reply:** We think this is Line 6. We corrected this sentence.

page 3
l62: the functions -> a function

**Reply:** Thanks. Corrected.

page 5
eq(8): In Lemieux et al. (2016) the argument of the Heaviside step function is is h-h_c, where h_c = A h_b / k_1, but their h is the mean thickness (volume per unit area), i.e. your (h*A), I think.

**Reply:** Yes. Thanks. Corrected.

l175: "differs from differs from"
fix repetition

**Reply:** Thanks. Corrected.

page 10
l278: shouldn't "N" be "X" now (after section 2.2.1)?
**Reply:** Thanks. Corrected

page 11
l291 and elsewhere "OSSE experiments". The "E" in "OSSE" already stands for "experiment", doesn't it?

**Reply:** Yes. Thanks. Corrected everywhere.

page 15
ll382: "This result suggests that accurate land fast ice modeling can be achieved by specifying non-zero kT only in the key regions and thus, there is no need to specify uniform kT as it was done in the experiments conducted by Lemieux et al. (2016)."

Interesting result. It implies that tensile strength is only required ***at*** the arch but not upstream of it. But how would it be possible to find a formulation that would achieve that (not knowing in advance where the arches occur)?

**Reply:** In operational oceanography, arching can be identified in multiple ways: e.g., through the analysis of SST, SAR images etc. It is also possible to define KT only in the regions with potential arching through the analysis of historical observations.

"In operational practice, the arching regions can be identified through the analysis of the SAR images and/or SST maps (e.g.  Ryan and Munchow 2016), or from the analysis of the historical sea ice maps from different sea ice data centers."  See lines  385-386.

References need work (formatting and completeness). I am listing a few instances, but there are definitely more:

**Reply:** Thanks. We knew, that some of the references may be in wrong format, but usually this problem is resolved before the final submission of the Latex and/or pdf file.

page 3
l80: (e.g. Posey et al., (2010), Metzger et al, (2014))

remove extra () around years

**Reply:** Corrected.

page 4
l94 remove "," in reference to Lemieux

**Reply:** Corrected.

l97: Stroh et al, (2019), replace "," by "."
**Reply:** Corrected..

l99: Lemieux et al.(2016). Insert space after ".".
**Reply:** *Corrected.*

page15, l348: Referencing scheme is inconsistent (replace "," by ".".?)
**Reply:** Corrected.

page 29
l641: couldn't find "Thorndike and Colony, 1982" in the references.
**Reply:** Corrected.

• Highlight, page 34
l750: Goldberg D.N and P.Heimbach, 2013: Parameter and state estimation with a time-dependent adjoint marine ice sheet model, The Cryosphere, 7, 1659–1678, www.the-cryosphere.net/7/1659/2013/ doi:10.5194/tc-7-1659-2013
Different referencing scheme than the other references.
**Reply:** Corrected.

l752: Harder,M., insert space
**Reply:** Corrected.

• Highlight, page 34
l770, 776, 824, 834, 836
in consistent capitalisation of title
**Reply:** Corrected.

• Highlight, page 35
l782 and 784, I believe it is "Le Dimet, F. X." and not "e Dimet …"
**Reply:** Corrected.

• Highlight, page 35
l798 Lemieux, J.-F., F. Dupont, is published:
https://doi.org/10.5194/gmd-13-1763-2020
**Reply:** Corrected.

• Highlight, page 36
l819: No authors for this reference? "Some analyses of observing systems simulation experiments in relation to the First GARP Global Experiment. GARP Working Group on Numerical Experimentation, Report No 10, 1-35. Plan for U.S. Participation in the Global Atmospheric Research Program, National Academy of Sciences, Washington, DC, 1969."
**Reply:** Corrected.

• Highlight, page 37
l860: Zhnag -> Zhang (???)

**Reply:** Corrected.